# Intelligent Agents with Emotional Intelligence: Current Trends, Challenges, and Future Prospects

## Abstract

The development of agents with emotional intelligence is becoming increasingly vital due to their significant role in human-computer interaction and the growing integration of computational systems across various sectors of society. Affective computing aims to design intelligent systems that can recognize, evoke, and express human emotions, thereby emulating human emotional intelligence. While previous reviews have focused on specific aspects of this field, there has been limited comprehensive research that encompasses emotion understanding, elicitation, and expression, along with the related challenges. This survey addresses this gap by providing a holistic overview of core components of artificial emotion intelligence into one cohesive map for researchers. It covers emotion understanding through multimodal data processing, as well as affective cognition, which includes cognitive appraisal, emotion mapping, and adaptive modulation in decision-making, learning, and reasoning. Additionally, it addresses the synthesis of emotional expression across text, speech, and facial modalities to enhance human-agent interaction. This paper identifies and analyzes the key challenges and issues encountered in the development of affective systems, covering state-of-the-art methodologies designed to address them. Finally, we highlight promising future directions, with particular emphasis on the potential of generative technologies to advance affective computing.

## 1 Introduction

Developing intelligent agents that possess human-level intelligence is a key goal in the field of Human-Computer Interaction (HCI) and general artificial intelligence (Jeon, 2017). In this survey, an intelligent agent is defined as an autonomous computational system that perceives information from its environment, processes the received information through reasoning or learning mechanisms, and generates actions or responses to achieve specific goals. Unlike conventional task-oriented AI systems, emotionally intelligent agents additionally incorporate affective capabilities that enable them to perceive, interpret, reason about, and express emotions during interactions with humans. These agents can be implemented as conversational systems, embodied agents, or multimodal interactive systems that utilize textual, auditory, visual, and contextual information. A crucial aspect of achieving more natural and effective human-agent interaction is the incorporation of emotional intelligence, which is essential for human cognition and social interaction, into these intelligent agents. Emotional intelligence encompasses three interrelated capabilities. First, emotion understanding involves accurately detecting and interpreting affective signals; for example, recognizing when a user is feeling frustrated during an interaction by analyzing their tone of voice or facial expressions. Second, emotion elicitation and experiences refer to interpreting the causes, context, and implications of emotions for both the individual and the interaction. For instance, an agent can perform a cognitive appraisal of the surrounding environment and contextual factors to infer its internal emotional state, which subsequently guides appropriate decision-making in the given situation. Third, emotion expression encompasses the capacity to generate, modulate, and convey appropriate emotional responses in a socially meaningful way, such as responding with a reassuring message or a sympathetic tone when the user is upset.

Affective Computing, coined by Rosalind Picard (Picard, 2000), emerged as a discipline dedicated to equipping machines with emotional intelligence, enabling them to recognize, interpret, and respond to human

emotions. By embedding emotional intelligence into intelligent agents, affective computing facilitates more naturalistic, adaptive, and socially competent interactions, which in turn enhance user trust, engagement, and satisfaction (Zall & Kangavari, 2024). Such emotionally intelligent systems not only improve usability but also enable advanced functionalities, including personalized assistance, empathetic dialogue, and context-aware decision-making.

Figure 1 presents the conceptual framework for intelligent agents with emotional intelligence. Building upon prior research, three fundamental components are identified as the core elements underpinning the development of emotional intelligence, described as follows:

1. **Emotion Understanding:** This stage involves analyzing the affective features embedded in user input, enabling the agent to accurately detect and interpret the user's emotional state during interaction (Afzal et al., 2024; Zhao et al., 2025).

2. **Affective Cognition:** In this phase, the agent assesses emotional events through cognitive reasoning processes to ensure accurate and context-sensitive interpretation. Subsequently, it performs emotional elicitation modeling to generate an internal affective state, which modulates higher-order cognitive functions such as learning, inference, and decision-making (Liu et al., 2024; Raggioli et al., 2025). This internal affective regulation drives the agent's adaptive behavior, resulting in responses aligned with the user's emotional context.

3. **Emotional Expression Synthesis:** Finally, the agent externalizes its emotional states through multimodal communication channels, including text, speech, and visual expressions, where cross-modal synchronization is essential for fostering coherence, authenticity, and naturalness in emotional interactions (Chen et al., 2023b; Abilbekov et al., 2024).

Emotion understanding in intelligent agents encounters substantial challenges in accurately detecting subtle or ambiguous affective signals, especially within diverse cultural contexts or in the presence of environmental noise (Rahmani et al., 2023) (Kamran et al., 2023). These challenges arise from practical limitations, including datasets with restricted size, noise, imbalance, and suboptimal quality (Ye et al., 2024; Oh & Kim, 2024). Further obstacles are presented by learning model deficiencies, such as low accuracy, limited interpretability, poor generalizability, and the lack of standardized evaluation metrics (Umair et al., 2024) (Cambria et al., 2024). Multimodal integration introduces additional complexity, particularly in the combination of heterogeneous data types and the management of missing modalities (Chandraumakantham et al., 2024; Geetha et al., 2024; Alsaadawi et al., 2024; Khan et al., 2024). The inherent complexity of emotions, including their diversity, overlap, and the challenges associated with experimental design, further complicates emotion understanding (Ye et al., 2024; Sharma et al., 2024). Moreover, contemporary large-scale models are affected by issues such as hallucinations, high annotation costs, and limited contextual comprehension (Farquhar et al., 2024; Schuller et al., 2026a). Recent advancements, such as data augmentation, synthetic data generation, transfer, and semi-supervised learning, model interpretability methods, enhanced evaluation standards, and multimodal fusion, present promising avenues for addressing these challenges. Affective cognition, a core component of emotional intelligence in intelligent agents, extends beyond emotion recognition to encompass reasoning about emotions and the generation of contextually appropriate responses through the integration of cognitive and affective theories of mind. Modeling affective cognition is inherently challenging due to the complexity of capturing contextual and causal factors, often necessitating sophisticated cognitive frameworks enriched with domain-specific knowledge.

This process involves identifying events and mental states that elicit emotions (emotion elicitation) and interpreting the resulting behavioral and cognitive outcomes (emotional experiences), as described in cognitive appraisal theories and data-driven approaches (Zall & Kangavari, 2024; Liu et al., 2024; Jokinen & Oulasvirta, 2025). However, the development of affective cognition models faces several significant challenges, including limited and ambiguous datasets, difficulties in computing cognitive appraisal variables, and scalability constraints in computational cognitive models (Gandhi et al., 2024; Somarathna et al., 2022; Bayro & Jeong, 2025). Additional challenges arise in large language model (LLM)-based systems, such as contextual misinterpretation and limitations in emotional reasoning capabilities (Tak & Gratch, 2024; Raggioli et al., 2025). To address these issues, recent studies have proposed solutions including advanced cognitive architectures,

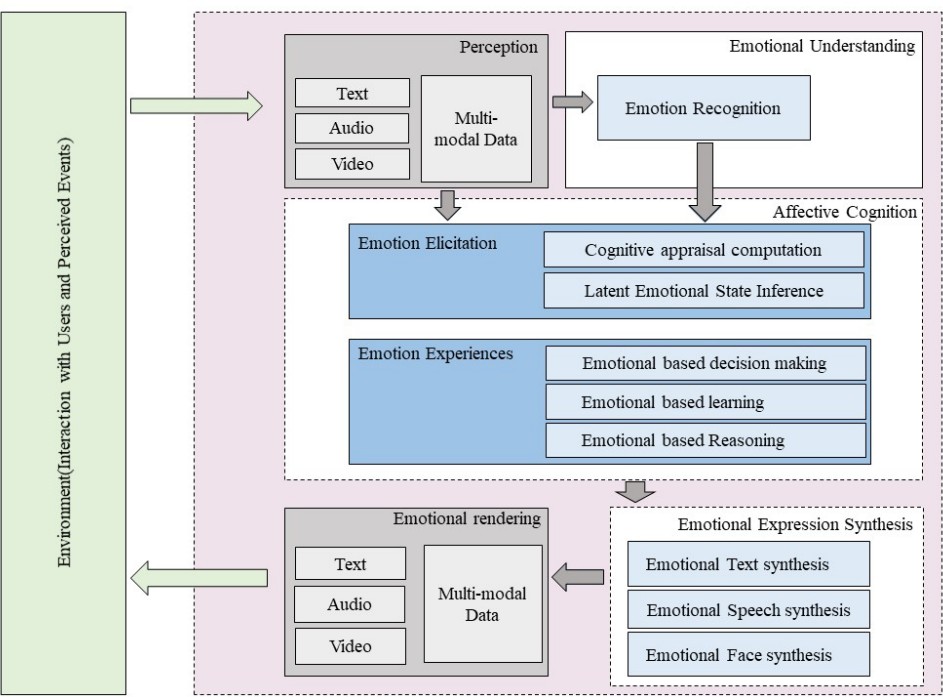

Figure 1: Overview of intelligent agent with emotional intelligence

virtual reality–based data collection, reinforcement learning integration, and explainable emotion-alignment frameworks. This study investigates these challenges and solutions to advance the development of emotionally intelligent agents capable of naturalistic and socially appropriate interactions.

Emotional text, speech, and face synthesis have become essential components of affective computing. They allow agents to generate responses that are contextually appropriate and emotionally resonant. In the realm of text synthesis, advancements like style transfer, conditional generation, and fine-tuned LLMs enhance expressiveness. However, challenges such as data sufficiency, consistency, and multimodal alignment still exist. Frameworks like MOPO and EmoBench aim to address these issues (Firdaus et al., 2023; Resendiz & Klinger, 2024; Sabour et al., 2024). Emotional Speech Synthesis (ESS) and voice conversion increasingly utilize non-parallel and diffusion-based methods, such as EmoConv-Diff, to improve scalability and emotional fidelity (Zhou et al., 2020a; Prabhu et al., 2024). Simultaneously, emotional face synthesis utilizes GANs, diffusion models, and multimodal systems such as UniPortrait and EmotiveTalk to address challenges related to data imbalance and synchronization. These advancements enable the creation of lifelike and coherent multimodal affective expressions.

## 1.1 Research questions

To guide our investigation into the integration of emotional intelligence in intelligent agents, we pose the following research questions:

- *RQ1*: How can intelligent agents robustly detect and interpret affective signals across diverse environments despite data and model limitations?

- *RQ2*: What are effective approaches for modeling emotion elicitation and experience in affective cognition, given challenges in ambiguity, computation, and scalability?

- *RQ3*: How can intelligent agents generate contextually appropriate and emotionally coherent responses across multiple modalities?

## 1.2 Contribution

This study investigates the fundamental components of emotional intelligence in intelligent agents, with an emphasis on their functional roles, the specific challenges that impede effective implementation, and the solutions proposed in existing literature. Through a systematic examination of these dimensions, the study aims to elucidate how emotional intelligence can be effectively integrated into artificial agents to enable more natural, adaptive, and human-like interactions. By synthesizing current research findings, this work highlights both the progress achieved and the critical gaps that remain unaddressed in the pursuit of genuine emotional intelligence in intelligent agents. Despite the substantial potential of emotionally intelligent agents, persistent challenges across core capabilities continue to constrain their performance and applicability in real-world environments.

Table 1 provides a comprehensive comparison between this survey and recent studies in the field of affective

Table 1: Comparison of the present survey with recent studies in affective computing (2024–2025) based on covered emotional aspects and modalities

|  | Emotion | | | Modality | | |
|---|---|---|---|---|---|---|
|  | Recognition | Elicitation | Expression | Text | Speech | Vision/Facial |
| Recent Trends of Multimodal Affective Computing: A Survey from an NLP Perspective (Hu et al., 2024) | ✓ | – | – | ✓ | ✓ | ✓ |
| A Review of Human Emotion Synthesis Based on Generative Technology (Ma et al., 2025a) | ✓ | – | ✓ | ✓ | ✓ | – |
| Affective Computing in the Era of Large Language Models: A Survey from the NLP Perspective (Zhang et al., 2024e) | ✓ | – | ✓ | ✓ | – | – |
| Artificial Emotion: A Survey of Theories and Debates on Realising Emotion in Artificial Intelligence (Li et al., 2025b) | – | ✓ | – | – | – | – |
| Emotion recognition and generation: a comprehensive review of face, speech, and text modalities (Mobbs et al., 2025) | ✓ | – | ✓ | ✓ | ✓ | ✓ |
| Intelligent Agents with Emotional Intelligence: Current Trends, Challenges, and Future Prospects | ✓ | ✓ | ✓ | ✓ | ✓ | ✓ |

computing published from 2024 to 2025. The comparison is organized along two principal dimensions: (1) emotional aspects, encompassing emotion recognition, elicitation, and expression, and (2) modalities, including text, speech, and visual or facial cues. This structure underscores the distinctive contribution of the present study, which offers an integrative perspective by addressing all relevant emotional dimensions across multiple modalities—thereby delivering a more holistic and unified understanding compared to previous research. The primary contributions of this study can be summarized as follows:

- To the best of our knowledge, this work represents the first comprehensive review of intelligent agents equipped with the full spectrum of emotional intelligence capabilities—namely, emotion understanding, affective cognition, and emotional expression, providing a cohesive framework for advancing naturalistic and empathetic human-computer interaction (HCI).

- It offers an in-depth analysis and categorization of the key challenges that hinder the effective realization of these three core capabilities within intelligent agents.

- It systematically evaluates recent methodological advancements proposed to address these challenges and delineates promising future research directions for the development of emotionally intelligent systems.

## 1.3 Paper organization

The remainder of this paper is organized as follows. Section 2 describes the systematic literature review methodology. Section 3 examines challenges and emerging solutions in emotion understanding. Section 4

explores challenges and emerging solutions in affective cognition, focusing on emotion elicitation in interactive systems and emotional experiences. Sections 5 to 7 address challenges and emerging solutions in emotional expression synthesis across multiple modalities: Section 5 focuses on Emotional Text Synthesis (ETS), Section 6 on ESS, and Section 7 on emotional face synthesis. Section 8 offers a comprehensive discussion of challenges and future research directions. Finally, Section 9 concludes with a summary of contributions and a vision for developing emotionally intelligent agents.

## 2 Methodology: Systematic Literature Review Process

This paper adheres to the Preferred Reporting Items for Systematic Reviews and Meta-Analyses (PRISMA) guidelines to ensure transparency and reproducibility in the study selection process. We adopted the PRISMA framework to systematically identify, screen, and include relevant literature on emotional intelligence in intelligent agents. Our focus is on three core capabilities: emotion understanding (emotion recognition), affective cognition (emotional elicitation and experiences), and emotional expression synthesis.

### 2.1 Identification

We conducted a comprehensive literature search across six major electronic databases, including Google Scholar, IEEE Xplore, ACM Digital Library, Scopus, Web of Science, and arXiv. The search strategy was designed to maximize coverage of affective computing research related to intelligent systems by incorporating multiple synonyms and closely related concepts.

The search query combined four groups of keywords corresponding to the research scope:

- **Affective computing concepts:** "affective computing", "emotion recognition", "emotion detection", "emotion classification", "emotion understanding", "emotion analysis", "emotion synthesis", "emotion generation", "emotional AI", and "affective intelligence".

- **Modalities:** "text", "speech", "audio", "facial expression", "gesture", "physiological signals", "multimodal", and "human-computer interaction".

- **Intelligent systems:** "intelligent agent", "intelligent system", "conversational agent", "social robot", "virtual agent", and "emotion-aware system".

- **Learning methods:** "deep learning", "transformer", "large language model", "reinforcement learning", "computational models of emotion", and "appraisal theory".

The final Boolean search query was adapted to the syntax of each digital library while preserving the same semantic structure. The systematic review primarily considered studies published between 2017 and 2025, reflecting the rapid development of deep learning methods in affective computing. Earlier publications (1990–2016) were not included in the quantitative synthesis but were selectively cited as foundational references to provide historical context and introduce established emotion theories.
To improve coverage, backward citation analysis was performed by examining the reference lists of highly relevant survey papers and seminal studies that were repeatedly retrieved across databases and widely cited in the field. Forward citation analysis was conducted using the "Cited by" functionality of Google Scholar to identify more recent studies extending these foundational works. Grey literature, including arXiv preprints and conference proceedings, was included only when the work directly addressed the review scope and had either been peer-reviewed subsequently or showed clear evidence of influence, such as repeated citation by later studies and inclusion in major bibliographic databases.

### 2.2 Screening

The titles and abstracts of all 2,700 records were screened for relevance. Studies were excluded at this stage if they were clearly outside the scope of affective computing (e.g., psychological studies on human emotion, medical studies on emotional disorders) or focused solely on human subjects without application to intelligent agents. This screening process reduced the number of records to 630.

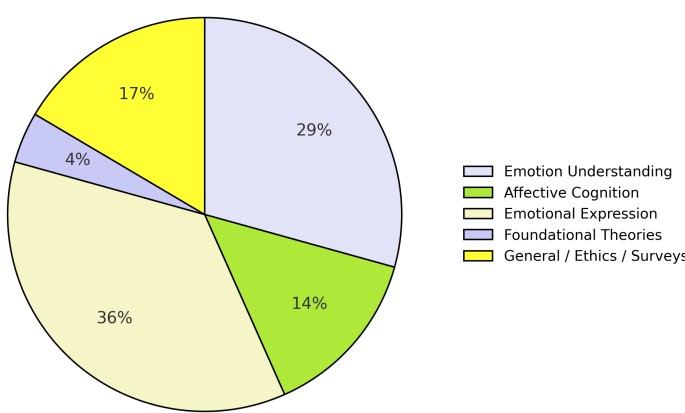

Figure 2: Distribution of the 406 included studies across the three core capabilities of emotional intelligence in intelligent agents.

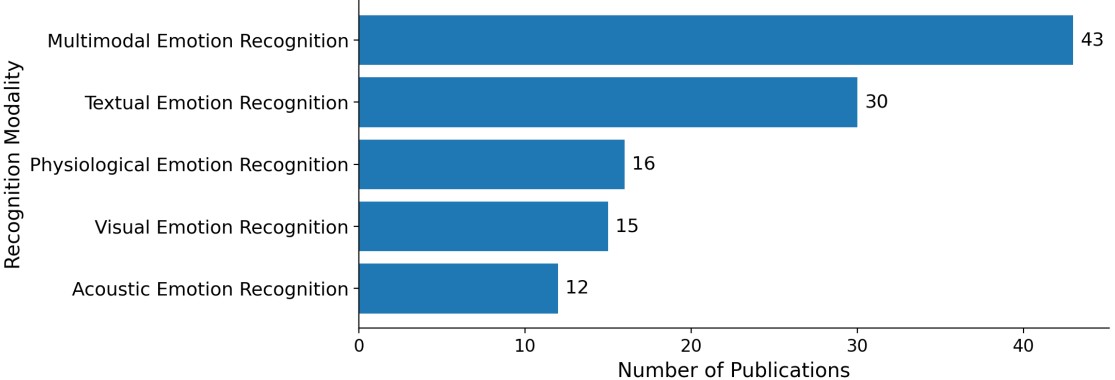

Figure 3: Breakdown of emotion understanding research by modality

## 2.3 Eligibility

The full text of the 630 remaining reports was retrieved and assessed for eligibility based on the following criteria:

- Inclusion criteria:
  The review incorporated studies published in English, including peer-reviewed journal articles, conference proceedings, books, and preprints from reputable repositories. To be included, a publication was required to explicitly address emotional intelligence in intelligent agents and demonstrate relevance to at least one of the three core capabilities examined in this study: (i) emotion understanding, such as emotion recognition from textual, acoustic, visual, or physiological data; (ii) affective cognition, including computational models of emotion elicitation, internal emotional state representation, or cognitive–affective architectures; and (iii) emotional expression synthesis, encompassing the generation of affective responses through textual, acoustic, or visual modalities. Both empirical studies presenting quantitative evaluations and theoretical or review articles offering insights into AI architectures, models, datasets, or applications were considered. Duplicate records identified across multiple databases were systematically removed to ensure a unique set of studies for analysis.

- Exclusion criteria:
  Non-English publications were excluded to maintain consistency in analysis and interpretation. Studies that did not explicitly focus on intelligent agents—such as works limited to general human psychology without a clear artificial intelligence context—were omitted. Additionally, research addressing unrelated topics, including non-affective AI tasks (e.g., conventional pathfinding or image classification without emotional components), was excluded. Low-quality sources, such as non-peer-reviewed blogs, opinion pieces, and outdated technical reports lacking empirical validation or academic citations, were also filtered out. Finally, all duplicate publications across the searched databases were removed to maintain a non-redundant corpus for the systematic review.

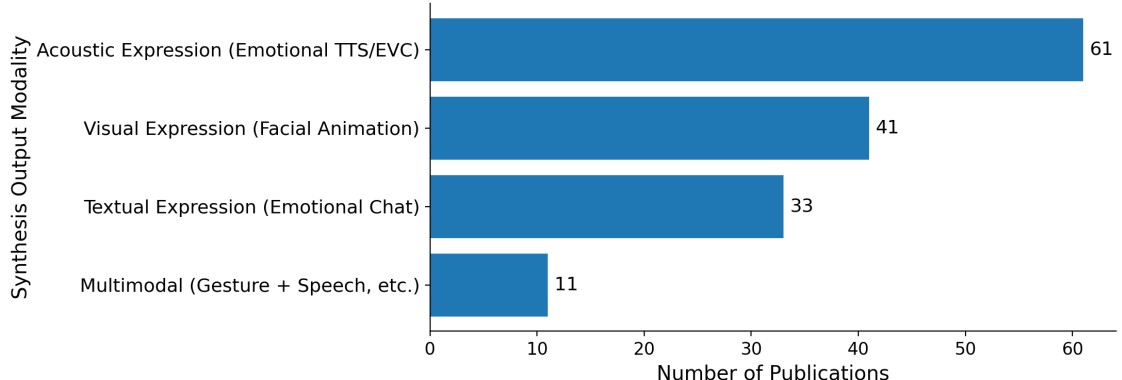

Figure 4: Breakdown of emotional expression synthesis research by output modality.

After this rigorous assessment, 224 reports were excluded, leaving 406 studies that formed the core basis of our survey.

## 2.4 Inclusion

The final 406 unique studies were subjected to an in-depth analysis to synthesize key findings, identify emerging trends, compare methodologies, and explore challenges and future directions. To provide a comprehensive overview of the research landscape, each study was assigned to exactly one primary category according to its main research objective, thereby avoiding double counting across capabilities. As illustrated in Fig. 2, Emotional Expression constitutes the largest category, accounting for 146 studies (36.0%), followed by Emotion Understanding with 119 studies (29.3%). General, Ethics, and Survey papers comprise 67 studies (16.5%), while Affective Cognition accounts for 57 studies (14.0%). Finally, Foundational Theories, including classical psychological models of emotion, represent 17 studies (4.2%). This categorization reflects the primary focus of each publication rather than its potential relevance to multiple research themes. Figures 3, 5, and 4 further analyze the literature within the three core capabilities of emotional intelligence. Figure 3 presents the distribution of emotion understanding studies across recognition modalities, highlighting the predominance of multimodal and text-based approaches. Figure 4 categorizes emotional expression research according to output modality, with emotional speech synthesis representing the largest body of work, followed by visual and textual emotional generation. Figure 5 summarizes the principal research themes within affective cognition, including cognitive architectures, LLM-based affective reasoning, reinforcement learning, appraisal and psychological theories, and social and ethical aspects. Unlike the capability-level categorization presented in Fig. 2, this subcategory analysis examines the thematic composition of affective cognition and therefore incorporates related foundational studies and cognitive architectures that provide essential theoretical and architectural support for this research area. Together, these analyses provide the foundation for the detailed state-of-the-art review presented in the subsequent sections.

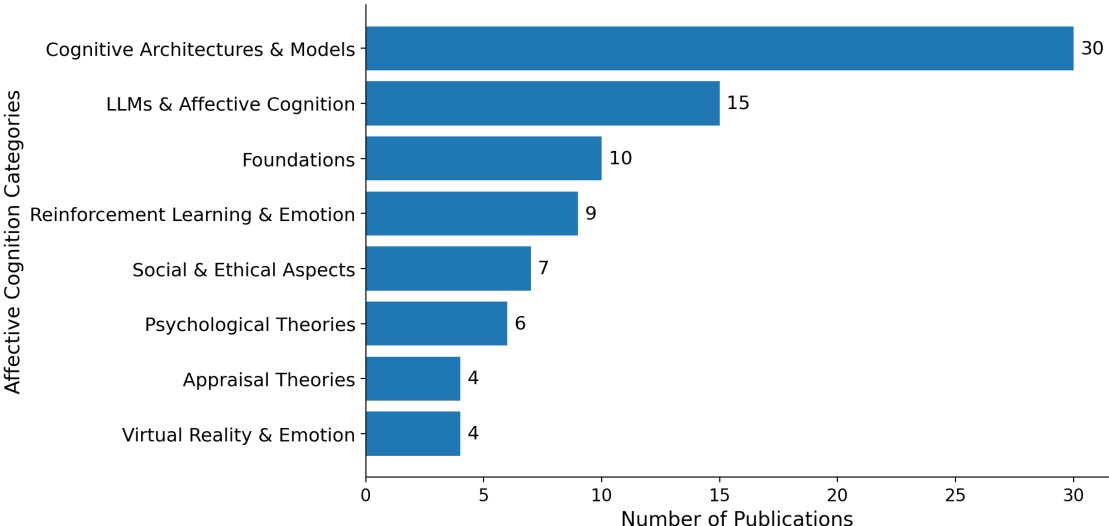

Figure 5: Categorization of affective cognition research by theoretical and computational focus.

## 3 Emotion Understanding

Emotion recognition is a fundamental task in artificial intelligence (AI) that aims to understand and interpret human emotions expressed through various forms of data (Afzal et al., 2024). This process identifies specific emotional states using facial expressions (Canal et al., 2022), vocal tones (Wani et al., 2021), language (Deng & Ren, 2021), and physiological signals (Dadebayev et al., 2022). Emotion recognition is crucial in enhancing HCI, mental health monitoring, and other applications where understanding human emotional responses is essential (Nayak et al., 2021). The inputs for emotion recognition are diverse and multimodal, making the task challenging and an intriguing opportunity. Text-based inputs, such as social media posts, reviews, or dialogues, are processed using Natural Language Processing (NLP) techniques to identify emotional nuances in the language. Machine learning models analyze words, phrases, sentence structure, and context to predict the underlying emotional tone of a text. In contrast, vision-based inputs leverage computer vision techniques to analyze facial expressions, body movements, and gestures, often using deep learning models for recognition. These visual cues offer valuable data to help AI systems accurately interpret emotions. This is especially important in real-time applications such as video conferencing or emotion-aware robotics. In addition to text and vision, sound or speech signals are critical for emotion recognition. Audio inputs, such as tone of voice, pitch, volume, and rhythm, are key indicators of emotional states. Finally, physiological signals, including heart rate, skin conductance, and facial muscle activity, provide additional insights into emotional responses, obtained through wearable devices or biosensors. These physiological markers reflect the autonomic nervous system's reactions to emotional stimuli. They are instrumental in applications like mental health monitoring and affective computing, where a deeper understanding of emotional states is necessary for more personalized interactions.

### 3.1 Approaches

Figure 6 presents a conceptual overview of common processing pipelines employed in emotion recognition systems. The framework is intended to summarize representative design choices rather than depict a mandatory sequence of processing stages, as different approaches may adopt different architectures depending on the input modality, learning paradigm, and application requirements.

The process begins with data acquisition, where emotion-related information may be obtained from one or more modalities, including textual data, speech, facial images or videos, physiological signals (e.g., electroencephalography (EEG) and heart rate), or other behavioral cues. Different modalities provide complementary information for affective analysis. For example, textual data capture semantic and linguistic

expressions of emotion, speech conveys prosodic and acoustic characteristics, facial images reveal visual expressions, and physiological signals reflect internal affective responses that may not be directly observable. Before model training, raw inputs are typically preprocessed to improve data quality and consistency. The specific preprocessing operations depend on the modality and may include noise reduction for speech signals, normalization and tokenization for text, image resizing or alignment for visual data, and artifact removal or filtering for physiological signals. Although preprocessing is commonly employed, some end-to-end learning frameworks perform minimal preprocessing and rely on the learning model to handle input variability.

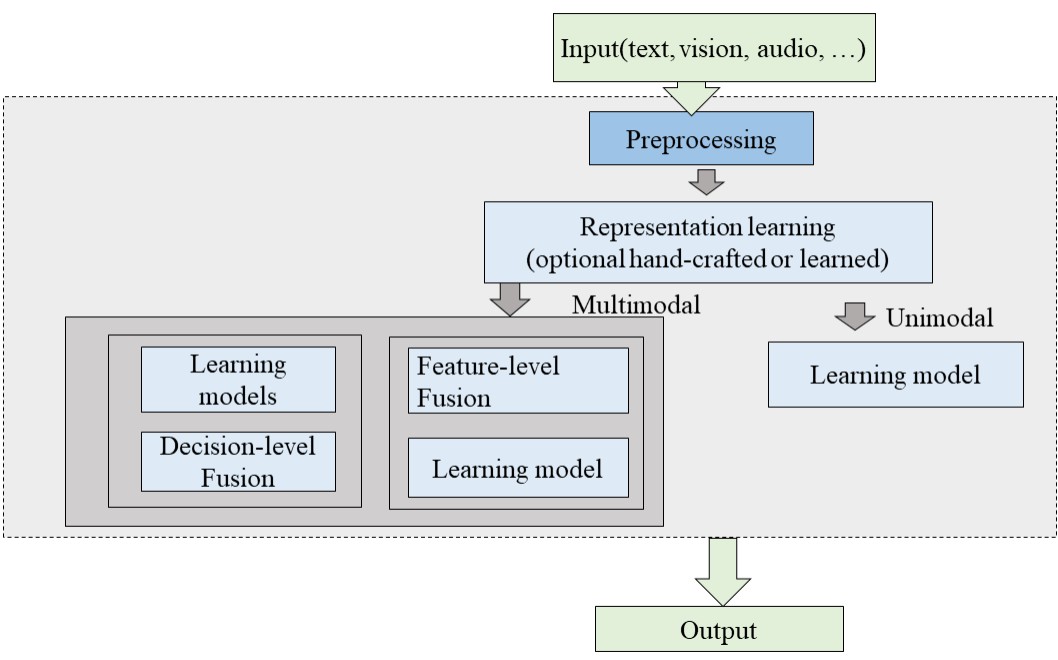

Figure 6: The overall framework of emotion recognition.

After preprocessing, informative representations are derived from the input data. Traditional emotion recognition systems rely on hand-crafted feature extraction, such as lexical and syntactic features for text, acoustic descriptors including pitch and spectral characteristics for speech, facial landmarks or appearance-based descriptors for images, and statistical or frequency-domain features for physiological signals. In contrast, modern deep learning and foundation-model-based approaches often replace manual feature engineering with representation learning, where latent feature embeddings are learned automatically from raw or minimally processed inputs through neural networks or pretrained foundation models. Consequently, the intermediate representations used for emotion recognition may consist of either manually designed features or learned latent embeddings, depending on the underlying architecture.

Following representation learning, emotion recognition may proceed through either unimodal or multimodal processing. In unimodal systems, a single modality is analyzed using a dedicated learning model to predict the emotional state. In multimodal systems, complementary information from multiple modalities is combined to improve robustness and recognition performance. Depending on the model architecture, multimodal integration may be performed using different fusion strategies rather than following a fixed processing sequence. The most common strategies include:

- **Feature-level (early) fusion:** Representations extracted from multiple modalities are combined into a unified feature space before being processed by a learning model.

- **Intermediate fusion:** Modalities interact within intermediate layers of a neural architecture, allowing cross-modal representations to be learned jointly during model training.

- **Decision-level (late) fusion:** Each modality is processed independently, and the individual predictions are subsequently combined to produce the final emotion classification.

Finally, learning models map the learned representations to emotion labels or continuous affective dimensions. Depending on the application, these models may include conventional machine learning algorithms, deep neural networks, transformer-based architectures, self-supervised models, or large foundation models. The choice of representation learning strategy, learning architecture, and fusion mechanism depends on the characteristics of the available data and the requirements of the target emotion recognition task.

## 3.2 Challenges

In this section, we aim to address the existing challenges in emotion recognition and explore the solutions that have been proposed for each challenge. We will review relevant studies that address these issues and their suggested remedies. These challenges arise from the complex nature of human emotions and the limitations of current technologies used to process and interpret them. As depicted in Figure 7, we classify these challenges into three major domains: data, learning models, and problem nature. Table 2 summarizes these categories, outlining their respective sub-issues and highlighting studies that address these challenges. Next, we will discuss the challenges and the methods proposed to overcome them.

### 3.2.1 Data-Related Challenges

Challenges related to data availability and quality, including limited dataset sizes, noisy or low-quality samples, and class imbalance, significantly constrain the accuracy, robustness, and generalizability of emotion recognition systems. Addressing these challenges is critical for the development of reliable affective computing models.

***Data collection:*** Collecting high-quality affective datasets remains one of the fundamental challenges in emotion recognition, as emotional responses are difficult to capture in a natural, diverse, and ethically acceptable manner (Mohammad, 2022). Unlike conventional classification tasks, emotion recognition requires data that accurately reflects genuine human affect across different modalities, cultures, demographic groups, and real-world environments (Ometov et al., 2025). Existing datasets are often collected under controlled laboratory settings, which may fail to represent spontaneous emotional expressions encountered in practical intelligent systems (Ometov et al., 2025). In addition, privacy concerns, the high cost of multimodal data acquisition, and the need for synchronized recordings further complicate large-scale dataset construction (Mohammad, 2022; Ometov et al., 2025). Existing approaches for addressing data collection challenges can be broadly categorized into three directions: naturalistic data acquisition, crowdsourcing, and synthetic data generation.

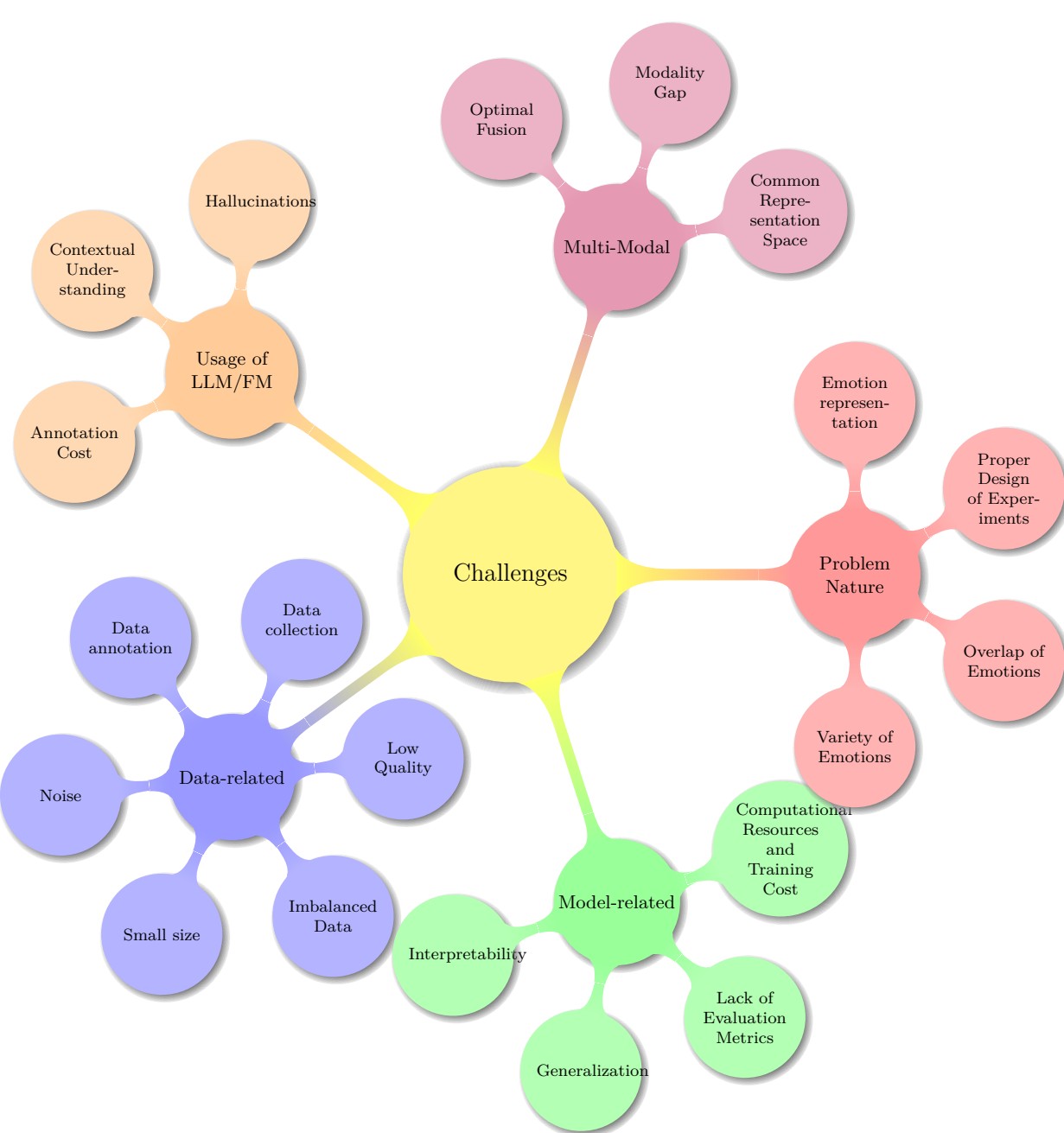

Figure 7: Challenges in emotion understanding

Table 2: Summary of challenges and solutions in emotional understanding

| Challenges | Sub-challenge | Solutions |
|---|---|---|
| **Data-related** | Data collection | Controlled-lab bias / low ecological validity and Naturalistic data acquisition (Ometov et al., 2025); Crowdsourcing with quality control (Mohammad, 2022); Synthetic data generation and augmentation (Schuller et al., 2026b; Sahu et al., 2025) ; Privacy-aware collection and synchronized recording protocols (Mohammad, 2022) |
| | Data annotation | Subjectivity, cost, and ambiguity of emotion labels; Consensus-based multi-rater annotation (Martínez-Miwa & Castelán, 2023; Shoer & Erzin, 2025); soft-label and multi-emotion modeling (Chou & Lee, 2025; Niu et al., 2025); active and weakly supervised labeling (Niu et al., 2025); continuous emotion representations ( (Khorram et al., 2019; Shoer & Erzin, 2025) |
| | Small size of data | Semi-supervised graph contrastive learning (Ye et al., 2024); GAN-based synthetic data generation (Schuller et al., 2026a); BERT-based augmentation techniques (Koufakou et al., 2023) |
| | Noise in data | Advanced data augmentation methods (Chowdary et al., 2023); Uncertainty-aware multimodal fusion (Tellamekala et al., 2023); Noise-robust CNN architectures (Oh & Kim, 2024) |
| | Data imbalance | SMOTE and Tomek Links (Ghafourian et al., 2022); GAN-based augmentation (Meng et al., 2024); Cost-sensitive learning (Li & Deng, 2016) |
| | Low data quality | Multi-source transfer learning (Sarkar et al., 2023); Probabilistic uncertainty modeling (Lo et al., 2023); Differential entropy features (Uyanık et al., 2022) |
| **Model-related** | Low accuracy | Multi-modal integration (Umair et al., 2024); Ensemble learning methods (Younis et al., 2022); Unsupervised representation learning (Ross et al., 2023) |
| | Computational efficiency | Sparse cross-modal attention (Dai et al., 2021); Hierarchical processing (Wei et al., 2022); Multiplicative fusion (Mittal et al., 2020) |
| | Interpretability | Attention mechanisms (Cortiñas-Lorenzo & Lacey, 2023); Grad-CAM visualization (Sharma et al., 2024); Neurosymbolic AI frameworks (Cambria et al., 2024); Coarse-to-fine training (Lian et al., 2024) |
| | Standardized metrics | Zero-shot evaluation (Schuller et al., 2026a); Quantitative quality metrics (de León Languré & Zareei, 2024) |
| | Generalization | Subject-independent models (Younis et al., 2022); Hybrid transformer approaches (Zanwar et al., 2022); Emotion-specific pretraining (Aroyehun et al., 2023) |
| **Problem Nature** | Variety of emotions | Graph contrastive learning (Ye et al., 2024); Multi-modal gating mechanisms (Sharma et al., 2024); Sensory knowledge integration (Zhao et al., 2025) |

Table 2: Summary of Challenges and Solutions in Emotional Understanding (Continued)

| Challenges | Sub-challenge | Solutions |
|---|---|---|
| | Overlap of emotions | Uniform label annotation (Du et al., 2025); Causal intervention (Yang et al., 2023); Graph attention networks (Li et al., 2023b) |
| | Experiment design | Real-world multimodal studies (Younis et al., 2022); Cross-subject evaluation (Hu et al., 2021); Continuous emotion assessment (Bota et al., 2019) |
| **Multi-modal** | Common representation | Rule-based conversion (Chandraumakantham et al., 2024); Attention mechanisms (Geetha et al., 2024); Semantic alignment networks (Ezzameli & Mahersia, 2023) |
| | Modality gap | Trimodal integration (Alsaadawi et al., 2024); Cross-attention mechanisms (Rajan et al., 2022); Contactless data fusion (Khan et al., 2024) |
| | Optimal fusion | Novel feature extractors (Middya et al., 2022); Shifted Window Transformers (Kim & Hong, 2024); Speaker-aware networks (Guo et al., 2024) |
| **LLM/FM** | Annotation cost | Synthetic data generation (Schuller et al., 2026a); LLM-based labeling (Li, 2024); Coarse-to-fine training (Lian et al., 2024) |
| | Contextual understanding | Enhanced prompting (Amin et al., 2024); Multimodal datasets (Zhang et al., 2025e); Unified feature conversion (Chandraumakantham et al., 2024) |
| | Hallucinations | Semantic entropy detection (Farquhar et al., 2024); Fact-checking methods (Sahoo et al., 2024); Reinforcement learning (Li et al., 2024a) |

*Naturalistic data acquisition* aims to capture spontaneous emotional responses in real-world environments rather than relying solely on laboratory-induced emotions. Recent studies increasingly employ wearable sensors, mobile devices, and in-the-wild recordings to improve ecological validity and enhance the diversity of emotional expressions observed during model training (Sahu et al., 2025; Ometov et al., 2025). However, these settings often introduce environmental noise, weaker control over recording conditions, and greater variability in annotation quality.

*Crowdsourcing-based data collection* enables researchers to gather large-scale emotional data from geographically and culturally diverse participants while reducing the cost of conventional laboratory experiments (Mohammad, 2022). Online annotation platforms and remote participation frameworks facilitate scalable data acquisition; however, maintaining data quality, annotator reliability, and consistency across raters remains a significant challenge (Mohammad, 2022).

*Synthetic data generation* complements real-world data collection by producing additional emotional samples using generative AI techniques (Schuller et al., 2026b; Sahu et al., 2025). Recent advances in GANs, diffusion models, and large generative models enable the synthesis of realistic facial expressions, emotional speech, and textual emotional content, helping alleviate data scarcity while improving demographic diversity and class coverage (Schuller et al., 2026b). Nevertheless, synthetic samples may not fully capture the complexity, authenticity, and contextual richness of real emotional behavior.

Constructing large-scale, representative, privacy-preserving, and multimodal affective datasets remains an open challenge for developing robust emotion recognition systems (Ometov et al., 2025).

***Data annotation challenges:*** Emotion annotation is inherently subjective because emotional perception varies across individuals, cultures, contexts, and even annotation protocols (Mohammad, 2022; Niu et al., 2025; Martínez-Miwa & Castelán, 2023). Unlike objective recognition tasks, the same emotional expression may receive different labels from different annotators due to ambiguous expressions, blended emotions, cultural differences, and differences in annotator background (Mohammad, 2022; Shoer & Erzin, 2025; Niu et al., 2025). Such inconsistencies reduce label reliability and directly affect the training, validation, and generalization of emotion recognition models (Mohammad, 2022; Martínez-Miwa & Castelán, 2023). Existing approaches for addressing annotation challenges can be broadly categorized into three directions: consensus-based annotation, active and weakly supervised annotation, and continuous emotion representation.

*Consensus-based annotation* improves label reliability by collecting annotations from multiple raters and aggregating them through majority voting, probabilistic fusion, or annotator reliability estimation (Martínez-Miwa & Castelán, 2023; Shoer & Erzin, 2025). Measuring inter-annotator agreement further helps identify ambiguous samples and improve dataset quality (Martínez-Miwa & Castelán, 2023; Niu et al., 2025). Recent work further proposes to retain all emotional ratings as soft-label distributions and to allow multi-emotion predictions, thereby capturing annotator subjectivity and ambiguity of emotions more faithfully (Chou & Lee, 2025).

*Active and weakly supervised annotation* seeks to reduce the cost of manual labeling by selectively annotating the most informative samples while exploiting large quantities of unlabeled data (Niu et al., 2025). Active learning, semi-supervised learning, and weak supervision substantially decrease annotation effort without significantly compromising model performance (Niu et al., 2025).

*Continuous emotion representation* addresses annotation ambiguity by replacing discrete categorical labels with dimensional affective representations, such as the Valence-Arousal-Dominance (VAD) model (Khorram et al., 2019). Continuous annotations better capture subtle emotional variations and mixed affective states, thereby improving label consistency for complex emotional expressions (Khorram et al., 2019; Shoer & Erzin, 2025).

Overall, research has evolved from conventional manual labeling toward more reliable and cost-effective annotation paradigms that combine multiple annotators, intelligent sample selection, and continuous affective representations. Despite these advances, obtaining accurate, consistent, and culturally unbiased emotion annotations remains one of the major challenges in affective computing.

***Small dataset size:*** Limited dataset size remains one of the most significant challenges in emotion recognition, as it restricts a model's ability to generalize across diverse emotional expressions, subjects, and application scenarios. Existing datasets are often small, domain-specific, and demographically biased, making it difficult to develop robust and transferable affective computing systems. To address this challenge, existing approaches can be broadly categorized into three directions: semi-supervised learning, generative data augmentation, and text augmentation techniques.

*Semi-supervised learning* aims to reduce the dependence on costly labeled data by leveraging abundant unlabeled samples. For example, (Ye et al., 2024) proposes a semi-supervised dual-stream self-attentive adversarial graph contrastive learning framework for EEG-based emotion recognition. By jointly exploiting a limited set of labeled samples and a larger pool of unlabeled data, the framework captures both structural and non-structural information, substantially improving recognition performance under low-resource conditions.

*Generative data augmentation* focuses on increasing dataset size and diversity through synthetic sample generation. (Schuller et al., 2026a) highlights recent advances in generative techniques (Bond-Taylor et al., 2021), particularly adversarial learning and generative adversarial networks (GANs) (Hajarolasvadi et al., 2020), which can synthesize realistic affective data to alleviate the scarcity of annotated emotional samples. These approaches improve model robustness by exposing learning algorithms to a broader range of emotional variations.

*Text augmentation techniques* have become increasingly popular for textual emotion recognition. (Koufakou et al., 2023) investigates contextual augmentation using BERT-based models (Devlin et al., 2019), where context-aware word replacement generates semantically consistent training samples. The study also explores synthetic text generation using pretrained language models to further enrich the training corpus. In contrast, traditional methods such as Easy Data Augmentation (EDA) (Wei & Zou, 2019), including synonym replacement, word insertion, deletion, and word shuffling, remain effective baselines for low-resource settings. These augmentation strategies improve dataset diversity and class coverage, leading to better performance of deep learning models such as RoBERTa (Liu et al., 2019).

Overall, research has evolved from conventional augmentation methods toward more data-efficient learning paradigms and generative approaches. While traditional techniques such as EDA remain useful because of their simplicity and low computational cost, recent studies increasingly rely on semi-supervised learning and generative AI to exploit unlabeled data and synthesize high-quality emotional samples. Despite these advances, ensuring the diversity, realism, and distribution consistency of generated emotional data remains an important open challenge for building robust emotion recognition systems.

**Noise in data:** Noise and irrelevant information can significantly degrade the performance of emotion recognition systems, particularly in real-world environments where audio, visual, or physiological signals are often affected by environmental interference, poor acquisition conditions, or sensor limitations. Common sources of noise include background sounds in speech recordings, illumination changes and occlusions in facial images, and artifacts in physiological signals. Existing approaches for handling noisy data can be broadly classified into three categories: data augmentation, uncertainty-aware multimodal fusion, and noise-robust deep learning.

*Data augmentation* improves robustness by exposing models to diverse corrupted samples during training. For example, (Chowdary et al., 2023) employs augmentation strategies such as translation, normalization, cropping, scaling, and artificial noise injection to increase the diversity of facial images. Training with these transformed samples enables the model to generalize more effectively to real-world noisy conditions and improves facial emotion recognition accuracy.

*Uncertainty-aware multimodal fusion* addresses noise by dynamically adapting the contribution of each modality according to its estimated reliability. (Tellamekala et al., 2023) proposes the COLD fusion framework, which estimates modality-wise aleatoric uncertainty and adjusts fusion weights accordingly. During evaluation, facial occlusions are simulated through face masking, allowing the framework to increase reliance on the audio modality when visual information becomes unreliable. This adaptive strategy improves robustness against modality-specific corruption.

*Noise-robust deep learning models* focus on learning discriminative representations despite image degradation. (Oh & Kim, 2024) combines data augmentation techniques, including image rotation, brightness adjustment, and cropping, with transfer learning and autoencoder-based denoising. In addition, attention-enhanced deep architectures such as CNNs (O'Shea & Nash, 2015), ResNet (He et al., 2016), and VGG (Simonyan & Zisserman, 2014), together with attention mechanisms (Vaswani et al., 2017), enable the model to focus on informative facial regions while suppressing irrelevant distortions, resulting in more reliable emotion classification under challenging conditions.

Recent advances have shifted the focus from conventional noise mitigation techniques toward adaptive frameworks that explicitly model data reliability. In particular, uncertainty-aware multimodal fusion and attention-enhanced deep learning have demonstrated greater robustness by dynamically exploiting the most informative modalities and features under noisy conditions. Despite these improvements, developing emotion recognition systems that can reliably cope with complex real-world noise and multiple degraded modalities remains a significant research challenge.

**Data imbalance:** Data imbalance is a common challenge in emotion recognition, where frequently occurring emotions, such as happiness, dominate the training data, while less common emotions, such as surprise or

disgust, are underrepresented. This imbalance biases learning algorithms toward majority classes, leading to poor recognition performance for minority emotions and reducing the fairness and robustness of emotion recognition systems. Existing solutions can be broadly categorized into three directions: data-level balancing, algorithm-level balancing, and representation learning.

*Data-level balancing* aims to equalize the class distribution by generating or selecting training samples. For example, (Ghafourian et al., 2022) combines SMOTE (Synthetic Minority Over-sampling Technique) (Chawla et al., 2002) with Tomek Links to simultaneously increase minority-class samples and remove ambiguous majority-class samples. This strategy enables deep learning models such as VGG-16 and ResNet-50 to learn more balanced decision boundaries. Similarly, GAN-based data augmentation (Hajarolasvadi et al., 2020) generates realistic samples for underrepresented emotional categories, improving class distribution and reducing model bias (Meng et al., 2024). Beyond conventional over-sampling, (Iranmehr et al., 2019) proposes Virtual Facial Sample Generation (VFSG), which synthesizes facial images with variations in illumination and pose, producing more diverse and realistic minority-class samples.

*Algorithm-level balancing* addresses class imbalance by modifying the learning objective rather than the data itself. (Li & Deng, 2016) reviews cost-sensitive learning approaches, such as SVM-Weight (Iranmehr et al., 2019), which assign larger misclassification penalties to minority classes to encourage balanced learning. Recent studies have also explored class-sensitive loss functions that explicitly increase the contribution of underrepresented emotions during optimization, improving recognition performance without altering the original dataset.

*Representation learning and multimodal feature integration* seeks to improve minority-class recognition by learning more informative feature representations. For instance, (Meng et al., 2024) integrates complementary semantic information from multiple modalities using a deep variational autoencoder, allowing the model to exploit correlations among text, speech, and visual cues to improve emotion discrimination, particularly for less frequent emotion categories.

Research on class imbalance has progressed from conventional sampling techniques toward more sophisticated learning strategies. While sampling-based methods such as SMOTE remain widely adopted because of their simplicity and effectiveness, recent studies increasingly combine generative models, cost-sensitive optimization, and multimodal representation learning to achieve more balanced and robust emotion recognition. Nevertheless, accurately recognizing extremely rare emotions and maintaining balanced performance across diverse datasets remain open challenges.

**Low data quality:** Low-quality data, including blurry facial images, low-resolution recordings, and noisy physiological signals, can substantially degrade the performance of emotion recognition systems by obscuring discriminative emotional cues. To mitigate this issue, existing approaches mainly focus on three directions: transfer learning, uncertainty-aware modeling, and robust feature extraction.

*Transfer learning* seeks to compensate for poor-quality target data by exploiting knowledge learned from high-quality source datasets. For example, (Sarkar et al., 2023) combines multi-source transfer learning (MSTL) (Lee et al., 2019) with multivariate correlation analysis (MCA) (Abdi, 2003) to identify informative features extracted from multiple pretrained models. By transferring only highly correlated and reliable knowledge while suppressing irrelevant information, the proposed framework improves facial emotion recognition in low-data and low-quality scenarios without requiring explicit image enhancement.

*Uncertainty-aware modeling* explicitly accounts for the ambiguity introduced by low-quality observations. (Lo et al., 2023) models probabilistic data uncertainty caused by low-resolution facial images and employs the Emotion Wheel theory to represent emotions in a continuous affective space. This representation enables the model to better distinguish visually similar expressions while providing more reliable uncertainty estimates.

*Robust feature extraction and preprocessing* aim to improve the quality of learned representations before classification. In EEG-based emotion recognition, (Uyanık et al., 2022) applies band-pass filtering to remove signal artifacts and uses Differential Entropy (DE) (Duan et al., 2013) to extract features that are less sensitive to noise. These features are then classified using machine learning models such as Support Vector

Machines (SVM) (Hearst et al., 1998) and Neural Networks, resulting in more reliable emotion recognition under noisy recording conditions.

The focus of recent research has moved from relying solely on preprocessing techniques toward learning-based approaches that enhance representation quality and explicitly model data uncertainty. In particular, transfer learning and uncertainty-aware techniques have emerged as effective solutions for mitigating the impact of limited high-quality data. However, effectively handling severe quality degradation across different sensing modalities while preserving discriminative emotional information remains an open research challenge.

### *Comparative Synthesis of Data-Related Strategies*

Across data-related challenges, no single strategy consistently outperforms others under all conditions. The choice of method depends on the dominant constraint in a given scenario, which may be ecological validity, annotation cost, data scarcity, class imbalance, or noise.

For *data collection*, naturalistic acquisition provides the highest ecological validity and is therefore most appropriate when modeling spontaneous, real-world emotional behavior is essential. Crowdsourcing is preferable when the primary goal is to scale up data volume and demographic diversity at moderate cost, but it is less suitable when strict control over annotator quality and label consistency is required. Synthetic data generation is particularly effective for alleviating data scarcity and class imbalance, but is best used to complement rather than replace real-world data because of potential realism and bias gaps.

For *data annotation*, consensus-based labeling (e.g., multi-rater aggregation, probabilistic fusion) generally provides the highest label reliability and is recommended when annotation quality is the main bottleneck. In contrast, active and weakly supervised annotation schemes are more suitable when annotation budget is limited and large pools of unlabeled data are available, as they substantially reduce labeling effort at the cost of potentially lower per-sample reliability. Continuous emotion representations (e.g., valence-arousal, soft-label distributions) are preferable when capturing subtle variations and mixed emotions is critical, whereas categorical labels may suffice for coarse-grained applications.

For *data-level modeling and augmentation*, semi-supervised learning is most beneficial when labeled data are scarce but abundant unlabeled data are available, whereas uncertainty-aware fusion and robust architectures demonstrate superior performance under noisy or low-quality multimodal settings. Data-level balancing methods, such as SMOTE and GAN-based augmentation, are particularly suitable for addressing class imbalance, whereas cost-sensitive learning is preferable when preserving the original data distribution is important.

Overall, the most effective data-related strategy depends on the specific characteristics of the dataset, the target application, and available computational and annotation resources. Rather than relying on a single universal approach, practitioners should select and combine data collection, annotation, and modeling techniques according to the dominant limitation in their setting.

### 3.2.2 Model-Related Challenges

Learning models are crucial for emotion recognition, as they map the given input, such as facial expressions, vocal cues, or physiological measurements, into corresponding emotional states. However, developing robust models for this task involves several challenges.

***Low accuracy:*** Low accuracy in emotion recognition models often stems from overfitting, where models that excel on training data struggle to generalize to unseen samples. This issue is particularly pronounced when models are trained on small or imbalanced datasets that cannot fully represent the diversity and complexity of human emotional expressions. Moreover, unimodal approaches that rely on a single source of information, such as facial expressions or text, may fail to capture the complementary aspects of emotions. (Umair et al., 2024) integrates facial expressions, speech, and textual information to overcome the limitations of unimodal systems. Similarly, ensemble-based approaches, as investigated in (Younis et al., 2022), have demonstrated improved accuracy by combining multiple predictive models. Beyond supervised

approaches, (Ross et al., 2023) introduces an unsupervised multimodal representation learning framework based on stacked convolutional autoencoders to extract latent representations from wearable physiological signals, including electrocardiogram (ECG) and electrodermal activity (EDA). These studies indicate a growing shift toward leveraging complementary information sources and more robust learning paradigms to enhance the generalization capability of emotion recognition models.

***Interpretability:*** Understanding the decision-making process of complex deep learning models remains a major challenge in emotion recognition, particularly in sensitive applications such as healthcare and psychological analysis, where transparency and reliability are essential. Deep learning architectures, including CNNs (O'Shea & Nash, 2015) and Recurrent Neural Networks (RNNs) (Schmidt, 2019), are often considered black-box models because their internal reasoning mechanisms are difficult to interpret. Although these models achieve strong predictive performance, explaining the factors that lead to a specific emotional prediction remains challenging. Existing explainability approaches can be broadly categorized into three groups: pre-model, in-model, and post-model methods (Cortiñas-Lorenzo & Lacey, 2023). Pre-model approaches improve interpretability by designing meaningful features and reducing data complexity before model training. In-model approaches aim to incorporate interpretability directly into the architecture, where mechanisms such as attention (Vaswani et al., 2017) enable the model to highlight important input components during prediction. Post-model approaches analyze model behavior after training through techniques such as Grad-CAM (Selvaraju et al., 2017) and sensitivity analysis.

Recent studies have explored these approaches to improve transparency in affective computing systems. For example, (Sharma et al., 2024) applies Grad-CAM to visualize the regions of meme images that contribute to emotional predictions, providing insights into how visual elements, such as facial expressions and contextual information, influence the model's decisions. Beyond conventional post-hoc explanations, (Cambria et al., 2024) proposes a neurosymbolic framework that combines commonsense knowledge representation with hierarchical attention networks to provide more interpretable affective analysis. By integrating symbolic reasoning with deep learning-based pattern recognition, the framework produces more traceable and trustworthy predictions. Similarly, (Lian et al., 2024) addresses interpretability in multimodal emotion recognition through a two-stage training strategy that combines large-scale coarse annotations with fine-grained human-verified labels. By learning associations among audio, visual, and textual information, the framework improves both recognition performance and the ability to connect predictions with supporting multimodal evidence.

Research on explainable emotion recognition is moving beyond purely post-hoc visualization techniques toward more integrated and transparent learning frameworks. While attention mechanisms, visualization methods, and neurosymbolic approaches have improved model interpretability, developing explainable systems capable of providing reliable reasoning across complex multimodal scenarios remains an open challenge.

***Lack of standardized evaluation metrics:*** The absence of standardized evaluation metrics remains a critical challenge in emotion recognition, as it limits the comparability of results across studies and makes it difficult to assess the actual effectiveness of proposed models. Differences in datasets, evaluation protocols, and reporting practices can lead to inconsistent performance analyses and prevent reliable benchmarking of affective computing systems. (Schuller et al., 2026a) investigates the challenges caused by the lack of standardized evaluation criteria, particularly for emerging foundation models in affective computing. The study emphasizes the necessity of developing rigorous evaluation methodologies, including comparative benchmarks based on established emotion recognition datasets and advanced assessment strategies such as zero-shot classification, to enable more meaningful analysis of model capabilities.

Similarly, (de León Languré & Zareei, 2024) highlights the importance of standardized evaluation in text emotion detection by introducing a comprehensive framework for assessing dataset quality. The proposed framework evaluates datasets using 14 quantitative metrics across four dimensions: representativity, readability, structure, and part-of-speech (POS) distribution. By systematically measuring factors such as class balance, linguistic complexity, lexical diversity, and syntactic characteristics, the framework provides consistent and reproducible criteria for dataset assessment. Experimental evaluations using BiLSTM (Huang et al., 2015) and BERT (Devlin et al., 2019) demonstrate that dataset quality significantly influences model performance, reinforcing the importance of standardized evaluation practices for reliable comparison of

TED approaches.

These studies indicate a growing need for unified evaluation protocols that consider both model capabilities and dataset characteristics. Establishing standardized benchmarks and reproducible assessment frameworks will be essential for enabling fair comparisons and accelerating progress in emotion recognition research.

***Generalization:*** A critical challenge in emotion recognition systems is their limited generalization capability across different domains, including variations in cultural background, environmental conditions, subjects, and input modalities. This domain shift problem often causes significant performance degradation when models trained on one dataset are applied to unseen scenarios. To improve cross-domain robustness, recent studies have explored strategies that enhance model adaptability and reduce dependence on dataset-specific characteristics.

(Younis et al., 2022) highlights the importance of subject-independent predictive models that avoid relying on individual-specific patterns and instead learn emotion-related characteristics that generalize across different populations. Beyond subject variation, recent approaches have focused on improving representation learning to better capture domain-invariant information. (Zanwar et al., 2022) combines transformer-based language models, including BERT (Devlin et al., 2019) and RoBERTa (Liu et al., 2019), with psycholinguistic features extracted through BiLSTM networks (Huang et al., 2015). This integration enables richer contextual representations and improves performance on unseen datasets. Similarly, (Aroyehun et al., 2023) introduces an emotion-specific Masked Language Model (eMLM) pre-training strategy, where emotion-related words are masked during training to encourage the model to learn broader contextual patterns rather than relying on explicit emotional cues. The resulting LEIA model demonstrates improved adaptability under domain shifts.

Another emerging direction is reducing dependence on manually annotated data by exploiting diverse datasets and unsupervised learning strategies. (Ross et al., 2023) proposes an unsupervised multimodal representation learning framework that combines multiple datasets to increase training diversity and improve generalization across emotional contexts. The reported results demonstrate competitive performance across several benchmarks, highlighting the potential of data diversity and self-supervised learning for addressing domain adaptation challenges.

***Computational resources, scalability, and training cost:*** The increasing complexity of affective computing models, particularly with the adoption of deep neural networks, large-scale multimodal architectures, and foundation models, introduces substantial computational and resource requirements. These challenges affect various affective computing tasks, including emotion recognition, affective reasoning, and emotional expression synthesis, regardless of whether the models rely on unimodal or multimodal inputs. High computational costs arise from large-scale data processing, complex model architectures, extensive pretraining and fine-tuning procedures, and the need for specialized hardware, which can limit scalability, accessibility, and real-world deployment. Several approaches have been proposed to improve computational efficiency.

*End-to-end architectures* reduce redundant processing stages by integrating feature extraction and prediction within a unified framework. For example, (Dai et al., 2021) introduces an end-to-end multimodal emotion recognition model with sparse cross-modal attention that selectively emphasizes informative features while reducing unnecessary computation. Although particularly beneficial for multimodal systems, such architectural optimization principles are applicable to broader affective computing scenarios.

*Efficient attention mechanisms and lightweight architectures* aim to reduce computational complexity while maintaining predictive performance. (Wei et al., 2022) proposes a hierarchical attention mechanism and a lightweight visual processing module that decrease computational and memory requirements. Similarly, *adaptive fusion strategies* can reduce redundant computation by dynamically selecting informative representations. For instance, (Mittal et al., 2020) introduces a multiplicative fusion mechanism that emphasizes reliable modalities and suppresses noisy information, thereby improving computational efficiency.

More recently, *parameter-efficient learning approaches* have emerged as an effective solution for adapting large pretrained models with limited computational resources. Prompt learning, adapters, and low-rank adaptation methods enable efficient fine-tuning by updating only a small fraction of model parameters while

reducing memory and training costs (Wu et al., 2025). These approaches are particularly important for the deployment of large affective models in resource-constrained environments.

Despite these advances, achieving an effective balance between computational efficiency, model scalability, and affective understanding capability remains a fundamental challenge. Future affective computing systems must consider not only recognition or generation accuracy but also computational sustainability, accessibility, and practical deployment constraints.

***Comparative synthesis of model-related strategies:*** The reviewed studies indicate that no single modeling strategy simultaneously optimizes all aspects of emotion recognition performance. Multimodal and ensemble learning approaches often achieve strong recognition accuracy by exploiting complementary emotional cues, making them well suited to applications where predictive performance is the primary objective. In contrast, explainable AI techniques, including attention mechanisms, Grad-CAM, and neurosymbolic frameworks, are particularly valuable in safety-critical domains such as healthcare, where transparency and user trust are essential despite potential trade-offs in model complexity and raw accuracy. Methods based on domain adaptation, self-supervised learning, and large-scale pre-training tend to provide better generalization across unseen datasets and heterogeneous populations, whereas standardized evaluation frameworks are indispensable for ensuring fair and reproducible comparisons rather than directly improving model performance. Overall, the most appropriate modeling strategy depends on the target application and involves balancing predictive accuracy, interpretability, generalization capability, and evaluation reliability rather than optimizing a single objective.

### 3.2.3 Problem Nature Challenges

The nature of the problem in emotion recognition poses significant challenges, as emotions are inherently subjective, overlapping, and context-dependent. Unlike fixed categorical tasks, emotions exist on a continuous spectrum, making it difficult to establish clear distinctions between them. Additionally, emotional expressions can vary widely between individuals and cultures, and environmental factors influence them, leading to inconsistencies in labeling. The presence of multiple modalities adds another layer of complexity, as different modalities can convey conflicting emotional cues. These factors make emotion recognition a complex task that requires adaptive models capable of handling ambiguity, variability, and contextual dependencies.

***Impact of emotion representation models:*** The challenges and corresponding solutions in affective computing are strongly influenced by the underlying emotion representation model. Existing approaches mainly follow two paradigms: categorical(Ekman, 1992) and dimensional emotion models(Russell, 1980). Categorical models represent emotions as discrete classes (e.g., happiness, sadness, anger), where major challenges include class imbalance, inter-class similarity, and ambiguity between closely related emotions(Wang et al., 2022). Consequently, solutions often focus on discriminative feature learning, robust classification architectures, and strategies for handling imbalanced data(Sethu et al., 2019).
In contrast, dimensional models represent emotions in continuous spaces, such as valence-arousal or valence-arousal-dominance, which introduce different challenges. These include subjective annotation variability(Ruiz et al., 2017), uncertainty in emotion intensity estimation(Wu et al., 2023), and difficulty in capturing subtle emotional changes. Therefore, approaches based on dimensional models typically require regression-based objectives(Wu et al., 2023), uncertainty modeling, and evaluation metrics such as correlation-based measures(Mohammad, 2018). Accordingly, challenges discussed in this survey should be interpreted considering the employed emotion representation paradigm, as the effectiveness of a solution may vary depending on whether emotions are modeled as discrete categories or continuous affective dimensions.

***Variety of emotions:*** The variety of human emotions poses a significant challenge for emotion recognition systems, which need to address the complex range of emotional states that individuals experience. While basic emotions such as happiness, sadness, anger, and fear serve as foundational categories, human emotional experiences extend beyond these simple labels. Furthermore, emotions can manifest differently in each individual due to factors like personality, context, and socio-cultural norms. These complexities make it difficult to distinguish between emotions. Consequently, emotion recognition systems

risk oversimplifying and misinterpreting the depth and context of human emotions.

Recent studies have explored more expressive learning strategies to better represent emotional diversity. (Ye et al., 2024) introduces a semi-supervised framework for EEG-based emotion recognition that employs graph contrastive learning (You et al., 2020) to improve the separation of different emotional states. By effectively utilizing both labeled and unlabeled data, the framework enhances the model's ability to capture diverse emotional patterns under limited annotation conditions. In multimodal scenarios, (Sharma et al., 2024) proposes a neural framework for meme emotion understanding that models enhanced visual emotional cues and employs a gating mechanism to integrate information from multiple modalities. This approach enables the model to better interpret complex emotional expressions conveyed through combined visual and contextual signals. Furthermore, (Zhao et al., 2025) integrates sensory knowledge into the T5 framework (Xue et al., 2021) by incorporating sensory information into the attention mechanism. This design improves contextual understanding and increases sensitivity to subtle emotional variations through joint optimization of contextual and sensory representations.

***Overlap of emotions:*** The coexistence and similarity between different emotional states create substantial challenges for emotion recognition systems. Many emotions share common physiological patterns, facial expressions, or contextual cues; for example, fear and surprise may exhibit similar observable characteristics. Moreover, emotional experiences are subjective, dynamic, and influenced by cultural factors, making it difficult to define clear boundaries between emotional categories. Existing approaches to address this challenge mainly focus on improving emotion annotations, reducing contextual biases, and enhancing multimodal representations.

*Improved annotation and dataset design* aim to provide more reliable supervision for learning complex emotional patterns. (Du et al., 2025) introduces a dataset with modality-independent labels designed to better represent mixed and overlapping emotions. The study employs robust annotation strategies and evaluates inter-annotator agreement (Braylan et al., 2022) to improve label reliability. By reducing inconsistencies in emotional annotations, the dataset provides a stronger foundation for training models capable of capturing complex affective states.

*Causal reasoning approaches* attempt to separate genuine emotional cues from misleading contextual information. (Yang et al., 2023) proposes the Contextual Causal Intervention Module (CCIM), which addresses context bias by removing spurious correlations between environmental factors and emotion labels. Through causal intervention and backdoor adjustment, the framework reduces the influence of irrelevant contextual features and encourages models to rely on intrinsic emotional expressions rather than dataset-specific biases.

*Multimodal representation and fusion* approaches exploit complementary information from different modalities to distinguish between similar emotions. GraphMFT (Li et al., 2023b) introduces a graph-based multimodal fusion framework that models visual-acoustic, visual-textual, and acoustic-textual relationships using heterogeneous graphs. By applying graph attention networks (GATs), the framework dynamically assigns importance to intra-modal and cross-modal interactions, enabling more precise discrimination between closely related emotions such as frustration and anger or excitement and happiness. Furthermore, incorporating speaker representations and adaptive modality interactions helps reduce confusion caused by overlapping emotional expressions.

Instead of assuming that emotions occur independently, recent approaches seek to model their inherent interdependence and contextual variability. Advances in annotation quality, causal analysis, and multimodal representation learning have improved the recognition of overlapping emotional states. However, developing models that consistently interpret complex and blended emotions across different individuals and environments remains an unresolved challenge.

***Proper design of experiments:*** The design of emotion recognition experiments plays a crucial role in determining the reliability, generalizability, and practical applicability of developed models. Since emotional responses are influenced by individual characteristics, cultural backgrounds, and environmental conditions, poorly designed experiments may introduce biases that limit model performance in real-world applications. Moreover, reproducing natural emotional responses in controlled laboratory environments remains inherently difficult. Experimental design in emotion recognition is increasingly moving toward

more realistic and unbiased evaluation settings. Recent studies place greater emphasis on ecological validity, subject-independent evaluation, and experimental protocols that minimize dataset-specific biases, leading to models with stronger generalization capabilities.

*Real-world experimental settings* aim to improve ecological validity by collecting emotions in natural environments rather than under highly controlled laboratory conditions. For example, (Younis et al., 2022) conducts a real-world study that captures participants' spontaneous emotional responses while integrating multimodal physiological and environmental data. The resulting subject-independent models are less dependent on individual-specific characteristics and demonstrate improved generalization across diverse users.

*Bias-resistant experimental protocols* focus on preventing models from learning irrelevant experimental artifacts. Traditional emotion recognition studies often expose all participants to the same emotional stimuli, allowing models to unintentionally exploit stimulus-specific information instead of genuine emotional patterns. To overcome this limitation, (Hu et al., 2021) proposes a novel data partitioning strategy in which training and testing samples are drawn from different video-induced datasets. This protocol encourages models to learn stimulus-independent emotional representations, thereby improving cross-subject generalization and reducing overfitting.

*Ecological validity and evaluation strategies* emphasize designing experiments that better reflect real-world emotional behavior. (Bota et al., 2019) discusses several limitations of conventional laboratory studies, including annotation difficulties, signal degradation caused by uncontrolled environmental factors, and variability arising from individual mood and cultural differences. This paper recommend conducting experiments in less constrained environments with more diverse participants and continuous emotion assessment to improve the validity and reliability of emotion recognition research.

***Comparative synthesis of problem nature strategies:*** The reviewed studies indicate that the effectiveness of problem-oriented solutions largely depends on the nature of the underlying challenge. Categorical emotion models are often more suitable for applications requiring discrete emotion classification and straightforward decision-making, whereas dimensional representations provide richer descriptions of emotional intensity and are preferable for modeling continuous affective states, despite their higher annotation complexity and subjectivity. For handling diverse and overlapping emotions, multimodal representation learning and causal reasoning approaches tend to outperform unimodal methods by exploiting complementary information and reducing contextual biases. Furthermore, experimental protocols emphasizing subject independence and ecological validity generally demonstrate stronger generalization to real-world scenarios than conventional laboratory-based evaluations. Overall, no single approach addresses all problem-related challenges, and selecting an appropriate strategy requires balancing representational richness, annotation complexity, contextual robustness, and experimental realism according to the target application.

### 3.2.4 Multi-Modal Challenges

Fusing multiple modalities is a crucial step in developing robust emotion understanding models. These models integrate diverse data sources, such as audio, visual, and textual cues, each offering distinct insights into emotional states. Effective fusion requires precise alignment of modalities to preserve their individual contributions, a process that can be computationally costly, especially with large-scale datasets. Furthermore, the inherent differences in how each modality encodes emotional information can complicate the optimization of their interactions. To ensure these sources complement rather than conflict with one another, fusion strategies must be carefully designed to balance performance and efficiency. Addressing these challenges demands advanced architectures and substantial computational resources, ultimately enhancing the system's ability to generalize across varied emotional expressions and contexts.

***Common representation space:*** Learning a common representation space is a fundamental challenge in multimodal emotion recognition because different modalities, such as speech, facial expressions, physiological signals, and text, encode emotional information in heterogeneous forms. Variations in feature characteristics, temporal dynamics, and semantic meaning make it difficult to align these modalities into a unified representation. Furthermore, differences in emotional expression across individuals and cultures increase the complexity of learning modality-invariant representations. Existing studies have addressed

this challenge through representation transformation, attention-based representation learning, multimodal fusion, and semantic alignment.

*Representation transformation* aims to map heterogeneous modalities into a common modality before emotion reasoning. For example, (Chandraumakantham et al., 2024) proposes a rule-based framework that converts non-verbal cues into textual descriptions. These textual representations are incorporated into prompts and processed by a pre-trained large language model (LLM), providing a unified textual representation while maintaining flexibility for integrating additional modalities.

*Attention-based representation learning* focuses on identifying emotionally relevant information across multiple modalities. (Geetha et al., 2024) discusses transformer architectures and attention mechanisms that selectively emphasize informative multimodal features while capturing their spatial and temporal relationships. Such architectures facilitate the learning of more coherent shared representations for complex emotional states.

*Fusion and semantic alignment* seek to explicitly associate heterogeneous feature spaces before multimodal reasoning. (Ezzameli & Mahersia, 2023) reviews feature-level, decision-level, and model-level fusion techniques that integrate complementary information from different modalities to improve recognition accuracy. Similarly, the Semantic Alignment Network (SAN) (Hou et al., 2023) introduces a Cross-Modal Alignment (CMA) module that projects heterogeneous features into a unified semantic embedding space. Combined with attention mechanisms that dynamically weight modality contributions, SAN effectively reduces semantic discrepancies and improves recognition of subtle emotional cues.

Recent research has increasingly focused on learning modality-invariant representations rather than simply combining heterogeneous features. Approaches based on semantic alignment, attention mechanisms, and large language models have significantly improved the ability to integrate complementary emotional information across modalities. Nevertheless, learning a robust shared representation that remains effective across diverse modalities, domains, and cultural contexts continues to be a key challenge in multimodal emotion recognition.

**Modality Gap:** The modality gap refers to the discrepancies between heterogeneous data sources, such as text, speech, facial expressions, and physiological signals, each of which captures emotional information from a different perspective. These discrepancies arise from differences in sensing mechanisms, feature representations, temporal characteristics, and data quality, making it difficult to align and fuse multimodal information effectively. Consequently, emotion recognition models may struggle to learn consistent cross-modal representations, reducing their overall robustness and recognition accuracy.

To address this challenge, recent studies have focused on developing more effective multimodal integration and alignment strategies. For example, (Alsaadawi et al., 2024) advocates a trimodal affective computing framework that jointly exploits textual, vocal, and visual information. By employing multimodal fusion techniques, the framework reduces inconsistencies among different modalities and improves emotion classification performance. Similarly, (Khan et al., 2024) investigates contactless emotion recognition and emphasizes the importance of integrating heterogeneous sensing modalities with different data characteristics and granularities. The proposed framework aligns complementary information across modalities to ensure that each modality contributes meaningfully to the final prediction, resulting in more robust emotion recognition. Furthermore, (Rajan et al., 2022) analyzes the effectiveness of attention mechanisms for multimodal integration by comparing self-attention and cross-attention. While self-attention models dependencies within each modality, cross-attention explicitly learns interactions across modalities, enabling better feature alignment and more effective reduction of modality discrepancies.

**Optimal fusion:** Optimal multimodal fusion aims to effectively combine complementary emotional information from different modalities to maximize recognition performance. Since the reliability of each modality varies across different situations and modalities may contain inconsistent or even conflicting information, determining how and when to fuse multimodal features remains a fundamental challenge. Existing studies have explored various fusion strategies to better capture cross-modal relationships while

preserving complementary emotional cues.

(Middya et al., 2022) proposes a model-level fusion framework that employs dedicated feature extractors for audio and video modalities before integrating them into a unified multimodal architecture. By systematically evaluating different combinations of feature extractors, the study identifies configurations that maximize the complementarity between audio and visual information, leading to improved emotion recognition accuracy. Similarly, (Kim & Hong, 2024) introduces a shifted window transformer encoder together with symmetric cross-attention mechanisms to model complex interactions among heterogeneous modalities. In addition to multimodal signals, the framework incorporates contextual metadata, such as environmental conditions and personal characteristics, allowing the fusion process to better adapt to individual differences. Furthermore, (Guo et al., 2024) proposes a speaker-aware cognitive network that combines text, audio, and visual features through a cross-modal attention mechanism while incorporating speaker-specific information. By modeling conversational dynamics with a GRU-based cognitive module, the framework captures richer emotional dependencies and produces more informative multimodal representations.

***Comparative synthesis of multi-modal strategies:*** Common representation learning and semantic alignment methods are particularly effective when heterogeneous modalities exhibit substantial representational discrepancies, whereas cross-modal attention mechanisms are more suitable for capturing complementary interactions among modalities in complex emotional scenarios. Dynamic and adaptive fusion strategies generally outperform static fusion approaches when the reliability of individual modalities varies because they can selectively emphasize informative modalities while suppressing noisy inputs. In contrast, lightweight architectures, end-to-end learning, and parameter-efficient adaptation are preferable for resource-constrained environments where computational efficiency is critical. Overall, no single multimodal strategy consistently dominates across all settings, and the optimal solution depends on the trade-off between representation quality, fusion flexibility, computational cost, and deployment requirements.

### 3.2.5 Usage of LLMs/FMs

The application of foundation models and LLMs to emotion recognition presents significant challenges due to the nuanced and diverse nature of emotional cues. Although these models excel at processing vast datasets and performing across multiple tasks, their generalization to diverse emotional expressions, particularly in complex social or cultural contexts, remains inconsistent. Unlike traditional emotion recognition models tailored to specific datasets or tasks, foundation models and LLMs are designed for broad applicability, which can compromise their precision in capturing subtle emotional nuances. Fine-tuning these models for emotion recognition is challenging, as it requires balancing general knowledge with context-specific adaptability to achieve high accuracy. To address this, techniques such as domain-specific pretraining, transfer learning with emotionally rich datasets, and the integration of multimodal data (e.g., text, audio, and visual cues) can enhance model performance. Additionally, incorporating cultural and social metadata into training pipelines may improve the models' ability to interpret diverse emotional expressions, thereby advancing their effectiveness in real-world applications.

***Annotation cost:*** The high cost of data annotation remains a major obstacle to applying foundation models (FMs) and LLMs in emotion recognition. Developing reliable affective computing systems requires large-scale labeled datasets that capture the diversity of emotional expressions across different contexts, cultures, languages, and modalities. Producing such datasets is both time-consuming and expensive because it relies heavily on expert annotation of subtle emotional cues in text, speech, images, and videos. Moreover, continuously updating these datasets to reflect evolving emotional expressions further increases the annotation burden.

Recent studies have explored alternative strategies to reduce manual annotation requirements. For example, (Schuller et al., 2026a) highlights the potential of large multimodal foundation models that are pretrained on diverse datasets to reduce dependence on task-specific labeled affective data. In addition to transferring general knowledge to emotion recognition tasks, these models can also generate synthetic emotional samples, further alleviating the need for extensive human annotation. Similarly, (Li, 2024) investigates the use of LLMs as a cost-effective alternative to traditional crowdsourcing. By combining LLM-generated annotations with human labels through appropriate aggregation strategies, the study demonstrates that high-quality

annotations can be obtained while substantially lowering annotation costs. Furthermore, (Lian et al., 2024) introduces the EMER-Coarse dataset and a two-stage training framework that first exploits large-scale coarsely labeled data and subsequently refines the model using a much smaller manually verified dataset. This coarse-to-fine learning strategy significantly reduces annotation effort while maintaining competitive emotion recognition performance.

Recent research has focused on minimizing manual annotation by leveraging pretrained foundation models, synthetic data generation, LLM-assisted labeling, and coarse-to-fine learning strategies. These approaches substantially reduce annotation costs while maintaining competitive performance. Finally, ensuring the quality, consistency, and fairness of automatically generated or weakly supervised emotion labels remains an important challenge for future emotion recognition systems.

***Contextual understanding:*** Accurate emotion recognition requires understanding the context in which emotions are expressed, including conversational history, speaker characteristics, social interactions, and cultural background. Although foundation models (FMs) and large language models (LLMs) possess strong language understanding capabilities, they often struggle to correctly interpret subtle emotional cues when contextual information is incomplete or ambiguous. As a result, recognizing complex emotions in real-world scenarios remains a significant challenge.

Recent studies have explored different strategies to enhance contextual reasoning in LLM-based emotion recognition. (Amin et al., 2024) investigates the performance of ChatGPT across several affective computing tasks and proposes a structured prompting framework that reformulates regression problems into pairwise ranking classification tasks. This prompting strategy improves the model's ability to distinguish subtle contextual differences in emotional expressions. Similarly, (Lu et al., 2024) demonstrates that GPT-4V exhibits limited performance in recognizing certain facial expressions and subjective emotions when sufficient contextual information is unavailable, highlighting the importance of explicit context modeling for multimodal emotion recognition. To better exploit conversational context, DialogueLLM (Zhang et al., 2025e) incorporates emotion knowledge and contextual information into a large language model that is fine-tuned on multimodal conversational datasets. By jointly modeling textual interactions and visual cues, the framework improves emotion recognition in dialogue scenarios. Furthermore, (Chandraumakantham et al., 2024) proposes a unified representation framework that converts audio and visual information into textual descriptions before processing them with an LLM. This representation enables the model to reason over multimodal contextual information within a common semantic space, leading to more consistent interpretation of complex emotional expressions.

***Hallucinations:*** Hallucinations remain a major challenge when applying large language models (LLMs) to emotion recognition, as models may generate incorrect emotional labels or infer emotional cues that are not supported by the input. Such errors typically arise from spurious associations learned during pre-training or from incomplete contextual reasoning, reducing the reliability of emotion recognition systems. This issue is particularly critical in high-stakes applications, such as mental health assessment, where inaccurate emotional interpretation may lead to inappropriate decisions.

Recent studies have investigated both the causes of hallucinations and effective strategies for detecting and mitigating them. (Sahoo et al., 2024) presents a comprehensive taxonomy that categorizes hallucinations into contextual disconnection, semantic distortion, and factual inaccuracies, together with mitigation techniques including self-checking, fact verification, prompt engineering, consistency checking, data augmentation, and fine-tuning. Complementing this work, (Farquhar et al., 2024) proposes a semantic entropy-based method for hallucination detection that estimates semantic uncertainty by clustering model responses according to their meaning rather than their lexical form. High semantic entropy indicates unreliable model outputs, providing an effective task-agnostic mechanism for identifying potential hallucinations. In addition, (Li et al., 2024a) systematically investigates the origins of factual hallucinations and introduces the HaluEval 2.0 benchmark for comprehensive evaluation across multiple domains. The study further evaluates several mitigation strategies, including retrieval-augmented generation, reinforcement learning from human feedback (RLHF), self-reflection, and improved prompting, demonstrating that RLHF is particularly effective in reducing hallucinations across diverse application domains.

Existing research demonstrates that addressing hallucinations requires a combination of complementary strategies rather than a single solution. Detection methods, response verification, retrieval-based enhance-

ment, and alignment techniques each contribute to improving output reliability. Despite these developments, ensuring faithful emotion recognition under complex and ambiguous conditions continues to present considerable challenges.

***Comparative synthesis of usage of LLMs:*** The reviewed studies demonstrate that different strategies address distinct limitations of LLM-based emotion recognition. Pretrained foundation models and coarse-to-fine learning frameworks are particularly effective for reducing annotation costs when large manually labeled affective datasets are unavailable. In contrast, prompt engineering, dialogue-aware architectures, and multimodal reasoning approaches achieve superior performance in scenarios where contextual understanding is essential for interpreting subtle emotional cues. To improve output reliability, hallucination mitigation techniques such as retrieval-augmented generation, response verification, semantic uncertainty estimation, and reinforcement learning from human feedback (RLHF) are generally more suitable, especially in high-stakes applications. Overall, no single strategy addresses all challenges associated with LLMs, and practical systems should balance annotation efficiency, contextual reasoning, and output reliability according to the target application and deployment requirements.

## 4 Affective Cognition

The Affective Theory of Mind extends beyond emotion recognition, requiring reasoning about emotions and responding appropriately. Achieving this requires not only emotional recognition but also internal modeling of affective strategies and context-appropriate expressive behavior (Raggioli et al., 2025). Although current affective computing primarily addresses emotion recognition and expression synthesis, advancing artificial emotional intelligence depends on modeling both emotional elicitation and experience (Zall & Kangavari, 2022). Thus, integrating cognitive and affective theories of mind is essential to enable affective cognition. This process involves two central components: first, identification of events and cognitive states that elicit emotions, and second, recognition of behaviors and cognitive states shaped by those emotions. This paper examines affective cognition from varied perspectives, highlighting its challenges and discussing approaches for progressing artificial emotional intelligence.

### 4.1 Approaches

Overall, the proposed methods for modeling artificial emotional elicitation are categorized into two main groups: theory-driven (Zall & Kangavari, 2024) and data-driven approaches (Liu et al., 2024). Theory-driven approaches often focus on cognitive appraisal theory, which is one of the most well-established psychological theories of emotional elicitation in humans. This theory outlines that emotions arise in response to both internal and external events. To elicit the appropriate emotion, it is necessary to evaluate the experienced event through various cognitive appraisals and personal experiences. These theories emphasize cognitive appraisal variables that assess an event from multiple perspectives. Some theory-driven approaches are based on a cognitive computational framework (Jokinen & Oulasvirta, 2025) that outlines essential cognitive information, such as concerns, goals, and needs, which are crucial for forming theory-based appraisals. The overall overview of these approaches is illustrated in Figure 8. As shown in this figure, the perceived stimuli are evaluated to calculate the cognitive appraisal variables. The mental states and experiences necessary for determining these appraisal variables are described within the cognitive computational model. Following this, the appraisal variables are mapped to specific emotional states. These resulting emotional states then influence emotional modulation, which in turn affects decision-making and emotional expression. Theory-based methods predict emotional states in future utterances by concentrating on cognitive processes. In contrast, data-driven methods analyze the conversation's history to identify specific events within the observed data and determine the emotions associated with those events for upcoming contributions.

### 4.2 Challenges

The modeling of emotional elicitation and experiences in intelligent agents is a crucial aspect of the affective theory of mind. A key component of this process involves developing a cognitive model that serves as a foundational framework for managing knowledge, information, experiences, and cognitive mental states

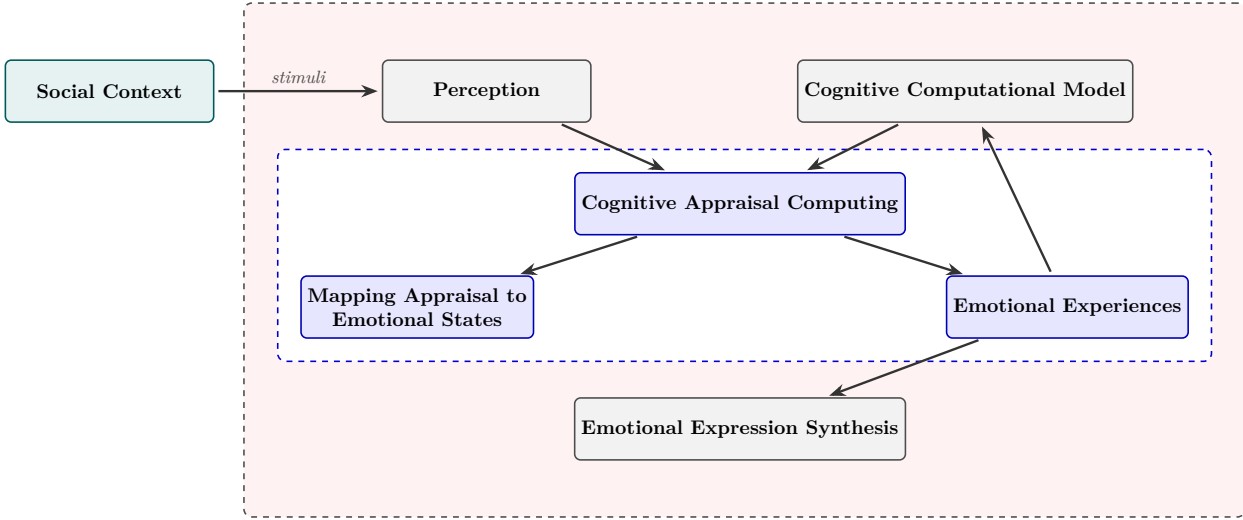

Figure 8: Block diagram of intelligent agent with affective cognition

related to emotions. One of the main challenges in this area is establishing reliable methods for computing cognitive appraisal variables based on underlying cognitive theories. Data-driven approaches also face significant obstacles, such as limited and ambiguous data, which impede the accurate capture and representation of the complex and multifaceted nature of human emotions. Additionally, technological and methodological barriers, including the need for robust models and the integration of LLMs, further complicate the task. Addressing these challenges is essential for advancing precise and effective models of emotional responses that are crucial for creating empathetic and socially intelligent systems. The primary challenges in this domain, along with their associated sub-challenges, are illustrated in Figure 9. Furthermore, Table 3 provides an overview of various studies and the solutions they propose for tackling these issues. In the following sections, we will explore these challenges in detail and review potential solutions to overcome them, to enhance the fidelity and applicability of emotion modeling in intelligent agents.

### 4.2.1 Data-Related Challenges

The field of emotion elicitation and experiences currently lacks a comprehensive and cohesive dataset. There is a clear need for data collections that explicitly clarify the relationship between various emotions and the appraisal variables and events that influence them. Integrating data-driven approaches with theoretical frameworks requires the availability of extensive and complete datasets. Although some efforts have been made in this area, a fully integrated and systematic dataset is still essential for advancing research. Developing such datasets would lead to a better understanding of emotional responses and their underlying mechanisms, ultimately contributing to the improvement of both theoretical models and practical applications in emotional analysis. Emerging evidence indicates that induction mechanisms are essential for eliciting emotions by simulating the sensory experiences needed for experimental paradigms. Over the years, many reviews have predominantly focused on passive elicitation methods, where individuals act as observers, often neglecting the importance of self-relevance in emotional experiences. To address this gap in the literature, (Somarathna et al., 2022) explores the potential of Virtual Reality (VR) as an active mechanism for emotion induction. Furthermore, to ensure the effectiveness and reliability of research outcomes, VR environments must incorporate well-selected stimuli to successfully evoke specific emotional responses. (Bayro & Jeong, 2025) presents experimental protocols for collecting datasets with virtual reality. (Gandhi et al., 2024) proposes an automatic dataset generation that includes diverse scenarios for benchmarking affective cognition, specifically focusing on understanding and reasoning about human emotions in foundation models. It presents a pipeline for generating diverse and naturalistic stimuli that can systematically and scalability evaluate affective reasoning. The generated scenarios explore relationships

between appraisals, emotions, expressions, and outcomes.

While these approaches all contribute to improving affective datasets, they address different aspects of the problem. Traditional and experimentally collected datasets provide high-quality and well-controlled emotional annotations but are often limited in scale and diversity. VR-based data collection offers greater ecological validity by enabling immersive and self-relevant emotional experiences, making it particularly suitable for studying complex emotional responses in realistic environments. In contrast, automatic dataset generation using foundation models provides a scalable solution for creating diverse affective scenarios and benchmarking reasoning capabilities, although the quality and authenticity of generated emotional content still require careful validation. Therefore, the choice of data collection strategy depends on the intended application, balancing annotation quality, ecological validity, scalability, and experimental control.

### 4.2.2 Model-related challenges

In this section, we review the challenges are related to learning model.

***Cognitive computational modeling*** Theoretical approaches to emotional elicitation modeling primarily depend on cognitive models to calculate appraisal variables. These models provide the foundational information required to determine emotional states. Most methods not only specify the emotion to be expressed but also examine how emotions influence decision-making and other cognitive processes, such as inference. A significant challenge in emotion elicitation modeling is the effective implementation of cognitive models. Several computational cognitive models are available for this purpose, including cognitive architectures (Anderson et al., 2004b; Laird, 2019b), Belief-Desire-Intention (BDI) frameworks (Pereira et al., 2005b; Sanchez et al., 2019b), and other specific designs (Hoorn et al., 2021; Becker-Asano & Wachsmuth, 2010). These computational models exhibit substantial differences, with each group demonstrating unique structural and functional characteristics.

*Cognitive architectures:* Cognitive architectures provide comprehensive computational frameworks for modeling intelligent behavior by integrating perception, memory, reasoning, learning, and action selection mechanisms. Architectures such as ACT-R (Anderson et al., 2004a), SOAR (Laird, 2019a), and LIDA (Franklin et al., 2007) include specialized memory systems, including procedural, semantic, episodic, and perceptual memory, together with control mechanisms and learning processes. These components enable agents to maintain knowledge, learn from experiences, and generate adaptive behaviors. For example, (Juvina et al., 2018) employs ACT-R to associate emotional values with memory elements, demonstrating how cognitive architectures can incorporate affective information into reasoning processes. Similarly, the E-VOX system (Perez et al., 2017) combines a SOAR-based cognitive architecture with the ALMA affective model (Flavian-Blanco et al., 2011), enabling real-time emotion processing, mood tracking, and personality representation. The Emotion and Adaptation Model (EMA) (Gratch & Marsella, 2004) further extends cognitive modeling by incorporating fast and slow appraisal mechanisms that evaluate events based on agent goals and beliefs, producing emotional states that influence behavior.

*Emotion-enhanced BDI frameworks:* BDI-based approaches focus on modeling decision-making through the interaction between beliefs, desires, and intentions. Although traditional BDI frameworks provide effective high-level reasoning mechanisms, they often lack detailed representations of emotional processes. To overcome this limitation, several studies have incorporated affective components into BDI architectures. Emotional-BDI (Pereira et al., 2005a) extends the traditional BDI model by integrating emotional states into agent resource and behavior management, providing a foundation for emotion-driven decision-making. Fuzzy-based affective BDI (Taverner et al., 2021) incorporates cultural and linguistic variability through fuzzy appraisal rules, mapping generated emotions into PAD dimensions. Similarly, ABC-EBDI (Sanchez et al., 2019a;c) connects beliefs, emotions, and behaviors based on ABC theory while considering personality and mood to produce more realistic agent responses.

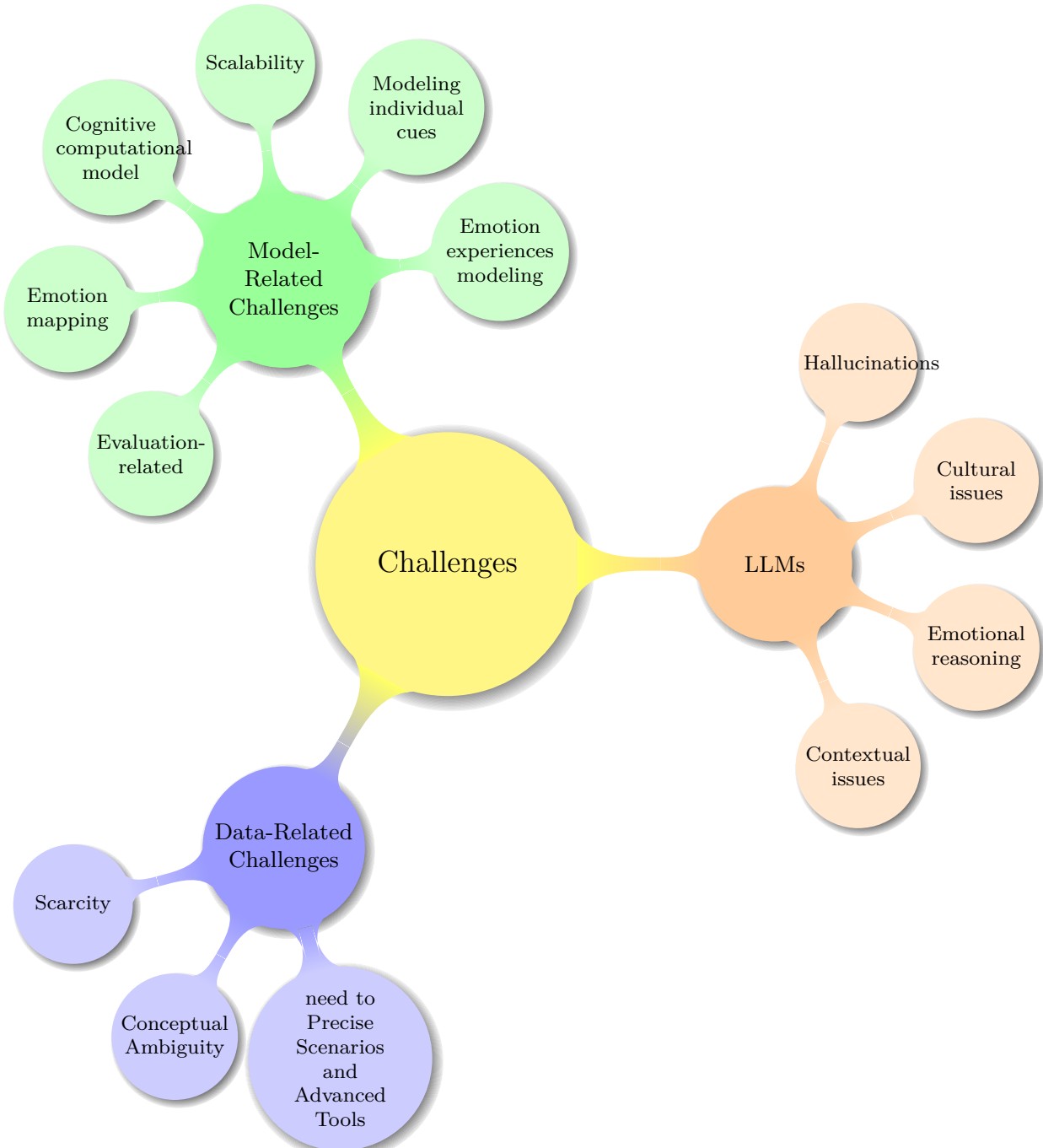

Figure 9: Challenges in affective cognition

Table 3: Summary of challenges and solutions in affective cognition

| Challenge | Sub-challenge | Solution |
| --- | --- | --- |
| **Data-Related** | Scarcity | Automatic dataset generation with diverse scenarios for benchmarking affective cognition (Gandhi et al., 2024) |
| | Conceptual Ambiguity | Experimental protocols for VR dataset collection (Bayro & Jeong, 2025) |
| | Precise Scenarios & Tools | VR as active mechanism for emotion induction (Somarathna et al., 2022) |
| **Model-Related** | Cognitive computational modeling | ACT-R architecture (emotional memory valuation) (Juvina et al., 2018); E-VOX SOAR + ALMA (real-time processing) (Perez et al., 2017); EMA Automatic/deliberate appraisal (Gratch & Marsella, 2004); Emotional-BDI Resource management (Pereira et al., 2005a); Fuzzy-based BDI Cultural/linguistic modeling (Taverner et al., 2021); ABC-EBDI BDI + ABC theory (Sanchez et al., 2019a); InFra Stimuli appraisal (Rodriguez et al., 2016); Silicon Coppelia Ethics/aesthetics variables (Hoorn et al., 2021); EIAEC Appraisal theories + memory (Zall & Kangavari, 2024) |
| | Scalability | Knowledge graphs + continuous learning (Zall & Kangavari, 2024) |
| | Emotion mapping | EIAEC Data-driven approach (Zall & Kangavari, 2024); EEGS Weighted mapping strategy (Ojha et al., 2020b); CPM + VR data-driven (Somarathna & Mohammadi, 2024); Regret-based RL (Soman et al., 2024) |
| | Emotion experiences modeling | SUSAN Inner speech modeling (Corvaia et al., 2025); Neural network framework (Hernández-Marcos & Ros, 2024); EIAEC Context-dependent values (Zall & Kangavari, 2024); Memory valuations (Juvina et al., 2018); Time-based annotations (Tsfasman et al., 2025); RL + appraisal theory (Zhang et al., 2024d); Value prediction (Zhang et al., 2024b); Weber-Fechner + Q-learning (Wu & Sun, 2025); Emotional motivation (Berto et al., 2025); Reward modulation (Nikodemou & Christodoulou, 2024); TD-based expressions (Nijeholt & Broekens, 2023) |
| | Evaluation | Scenario simulation & human comparison (Sanchez et al., 2019b; Zall & Kangavari, 2024); Multi-agent performance assessment (EBDI) (Jiang et al., 2007); Emotional reasoning in critical situations (Bourgais et al., 2016); Human feedback evaluation (E-VOX) (Perez et al., 2017) |
| **LLM-based** | Emotional reasoning | Distinguishes self-attribution vs perception of emotions using appraisal theory (Tak & Gratch, 2024); Combines cognitive/affective reasoning with intrinsic motivation (Raggioli et al., 2025); Explainable emotion alignment for Metaverse agents (Ma et al., 2025b) |
| | Contextual misunderstandings | Assesses affective cognition in LLMs (GPT-4, Claude-3, Gemini-1.5-Pro) (Gandhi et al., 2024) |

*Specialized cognitive-affective models:* Some approaches design specific cognitive components according to the requirements of particular applications. The Integrative Framework (InFra) (Rodriguez et al., 2016) models the interaction between perception, appraisal, emotion generation, and behavior while considering personality and cultural influences. Silicon Coppelia (Hoorn et al., 2021) uses appraisal variables related to ethics, aesthetics, epistemics, and affordances to generate emotionally appropriate responses. By applying fuzzy representations, it can handle simultaneous and ambiguous emotional states. More recently, the EIAEC framework (Zall & Kangavari, 2024) introduces an emotion-aware intelligent agent architecture that combines appraisal theories with episodic memory and data-driven emotion modeling. The framework learns affective values from previous experiences and uses emotional states to influence action selection.

Cognitive computational models have evolved from general-purpose architectures that simulate human cognitive processes toward more specialized frameworks that explicitly integrate emotional reasoning and adaptation mechanisms. While cognitive architectures provide comprehensive representations of memory and reasoning, BDI-based and specialized affective models offer more flexible approaches for modeling emotion-driven behaviors. Future research is expected to focus on developing unified architectures that combine cognitive reasoning, emotional understanding, memory, and adaptive decision-making to support more human-like intelligent agents.

Additionally, different cognitive computational models are suitable for different application scenarios. Cognitive architectures such as ACT-R and SOAR are preferable when comprehensive reasoning, memory, and learning mechanisms are required, making them appropriate for long-term autonomous agents. In contrast, emotion-enhanced BDI frameworks are more suitable for decision-oriented applications where beliefs, goals, and intentions drive agent behavior with relatively lower implementation complexity. Specialized cognitive-affective models provide greater flexibility and domain-specific performance by incorporating appraisal theories, personality, or cultural factors, although they are generally less transferable across applications. Therefore, selecting an appropriate computational framework depends on the desired balance between cognitive completeness, adaptability, and application specificity.

**Scalability** Most existing emotion models are explicitly designed for specific scenarios or contexts (Hernández-Marcos & Ros, 2024). For example, the MAMID model (Hudlicka, 2004) is exclusively tailored to treatment settings, relying on specialized if–then rules that render its action evaluation and selection highly domain-dependent. Similarly, WASABI (Becker-Asano, 2008) has been primarily validated in virtual character gaming environments, which limits its applicability to broader use cases. More generally, many computational emotion models concentrate on only one or two components of the emotional process, depending on the target application. Due to limited scalability and difficulties in seamless integration, extending these models with additional components remains challenging. In contrast, the EIAEC framework (Zall & Kangavari, 2024) supports scalability across diverse scenarios and applications by leveraging multiple memory types enriched with knowledge graphs. Rather than relying on predefined rules or fixed relationships, this framework enables the continuous learning and refinement of emotional relationships and experiences over time.

Overall, rule-based computational models are appropriate for well-defined and domain-specific applications where expert knowledge is readily available. However, their scalability is limited when new domains, emotions, or cognitive components must be incorporated. In contrast, memory-driven and knowledge-graph-based frameworks provide greater flexibility and continuous adaptation, making them more suitable for open-world intelligent agents operating in dynamic environments, albeit at the cost of increased model complexity.

**Emotion mapping modeling** An emotion model implanted in appraisal theory aims to map event assessments (appraisals) into corresponding emotion intensity levels, as specified by the theoretical framework. However, most appraisal theories do not provide explicit, quantifiable rules for mapping these appraisals to emotion intensities. Consequently, many computational implementations adopt heuristic or ad-hoc methods to approximate this relationship, primarily to facilitate research and experimental validation (Ojha et al., 2021). For instance, EEGS (Ojha et al., 2020b) introduces a mathematical formula for calculating emotion intensities based on appraisal theory, employing a weighted mapping strategy that utilizes quantitative appraisal variables to derive implications for various emotions. Similarly, (Somarathna

& Mohammadi, 2024) leverages the Component Process Model (CPM), which encompasses appraisal, expression, motivation, physiology, and feeling components, to explore their interrelations with emotion elicitation. This study adopts a data-driven approach using interactive virtual reality (VR) games and multi-modal data collection, including self-reports, physiological signals, and facial expressions, to establish a dataset that delineates the relationship between appraisal variables and emotional states. Furthermore, (Zall & Kangavari, 2024) presents an advanced mapping methodology grounded in datasets from (Mohammadi & Vuilleumier, 2022) and (Somarathna et al., 2023), employing a data-driven framework to refine the mapping between appraisal processes and emotional responses.

Path planning (PP) in autonomous systems can significantly benefit from reinforcement learning (RL) due to its flexible decision-making capabilities. (Soman et al., 2024) has introduced emotion-inspired mechanisms, particularly regret-based models, to enhance the performance of RL. These models incorporate two key concepts: experienced regret, which measures past suboptimal actions, and anticipated regret, which predicts future errors. By utilizing regret dynamics to adjust the epsilon decay in epsilon-greedy policies, agents can achieve a better balance between exploration and exploitation. This leads to improved learning efficiency.

Existing emotion mapping approaches differ in both interpretability and adaptability. Rule-based and mathematical mapping functions provide transparent and theoretically grounded emotion estimation but often rely on manually designed appraisal relationships. In contrast, data-driven mapping approaches can learn complex appraisal-emotion relationships from multimodal observations and generalize better to diverse scenarios, although they require larger annotated datasets and may sacrifice interpretability. Consequently, theory-driven methods remain preferable when explainability is essential, whereas data-driven approaches are more effective for large-scale adaptive systems.

***Emotion experiences modeling*** Developing emotional intelligence in an intelligent agent requires more than modeling emotional elicitation or artificial emotion synthesis. A fundamental challenge in this domain is modeling emotional experience, particularly understanding how emotions influence cognitive processes such as inference, reasoning, memory, and decision-making (Hernández-Marcos & Ros, 2024). By incorporating emotional experiences, intelligent agents can develop more personalized behaviors, improve communication, and establish more natural interactions with humans. When emotions are generated during emotional elicitation, they affect various cognitive components and shape how agents interpret situations and select actions.

Several studies have investigated computational approaches for modeling emotional experiences and their influence on cognition. Self-dialogue Utility in Simulating Artificial Emotions (SUSAN) (Corvaia et al., 2025) introduces a novel perspective by modeling inner speech to simulate human-like emotional experiences. By leveraging inner dialogue, SUSAN attempts to infer contextual and cognitive processes underlying emotional experiences and is grounded in Damasio's theory of emotion. Similarly, (Hernández-Marcos & Ros, 2024) proposes a self-learning emotional framework based on the assumption that emotions correspond to specific temporal patterns in important environmental variables. The framework employs an unsupervised neural network trained on unlabeled experiential data to identify eight basic emotional patterns that are contextually coherent and resemble natural emotional dynamics.

Memory-based approaches have also been explored to model how emotional experiences influence learning and decision-making. EIAEC (Zall & Kangavari, 2024) learns context-dependent affective values by associating events with emotional experiences across different situations and stores these relationships in episodic memory. It also updates action activation levels in procedural memory according to the agent's emotional state, allowing emotions to directly influence decision-making. Similarly, (Juvina et al., 2018) associates emotional values and arousal levels with declarative memory elements by considering their usage frequency and reward signals derived from core affect dynamics. This enables the model to analyze how emotional information affects memory retrieval and subsequent decisions. Since emotional events are generally more memorable, emotional annotations may also provide valuable information for estimating memorability. However, most existing emotion recognition systems rely on external annotations, which may not fully represent individuals' subjective emotional experiences. To address this limitation, (Tsfasman et al., 2025) investigates the relationship between perceived emotions and collective memorability in conversational interactions through continuous annotations of emotional states and memorability in natural group settings.

Recent research has further explored the integration of emotional experiences into reinforcement learning (RL) agents, where emotional variables are often incorporated as reward signals to improve decision-making processes (Moerland et al., 2018; Alkam et al., 2025). (Zhang et al., 2024d) combines reinforcement learning with appraisal theory to establish a computational model that links reward processing, goal-directed learning, cognitive appraisal, and emotional experience. The model defines appraisal variables, including suddenness, goal relevance, goal conduciveness, and power, based on the component process model (CPM), and incorporates them into temporal difference learning. Unlike task-specific approaches, this framework can be applied to environments represented as Markov decision processes (MDPs), enabling the simulation of emotional responses during decision-making. Similarly, (Zhang et al., 2024b) models emotion as a dynamic and continuous process rather than a static state by integrating cognitive theories of emotion with computational rationality and reinforcement learning.

Beyond general decision-making, emotional experience modeling has also been applied to specific interaction scenarios. In automated negotiation, (Wu & Sun, 2025) develops an agent-based negotiation model using reinforcement learning, where emotions are quantified through the Weber-Fechner law and incorporated into the reward function of Q-learning. This allows emotional states, including emotional deception, to influence attribute preferences and negotiation strategies. Furthermore, (Berto et al., 2025) presents a cognitive architecture that simulates human cognitive functions, including perception, motivation, and decision-making, where agent behavior is influenced by affective states.

Emotions also play an important role in modeling self-control and adaptive behavior. Human self-control involves a conflict between immediate and delayed rewards, and previous computational models have often ignored the influence of emotional states. (Nikodemou & Christodoulou, 2024) addresses this limitation by incorporating emotion as a modulatory factor in reinforcement learning rewards, allowing the model to capture temporal variations in positive and negative emotional intensity and their effects on self-control.

In human-robot interaction, emotional experience modeling can improve the transparency and interpretability of agent behaviors. Reinforcement learning provides an effective mechanism for adaptive behavior generation, but the resulting policies are often difficult for humans to interpret due to the exploration-exploitation tradeoff. Since emotions naturally communicate intent and situational relevance, incorporating emotional expressions can make agent decisions more understandable. In this direction, (Nijeholt & Broekens, 2023) employs emotion expressions based on temporal differences (TD) to visualize the robot's learning process and improve the interpretability of reinforcement learning behaviors.

Research on emotion experience modeling has gradually moved from representing emotions as isolated states toward integrating emotions with memory, learning, and decision-making mechanisms. Recent approaches increasingly combine appraisal theories, reinforcement learning, and cognitive architectures to simulate dynamic emotional experiences and their influence on behavior. However, developing computational models that fully capture the complexity of human emotional experiences, including their subjective, contextual, and adaptive aspects, remains an open challenge.

Recent approaches indicate that different modeling strategies address complementary aspects of emotional experience. Memory-based cognitive architectures are particularly suitable for long-term personalization and adaptive decision-making, whereas reinforcement learning approaches are more effective in sequential decision problems where emotions dynamically influence action selection through rewards. Models based on appraisal theory provide stronger psychological interpretability, while data-driven frameworks offer greater flexibility in complex environments. Therefore, future emotionally intelligent agents will likely require hybrid frameworks that combine cognitive reasoning, memory, appraisal mechanisms, and reinforcement learning.

***Evaluation-related*** Evaluating computational models of emotion remains a significant challenge in developing emotionally intelligent agents. Existing evaluations often rely on subjective assessments within predefined scenarios, where the effectiveness of a model is determined based on whether the agent's behavior matches expected emotional responses. Two important criteria commonly considered for evaluating emotional behaviors are believability and social acceptability (Ojha et al., 2020a). Believability measures whether an agent's emotional responses appear natural, realistic, and contextually appropriate, while social acceptability evaluates whether these responses are consistent with social norms and expectations. Although an agent may demonstrate believable emotional behavior, it may not necessarily produce socially appropriate responses; therefore, both aspects are required for a comprehensive evaluation (Zall & Kangavari, 2022).

Despite the importance of these criteria, evaluating computational emotion models remains difficult due to the diversity of existing architectures and the lack of standardized evaluation frameworks. Each reviewed approach extends cognitive components for emotional elicitation from different perspectives, and a consistent conceptual structure across models is often missing. Furthermore, implementation details of many architectures are not publicly available, making direct comparison and reproduction challenging.

Several approaches evaluate emotional agents by simulating scenarios and comparing agent responses with human reactions under similar conditions (Sanchez et al., 2019b; Zall & Kangavari, 2024). EBDI (Jiang et al., 2007) provides a multi-agent environment for analyzing different emotional architectures by modifying various parameters and examining how emotions influence decision-making processes. Similarly, (Bourgais et al., 2016) investigates emotionally driven reasoning in multi-agent systems during critical situations, demonstrating the role of emotions in adaptive behavior. In another direction, E-VOX (Perez et al., 2017) evaluates emotional behavior through human-centered assessment, where users interact with the agent and rate its believability and acceptability using questionnaires.

Current evaluation methods mainly focus on assessing whether agents generate believable and socially appropriate emotional behaviors. However, the absence of unified benchmarks and standardized protocols limits meaningful comparison between computational emotion models. Future research should focus on developing comprehensive evaluation frameworks that consider emotional realism, cognitive consistency, social appropriateness, and adaptability across diverse scenarios.

Existing evaluation strategies complement rather than replace one another. Human-centered evaluations are preferable when assessing believability, trustworthiness, and social acceptability in interactive applications, whereas simulation-based evaluations provide more reproducible and controlled comparisons of agent behaviors across different scenarios. Benchmark-oriented evaluations facilitate quantitative comparison between computational models but currently remain limited by the absence of standardized protocols. Consequently, future evaluation frameworks should integrate objective benchmarking with human-centered assessment to provide more comprehensive and reliable evaluation of emotionally intelligent agents.

### 4.2.3 LLMs-Related Challenges

The emergence of large language models (LLMs) has introduced new opportunities for developing emotionally intelligent agents by providing advanced capabilities in language understanding, reasoning, and interaction. These models have the potential to transform intelligent agents from task-oriented systems into autonomous entities capable of making decisions, interacting naturally with humans, and adapting their behaviors based on emotional information. However, integrating emotional capabilities into LLM-based agents introduces several challenges related to emotional reasoning, alignment, and reliability.

Recent studies have explored how LLMs can support the development of emotionally intelligent agents. (Ma et al., 2025b) investigates the role of LLM-based agents in the Metaverse service ecosystem, where digital avatars, digital twins, and digital natives require advanced interaction capabilities. The study demonstrates that incorporating emotions into decision-making processes can improve user satisfaction and behavioral acceptability. To enhance reliability and social realism, the authors propose an explainable emotion alignment framework that incorporates factual information into agent decision-making. Similarly, (Raggioli et al., 2025) introduces an integrated architecture that combines cognitive and affective reasoning through LLMs. In this framework, LLMs generate beliefs from natural language inputs, while an internal emotional state influences intrinsic motivation and reward generation.

Beyond using LLMs as components of intelligent agents, recent research has increasingly focused on evaluating whether these models possess genuine emotional reasoning capabilities. (Tak & Gratch, 2024) examines the emotional reasoning abilities of LLMs based on appraisal theory and distinguishes between self-attributed emotions and the perception of others' emotions. The study shows that GPT-4 performs well in reasoning about emotionally evocative scenarios and often aligns with human judgments; however, its responses more closely resemble an observer perspective rather than human-like self-assessment. Similarly, (Gandhi et al., 2024) evaluates affective cognition in foundation models, including GPT-4, Claude-3, and Gemini-1.5-Pro, and investigates their ability to determine emotional elicitation in different scenarios. The findings indicate that these models can achieve strong alignment with human judgments, particularly when

chain-of-thought reasoning is applied.

Recent work has further moved beyond evaluating emotion recognition accuracy toward examining the cognitive mechanisms underlying emotional reasoning in LLMs. (Bhattacharyya et al., 2025) introduces CoRE, a large-scale benchmark based on cognitive appraisal theory, to analyze whether LLMs demonstrate coherent and interpretable reasoning when processing emotionally charged situations. The results suggest that although many models exhibit human-like appraisal patterns, they still struggle with subtle emotional distinctions and show inconsistencies across different cognitive dimensions, revealing limitations in current training approaches.

# 5 Emotional Text Synthesis

ETS in NLP focuses on producing emotionally resonant text through advanced techniques such as cross-modal emotion generation and emotion-controlled language models. The process involves creating written or spoken content that conveys specific emotional states, enabling machines to produce emotionally resonant and contextually appropriate responses.

It employs techniques such as style transfer, conditional generation, and fine-tuning of LLMs. This capability is a cornerstone of affective computing, which seeks to enhance HCI by imbuing systems with emotional intelligence (Schuller & Schuller, 2018).

Emotion generation plays a pivotal role in applications such as virtual assistants (Xue et al., 2024), mental health support systems (Feng, 2024), and personalized content creation (Abilbekov et al., 2024), where the ability to express and evoke emotions significantly impacts user engagement and satisfaction (Becker et al., 2024). For instance, virtual assistants equipped with emotion generation capabilities can provide more empathetic and context-aware responses, fostering a more natural and human-like interaction (Zheng et al., 2021).

In NLP, techniques like sentiment analysis and emotion detection lay the groundwork for generating emotionally charged text by identifying and interpreting emotional cues in existing data (Truong, 2024). Advances in AI, particularly in generative models and LLMs, have further propelled the field by enabling fine-grained control over emotional expression in generated text (Becker et al., 2024). Here, we first briefly introduce approaches to generate emotional text. Then, we proceed with the challenges within this area and how different research studies have tried to overcome them.

## 5.1 Approaches

Before discussing specific approaches, it is essential to acknowledge that emotional text synthesis methods differ fundamentally based on the underlying emotion model employed.

**Categorical emotion models**, such as Ekman's basic emotions (Ekman, 1992) or Plutchik's wheel (Plutchik, 1980), represent emotions as discrete classes (e.g., joy, sadness, anger). These models facilitate straightforward classification and generation tasks but may oversimplify the continuous nature of emotional experiences and struggle with ambiguous or blended emotions. In contrast, **dimensional emotion models**, such as the valence-arousal-dominance (VAD) model or circumplex models, represent emotions as points in a continuous space, enabling fine-grained control over emotional intensity and the representation of subtle emotional states. However, dimensional models often lack the intuitive interpretability of categorical labels and may require more complex training procedures. The choice of emotion model profoundly influences data annotation strategies, model architecture design, evaluation metrics, and the interpretation of generated outputs. Hybrid approaches that combine categorical and dimensional representations are increasingly common, leveraging the strengths of both paradigms.

***Style Transfer*:** Style transfer alters the emotional tone of a text while maintaining its meaning, which is beneficial for personalized content creation. In categorical models, style transfer typically involves mapping text from one discrete emotion class to another (e.g., converting angry text to calm text), often

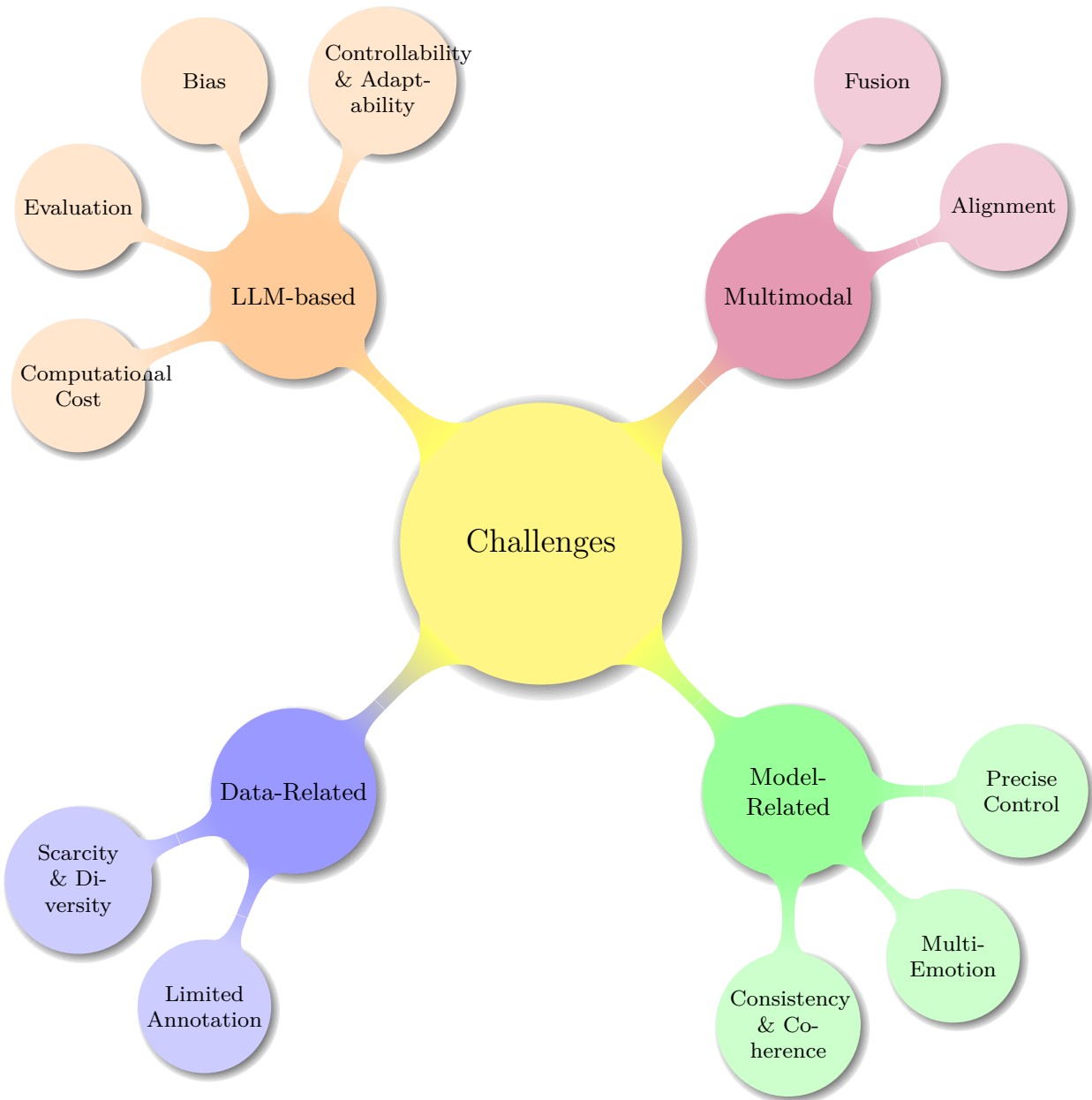

Figure 10: Challenges in ETS

using class-conditional GANs or encoder-decoder architectures with emotion embeddings. Dimensional models, conversely, enable continuous manipulation of emotional attributes along valence and arousal dimensions, allowing for more nuanced transformations. However, style transfer often struggles with maintaining semantic coherence and emotional consistency in more complex texts. Innovations such as the lexicon-based attention mechanism and methods separating content from style (Fu et al., 2018) have been developed to improve emotional nuance and address the lack of parallel corpora.

***Conditional Generation***: Conditional generation controls emotional content through specific prompts, enhancing applications like virtual assistants. The conditioning mechanism differs significantly between emotion models: categorical approaches use one-hot encoded emotion labels or emotion-specific embeddings, while dimensional approaches condition on continuous VAD values or arousal-valence coordinates.

Table 4: Summary of challenges and solutions in emotional text synthesis

| Challenge | Sub-challenge | Solutions |
|---|---|---|
| **Data-Related** | Scarcity | Data augmentation (Firdaus et al., 2023); Transfer learning (Li et al., 2018b); SentiGAN (Wang & Wan, 2018); CS-GAN (Li et al., 2018b); LeakGAN (Guo et al., 2018) |
| | Limited Annotation | Desire annotation (Jia et al., 2022) |
| **Model-Related** | Emotional Consistency & Coherence | Appraisal theories (Xia & Ding, 2019; Singh et al., 2020; Resendiz & Klinger, 2023); Emotion embeddings (Tan et al., 2023); MSEG (Firdaus et al., 2023); SEPRG (Firdaus et al., 2021); RCVAE (Zhou & Wang, 2017); DecoupledESC (Zhang et al., 2025a); CARE (Zhu et al., 2026); COMPEER (Wang et al., 2025c) |
| | Handling Multi-Emotion | Emotion blending (Ghosh et al., 2017; Resendiz & Klinger, 2023); ECM (Zhou et al., 2018) |
| | Precise Emotion Control | FUDGE (Yang & Klein, 2021); MOAEP (Resendiz & Klinger, 2024); Prefix-tuning (Qian et al., 2022); DEXPERTS (Mahmood & Manning, 2023) |
| **Cross-Modal** | Alignment | Hierarchical attention (Zhang et al., 2017); Word-level alignment (Zhang et al., 2017) |
| | Fusion | Time-dependent fusion (Zhang et al., 2017); FaceChat (Alnuhait et al., 2023); Omni-perception Pre-Trainer (Kaur et al., 2024); MAGIC (Su et al., 2022); emoTTS (Luo et al., 2024); FIRES (Wang et al., 2025a) |
| **LLM-based** | Controllability & Adaptability | Coda (Evuru et al., 2024); MOPO (Resendiz & Klinger, 2024); Emotion Vectors (Dong et al., 2025) |
| | Bias | Debiasing techniques (Sheng et al., 2021) |
| | Computational Cost | Affective prompt-tuning (Gu & He, 2024) |
| | Evaluation | EmoBench (Sabour et al., 2024); Kardia-R1 (Yuan et al., 2025) |

Models such as the co-attention neural network (Li et al., 2018a) and Emotional Tacotron (Lee et al., 2017) exemplify this approach, though they require high-quality datasets and struggle with multi-emotion scenarios, particularly when the underlying model does not naturally support emotion blending (as in strict categorical approaches). Affective Chatbot (Jiang et al., 2022) and emotion prediction in TTS systems (Yoon et al., 2022) highlight its versatility.

***Fine-tuning LLMs***: Leveraging models like GPTs and BERT enables high-quality emotional text generation with minimal training (Singh et al., 2020). The fine-tuning strategy depends critically on the emotion model: categorical models typically add classification heads or modify output layers to predict discrete emotion labels, while dimensional models incorporate regression heads to predict continuous emotional dimensions. This method requires significant computational resources and careful bias management, with the choice of emotion model influencing both the loss function design and the evaluation protocols used to assess generated text.

## 5.2 Challenges

In affective text generation, defining and representing emotions presents significant challenges, crucial for accurately conveying emotional states and their contextual expressions. Accurately capturing and generating emotions requires a deep understanding of emotional states and their contexts. Foundational models like Ekman's six basic emotions (Ekman, 1992) and Plutchik's wheel of eight emotions (Plutchik, 1980) provide structured frameworks for categorizing emotions, highlighting their interconnected nature (Zhang et al., 2024a). This section explores these complexities, emphasizing their foundational role in emotion generation and how different studies have tried to overcome these challenges. Main challenges in this area and their related sub-challenges are depicted in Figure 10. Moreover, Table 4 highlights different studies and their solutions to these issues.

### 5.2.1 Data-Related Challenges

The performance of affective text generation models is heavily dependent on the availability of large, high-quality, and culturally diverse datasets. The scarcity of annotated datasets limits a model's ability to replicate nuanced emotional states, which in turn affects the accuracy of the generated text (Li et al., 2023a). This is particularly evident in the lack of annotations for emotions like desire (Jia et al., 2022). A fundamental challenge in affective text generation lies in the data collection and annotation process itself. Collecting emotionally rich text data requires careful consideration of context, speaker intent, and cultural nuances. Annotation of emotional text presents unique difficulties, as emotions expressed through text are often subtle, context-dependent, and subject to annotator disagreement. The lack of standardized annotation protocols across different emotional frameworks (e.g., discrete emotion categories versus dimensional models) further complicates the creation of consistent training datasets. Moreover, the subjective nature of emotion perception leads to inter-annotator reliability issues, where different annotators may assign different emotional labels to the same text. Crowdsourcing approaches, while scalable, often suffer from annotation quality concerns, as annotators may lack the expertise to distinguish between similar emotions or may be influenced by their own cultural and linguistic backgrounds. These annotation challenges directly impact model training, as inconsistent or noisy labels can lead to poor generalization and reduced performance in real-world applications. To mitigate these challenges, strategies such as data augmentation and transfer learning are used to enhance the diversity of training data and improve model generalization (Li et al., 2018b; Firdaus et al., 2023). GANs have been used to address data scarcity. SentiGAN (Wang & Wan, 2018) and CS-GAN (Li et al., 2018b), for example, enhance emotional diversity in generated text, and LeakGAN (Guo et al., 2018) tackles challenges related to discrete outputs and provides continuous guidance.

### 5.2.2 Model-Related Challenges

Generating text that captures multiple emotions, emotional transitions, and maintains coherence is a complex task. The complexity of these challenges is fundamentally shaped by the underlying emotion model. Categorical emotion models face distinct difficulties in representing blended emotions (e.g., bittersweet combinations of happiness and sadness) or emotional transitions, as discrete class boundaries do not naturally accommodate intermediate states. Multi-label classification approaches or emotion intensity scores can partially address this limitation, but they increase model complexity and annotation requirements. Dimensional emotion models, while inherently suited to representing continuous emotional states and smooth transitions, introduce challenges in interpretability and controllability—it is less intuitive for users to specify desired emotions as VAD coordinates than as categorical labels. Furthermore, the mapping between dimensional representations and human-interpretable emotion terms is not one-to-one, complicating evaluation and user interaction. Hybrid approaches that maintain both categorical labels and dimensional coordinates offer a potential solution but require careful design to ensure consistency between representations.

Emotion blending and sequential emotion modeling are key approaches to integrate multiple emotions into a coherent expression (Ghosh et al., 2017). Despite progress, achieving fine-grained control over emotional expressions remains a significant challenge (Zhang et al., 2023b). This is particularly difficult

for non-native speakers who often struggle with expressing mixed emotions, highlighting the need for models that can replicate these nuanced expressions (Feng, 2024). The Emotional Chatting Machine (Zhou et al., 2018) addresses coherence with specified emotions through the use of emotion embeddings and attention mechanisms. For precise emotion control, techniques like FUDGE (Yang & Klein, 2021) have been developed to offer fine-grained control over emotional intensity and category. Advanced model architectures like prefix-tuning (Qian et al., 2022) and DEXPERTS (Mahmood & Manning, 2023) have been proposed to address challenges like mode collapse. The SemEval(Mohammad et al., 2025) shared task on bridging the gap in text-based emotion detection highlights the importance of developing culturally aware and context-sensitive models that can accurately interpret and generate emotions across different languages and cultures. Modifying generative models to incorporate new emotional attributes without compromising the integrity of the content is a major obstacle, exacerbated by the non-differentiable nature of discrete text (Wang & Wan, 2018). Furthermore, cultural and linguistic diversity introduces additional complexity, necessitating advanced frameworks that merge linguistic and psychological theories with machine learning techniques (Jain et al., 2017). The absence of non-verbal cues in text poses another layer of difficulty, often leading to misinterpretations, especially among non-native speakers. Diffusion and flow-based models are also at the forefront of generating nuanced emotional text. Diffusion models, such as Diffusion-LM (Li et al., 2022b), iteratively transform noise into structured data, allowing for precise emotional control. Flow-based models use invertible transformations for high-dimensional data generation, capturing emotional nuances while maintaining linguistic quality. These models can integrate affective parameters, enhancing the expressiveness of conversational language (Ghosh et al., 2017). DVAE-interVA (Chen et al., 2023a) is proposed to solve the emotion-semantic entanglement problem in continuous sentiment text generation via adversarial sentiment decoupling, continuous sentiment embeddings, denoising training, and interactive attention to maintain emotional consistency throughout generation. Another challenge is maintaining emotional consistency and coherence throughout a piece of text or a conversation. This requires models to sustain a consistent emotional tone and narrative. Integrating nuanced understanding, such as appraisal theories, is one approach to enable contextually relevant emotional expression (Truong, 2024; Xia & Ding, 2019; Resendiz & Klinger, 2023). However, the variability of language and the informal nature of social media continue to make emotional consistency a complex problem (Tan et al., 2023). Context understanding is critical, as many chatbots struggle to maintain coherent conversations and lose track of the emotional context (Shum et al., 2018). To address this, methods like MSEG (Firdaus et al., 2023) have been developed to generate responses that align with identified emotions. The SEPRG model (Firdaus et al., 2021), for instance, aims to maintain emotional connections while adhering to a consistent persona. Disentangled representations are vital for isolating emotion-specific features, enabling precise emotional expression while maintaining semantic coherence. This is particularly crucial for applications like virtual assistants and mental health support systems (Truong, 2024). The integration of these representations with LLMs, like the MOPO method, improves emotion control and prediction accuracy (Resendiz & Klinger, 2024). The RCVAE model demonstrates the effectiveness of disentangled representations in generating emotionally rich responses from tweets and emojis (Zhou & Wang, 2017). A key challenge in emotional support conversation is that supervised fine-tuning of large language models produces psychologically inconsistent responses, and applying direct preference optimization is hindered by the entanglement of psychological strategies and response content in existing emotional support conversation data, leading to ambiguous optimization objectives. DecoupledESC (Zhang et al., 2025a) addresses this by decomposing the emotional support conversation task into two sequential subtasks, strategy planning and empathic response generation, inspired by Gross's Extended Process Model of Emotion Regulation. An Inferential Preference Mining method constructs high-quality preference pairs for each subtask, and each is independently trained via supervised fine-tuning followed by direct preference optimization. This decoupled approach reduces preference bias and improves both strategic soundness and emotional appropriateness of generated responses. Current emotional support conversation approaches often overlook the deeper cognitive reasoning processes that underpin effective emotional support, focusing instead on data augmentation and synthetic corpus construction. CARE (Zhu et al., 2026) proposes a framework that strengthens cognitive reasoning in emotional support conversation without relying on large-scale synthetic data. It leverages the original emotional support conversation training set to guide models in generating logically coherent and supportive responses, then employs reinforcement learning to refine and reinforce the reasoning process. Experimental results demonstrate significant improvements in both the logical soundness and supportive quality of generated

responses. Generating emotionally supportive text that is both empathetically grounded and non-repetitive remains challenging, as current models lack deep empathetic reasoning rooted in psychological principles, and reinforcement learning-based training often suffers from entropy collapse leading to repetitive outputs. COMPEER (Wang et al., 2025c) proposes controllable empathetic reasoning that combines natural language reasoning with structured psychological steps. It constructs a fine-grained dataset annotated with reasoning correctness and response preferences, and employs reinforcement learning with a unified process-outcome reward model for precise feedback. To mitigate repetitiveness, COMPEER introduces personality-based dialogue rewriting and a redundancy-aware reward reweighting strategy.

### 5.2.3 Multimodal-Related Challenges

Cross-modal learning enhances emotional accuracy by integrating text with other modalities like speech and facial expressions. Since this type of challenge is associated with the following sections in emotional content synthesis, here some challenges are underscored briefly and meticulous information will be delivered there. A key challenge in this area is alignment, which involves synchronizing information from different sources, such as aligning visual cues with corresponding text segments. This is particularly difficult when the relationship between modalities is not explicit, as is often the case with text that only describes a video or audio track in general terms (Zhang et al., 2017). Another significant challenge is fusion, which refers to the method of combining features from heterogeneous data sources. Many early models fused modalities only at an abstract level, which fails to capture the time-dependent interactions between them (Zhang et al., 2017). Addressing these challenges is crucial for creating seamless and contextually aware affective experiences. Systems like FaceChat (Alnuhait et al., 2023) have demonstrated real-time improvements in emotional accuracy by combining speech recognition, NLP, and facial analysis. The Omni-perception Pre-Trainer (Kaur et al., 2024) integrates emotional cues from multiple modalities to enhance the quality of generated text. The MAGIC model highlights the potential of cross-modal learning to improve emotional accuracy and adapt text to diverse user needs (Su et al., 2022). The emoTTS model (Luo et al., 2024) is another example of a cross-modal application that enhances emotion control and generation accuracy. Chain-of-thought based emotional support conversation methods typically employ rigid, text-only reasoning, which limits adaptability in dynamic multimodal interactions and introduces reasoning noise that degrades support quality. FIRES (Wang et al., 2025a) introduces "Flexible Thinking" for multimodal emotional support conversation, enabling models to adaptively select contextually relevant reasoning aspects, including visual scene, emotion, situation, and response strategy. The framework integrates supervised fine-tuning for initial learning with reinforcement learning for refinement, directly linking thinking processes to response quality via tailored rewards. Experiments on the MESC and EMOTyDA datasets demonstrate improved quality and generalizability of emotional support responses through this adaptive multimodal reasoning approach.

***LLM-Based Challenges***: Recent research has also focused on leveraging LLMs and developing novel techniques to enhance emotional expression while maintaining semantic coherence. The high computational cost associated with fine-tuning large models for emotional text generation is of importance. APT-LM (Gu & He, 2024), a parameter-efficient solution, overcomes computational inefficiency in emotional text generation by freezing pre-trained language models and using minimal affective parameters through prompt-tuning, enhanced by affective decoding that systematically strengthens emotional expression at multiple linguistic levels while preserving fluency. Another key challenge is the evaluation of emotional intelligence in LLMs. While models are becoming more sophisticated, their ability to understand and apply emotional intelligence is still limited. To this end, benchmarks like EmoBench (Sabour et al., 2024) have been developed to assess the emotional intelligence of LLMs through a series of hand-crafted multiple-choice questions that cover both emotional understanding and application. This benchmark has revealed that even the most advanced LLMs still have a considerable gap to bridge to reach human-level emotional intelligence. Multi-Objective Prompt Optimization (MOPO) (Resendiz & Klinger, 2024), a method that optimizes prompts for LLMs to generate text meeting multiple objectives simultaneously, specifically transforms the desired emotion and fits the stylistic requirements of different domains. MOPO employs a three-layer optimization process: Layer-1 generates initial text prompts, Layer-2 refines these by paraphrasing or combining them using genetic operations, and Layer-3 provides fixed prompts to guide

the optimization and improves performance by up to 2% for individual objectives while providing balanced, flexible solutions, enhancing the adaptability and effectiveness of affective text generation across diverse applications. Generating affective text that conveys specific emotions with controlled intensity and topic relevance, while preserving grammatical accuracy, has been addressed by enhancing the GPT-2 model and integrating emotional priors and applying a gradient descent-based perturbation technique (Singh et al., 2020). CoDa (Evuru et al., 2024) introduces a training-free data augmentation framework for low-resource NLP that extracts simple heuristic-based constraints from limited training data and verbalizes them to prompt off-the-shelf instruction-tuned LLMs, achieving controlled generation that balances diversity and consistency while outperforming existing methods across 11 datasets. While large language models exhibit strong reasoning capabilities, they struggle to express emotions in a consistent, controllable, and contextually appropriate manner, and existing prompt-based or fine-tuning-based methods lack flexibility or require costly retraining. The Emotion Vector framework (Dong et al., 2025) addresses this by extracting latent representations from internal activation shifts between neutral and emotion-conditioned responses. By injecting these vectors into the hidden states of pretrained large language models during inference, the method enables fine-grained, continuous modulation of emotional tone and intensity without any additional training or architectural modification, while preserving semantic fidelity and linguistic fluency. Existing emotional support conversation systems rely on situation-centric datasets that lack persistent user identity, which limits the capture of personalized affective nuances. Moreover, opaque and coarse reward signals hinder the development of verifiable empathetic reasoning in large language models. Kardia-R1 (Yuan et al., 2025) introduces KardiaBench, a large-scale user-grounded benchmark comprising 178,080 question-answer pairs anchored to 671 real-world user profiles, and proposes Rubric-as-Judge Empathetic Reinforcement Learning. This group relative policy optimization-based method uses explainable, human-aligned rubric rewards that tightly couple user understanding, emotional inference, and supportive response generation, enabling interpretable stepwise empathetic cognition with consistent improvements in emotion accuracy, empathy, and persona consistency across multiple large language model backbones. Finally, another hurdle is modifying generative models to incorporate new emotional attributes without compromising the integrity of the content, a problem exacerbated by the non-differentiable nature of discrete text (Wang & Wan, 2018). The community is also addressing the ethical implications of affective computing, particularly the issue of bias. As models become more capable of generating emotional text, there is a growing need to ensure that they are used responsibly and do not perpetuate biases present in the training data (Sheng et al., 2021). Integration of disentangled representations with LLMs improves emotion control and prediction accuracy.

### 5.3 Comparative Synthesis of Emotional Text Synthesis Strategies

The diverse landscape of emotional text synthesis methods presents practitioners with multiple options, each offering distinct advantages under specific conditions. This section synthesizes the key trade-offs to guide method selection based on application requirements, resource constraints, and desired emotional control granularity.

**Style Transfer vs. Conditional Generation vs. Fine-tuned LLMs:** Style transfer methods excel when the primary goal is transforming the emotional tone of existing text while preserving semantic content, making them ideal for content adaptation (e.g., converting customer feedback from negative to neutral tone). However, their dependence on parallel or pseudo-parallel corpora and limited scalability to complex emotional nuances restrict their applicability. Conditional generation approaches, by contrast, offer superior control over emotional attributes through explicit conditioning signals and are well-suited for applications requiring diverse emotional outputs from neutral inputs (e.g., personalized chatbot responses). Fine-tuned LLMs dominate in scenarios demanding high-quality, contextually rich emotional text with minimal task-specific training data, particularly when leveraging pre-trained knowledge. However, their computational cost and risk of bias propagation necessitate careful consideration in resource-constrained or fairness-critical applications.

**Categorical vs. Dimensional vs. Hybrid Emotion Models:** Categorical emotion models are preferable when interpretability and alignment with human emotion labels are paramount (e.g., sentiment analysis dashboards, user-facing emotion selection interfaces), and when annotated datasets with discrete emotion labels are readily available. Dimensional models (VAD-based) provide superior performance for applica-

tions requiring fine-grained emotion intensity control, smooth emotional transitions, or modeling of ambiguous/mixed emotions (e.g., expressive virtual agents, therapeutic dialogue systems). Hybrid approaches combining categorical labels with dimensional intensity scales offer the best of both worlds, enabling intuitive user interaction while supporting nuanced emotional expression, though at increased model complexity.

**Data Augmentation Strategies:** For low-resource scenarios (<10K labeled samples), transfer learning from pre-trained LLMs combined with back-translation or paraphrasing-based augmentation yields the highest performance gains. GAN-based synthetic data generation (e.g., SentiGAN, CS-GAN) proves most effective when training data exhibit severe class imbalance or when generating rare emotion categories, though quality control remains critical. Semi-supervised approaches leveraging unlabeled emotional text are optimal when large unlabeled corpora are available but annotation budgets are constrained.

**Evaluation and Deployment Considerations:** For real-time applications (e.g., conversational AI), lightweight conditional generation models or efficiently fine-tuned small LLMs (e.g., DistilBERT-based) offer the best latency-accuracy trade-off. When emotional authenticity and human preference alignment are critical (e.g., mental health support, empathetic dialogue), LLM-based methods with reinforcement learning from human feedback (RLHF) or preference optimization (e.g., MOPO, Coda) demonstrate superior performance, albeit at higher computational cost. For multi-emotion scenarios requiring coherent expression of blended emotional states, disentangled representation learning approaches (e.g., DVAE-interVA, DecoupledESC) combined with dimensional emotion models provide the most flexible and accurate synthesis.

In summary, method selection should be guided by: (1) *data availability*—use transfer learning/LLMs for low-resource settings, GANs for imbalanced data; (2) *emotion model requirements*—categorical for interpretability, dimensional for granularity, hybrid for versatility; (3) *computational budget*—conditional generation for efficiency, fine-tuned LLMs for quality; and (4) *application domain*—style transfer for content adaptation, LLMs with RLHF for empathetic interaction, disentangled models for multi-emotion synthesis.

## 6 Emotional Speech Synthesis

Emotional speech generation is a key area in the field of AI and HCI, aimed at enabling machines to produce emotionally natural, meaningful speech. This process involves transferring the intended emotions in the speech generation process. In recent years, advancements in machine learning, NLP, and signal processing have propelled speech generation technologies forward, enabling virtual assistants, chatbots, and social robots to engage with users in more natural and fluid ways. Emotional speech generation can be categorized into two primary subfields: Emotional voice Conversion and ESS. These areas, while related, serve different purposes and are implemented through distinct technologies and techniques. Even though TTS technology has achieved mature and reliable outcomes, ESS still faces significant challenges. Developing robust EVC systems capable of performing any-to-any emotion conversion could address some of the key issues related to ESS in a two-stage manner and pave the way for substantial progress in this field. Inspired by (Triantafyllopoulos et al., 2023), we present a typical ESS workflow that leverages EVC. Initially, based on the stimuli and the given context, an appropriate response is generated, along with the recognition of the desired emotion to be imbued. A TTS system then generates the spectrum of the target speech in a neutral format. Finally, the EVC component processes the input speech and converts it to the desired emotional output. During the generation and utterance of emotional speech, capturing feedback and stimuli from the environment is crucial. This is because the state of the situation may change, requiring adjustments to both the emotion being expressed and the context of the speech. In the following, the first approaches for emotional voice generation are discussed. Then, we dive into the challenges in the realm and propose solutions for them.

### 6.1 Approaches

The field of emotional speech synthesis is fundamentally influenced by the choice of emotion representation model. **Categorical emotion models** (e.g., discrete emotion labels such as happy, sad, angry, neutral) are prevalent in ESS due to their intuitive interpretability and alignment with traditional speech emotion recognition databases. These models typically employ one-hot encoding or emotion-specific embeddings to condition

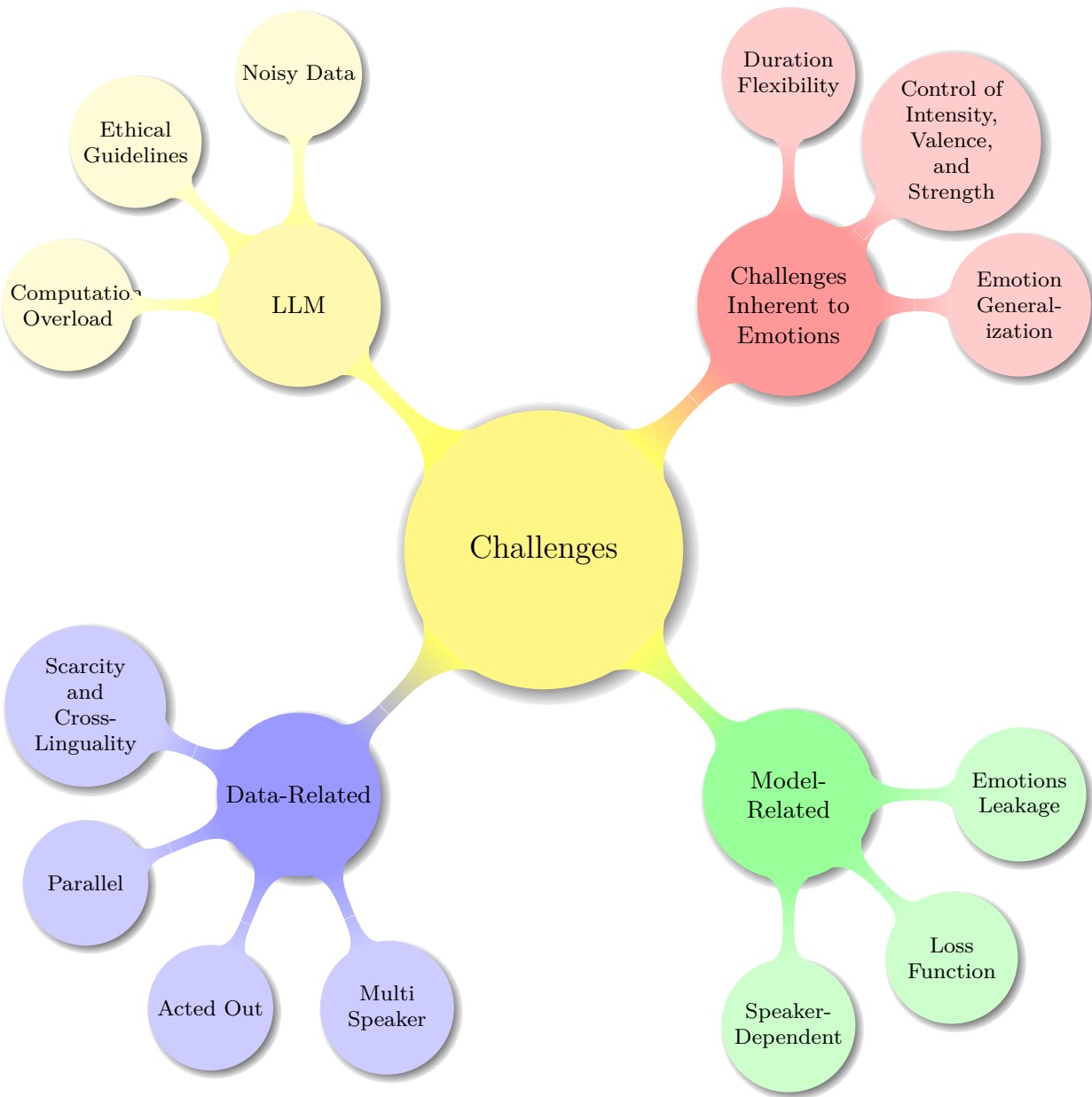

Figure 11: Challenges in ESS

speech generation on discrete emotion classes. However, categorical models face inherent limitations in representing emotion intensity variations, subtle emotional nuances, and blended emotional states, which are common in naturalistic speech. **Dimensional emotion models**, such as valence-arousal-dominance (VAD) or valence-arousal models, address these limitations by representing emotions as continuous coordinates in a multidimensional space. This representation enables fine-grained control over emotional intensity and facilitates the modeling of emotional transitions and mixed emotions. Methods like EmoConv-Diff (Prabhu et al., 2024), which condition on continuous arousal dimensions, exemplify this approach. However, dimensional models require more sophisticated annotation procedures (continuous rating scales rather than discrete labels) and may lack the intuitive interpretability of categorical schemes, complicating user interaction and evaluation. Hybrid approaches that combine categorical emotion classification with dimensional intensity

control (e.g., predicting both an emotion category and its intensity) are increasingly common, offering a balance between interpretability and expressiveness.

***ESS***: ESS or Affective Speech Synthesis focuses on generating speech imbued with emotional expressivity, enabling machines to produce human-like, emotionally rich communication(Triantafyllopoulos et al., 2023). ESS primarily relies on text-to-speech (TTS) technology and plays a crucial role in enhancing HCI and enriching audio broadcast scenarios(Lei et al., 2022). The emotion model employed directly influences the architecture and training objectives: categorical ESS systems typically use emotion-conditioned acoustic models with separate emotion embeddings or multi-task learning frameworks, while dimensional ESS systems incorporate regression heads to predict continuous emotional parameters that modulate prosodic features (pitch, duration, energy). Conversational Emotional Speech Synthesis (CESS) builds upon ESS by incorporating the ability to maintain context from previous interactions. While ESS focuses on generating speech with emotional expressiveness, CESS enhances this by enabling the system to remember and reference past conversations. This memory capability distinguishes CESS from ESS, which typically lacks such continuity.

***Emotional Voice Conversion***: Emotional voice conversion (EVC) involves transforming the emotional expression in speech from a source emotion to a target emotion while preserving both the linguistic content and the speaker identity (Zhou et al., 2022b). In contrast, voice conversion (VC) focuses on modifying one's voice to sound like another's without changing the linguistic content (Sisman et al., 2020). EVC involves altering specific acoustic features, such as pitch, timbre, and formant frequencies, to reflect the target emotion while keeping the original speech intact. This process can be used in various applications, including virtual assistants(Elgaar et al., 2020), Human-Robot Interaction(Crumpton & Bethel, 2016), and HCI(Pittermann et al., 2010). Early approaches to EVC relied on statistical models such as Gaussian Mixture Models (GMMs)(Aihara et al., 2012), which modeled the relationship between source and target features, and techniques like frequency warping(Sheikhan et al., 2012), which directly modified spectral features to reflect emotional changes. While effective for their time, these methods were limited by their reliance on parallel data and their inability to capture complex, non-linear transformations. With advancements in machine learning, modern approaches have shifted towards deep learning techniques(Walczyna & Piotrowski, 2023), which enable more flexible and robust emotional conversions. Encoder-decoder frameworks have been widely adopted to disentangle speaker identity, linguistic content, and emotional features, facilitating more effective style transfer(Zhou et al., 2021a).

## 6.2    Challenges

In this section, we investigate the challenges associated with emotional speech generation from diverse standpoints. Consequently, we review the studies on these challenges and their proposed solutions. As illustrated in Figure 11, we categorize these challenges into four main domains: data-related challenges, model-related challenges, challenges inherent to emotions, and challenges related to LLMs. Table 5 illustrates these categories, detailing their associated sub-issues and highlighting studies attempting to mitigate these challenges. Following, we will discuss challenges and methods proposed to address them.

### 6.2.1    Data-Related Challenges

One of the main issues in developing emotional speech generation methods is related to speech datasets. Data-related challenges encompass issues such as the reliance on parallel data(Liu et al., 2020; Zhou et al., 2020b; Prabhu et al., 2024), data scarcity(Schnell et al., 2021; Zhou et al., 2022b), monolingual datasets(Zhou et al., 2022b), and the need for realistic (not acted), multi-speaker datasets(Lotfian & Busso, 2019). The collection and annotation of emotional speech data present substantial methodological challenges that directly impact the quality and applicability of emotional speech synthesis systems. Unlike neutral speech, collecting emotionally expressive speech requires controlled elicitation protocols or naturalistic recording scenarios, each with distinct limitations. Laboratory-based approaches using emotion induction techniques (e.g., recalling emotional memories, reading scripted emotional content) offer controlled conditions but often result in acted or exaggerated expressions that may not reflect authentic emotional speech. Conversely, collecting spontaneous emotional speech from real-world interactions (e.g., call centers, therapy sessions, podcasts) provides more naturalistic data but introduces ethical concerns regarding consent and privacy, as well as technical

challenges related to audio quality, background noise, and speaker variability. The annotation process for emotional speech is further complicated by the multimodal nature of emotion expression. Annotators must consider not only lexical content but also prosodic features (pitch, intensity, duration), voice quality, and speaking rate, all of which contribute to emotional perception. Inter-annotator agreement tends to be lower for emotional speech than for neutral speech, particularly for subtle or complex emotions. Additionally, the choice of annotation scheme—whether categorical (discrete emotion labels), dimensional (valence-arousal-dominance), or appraisal-based—significantly influences the applicability of resulting datasets to different synthesis tasks. The temporal granularity of annotations (utterance-level versus frame-level) also affects model training, with finer-grained annotations enabling more precise emotion control but requiring substantially more annotation effort.

There are two major types of EVC according to data: 1)Parallel and 2)Non-Parallel. In the former, methods utilize pairs of utterances that contain the same content from the same speaker but are expressed with different emotions. During training, the conversion model learns to map features from the source to the target emotion using these paired feature vectors. Most EVC systems are implemented in this manner cite zhou2022. In contrast, non-parallel EVC involves learning to map source speech features (e.g., spectral, prosodic) to target emotional features without relying on paired utterances (Gao et al., 2018). Parallel data requires both input and output audio recordings for each utterance, which is resource-intensive and time-consuming to collect. Consequently, there is a growing preference for non-parallel data to alleviate the burdens associated with parallel datasets.

Adversarial Generative Network (GAN) models, such as CycleGAN(Zhou et al., 2020a) and StarGAN(Rizos et al., 2020), have proven particularly powerful in non-parallel EVC scenarios by learning mappings between source and target emotional styles without requiring paired data. CycleTransGAN(Fu et al., 2021), a CycleGAN-based model enhanced with transformers for non-parallel EVC, uses transformers to capture temporal intra-relations over wider receptive fields. Also, it adopts curriculum learning, gradually increasing the frame length during training to improve feature comprehension, and incorporates fine-grained discriminators for detailed emotional mapping. In (Meftah et al., 2023), the use of the StarGANv2-VC framework (Li et al., 2021) for EVC in English has been explored. Current systems often struggle with the ability to handle multiple speakers and emotions, especially when limited data is available. This research aims to address these gaps by evaluating the StarGANv2-VC model across various configurations, including speaker-dependent, gender-dependent, and gender-independent scenarios. CycleGAN (Liu et al., 2020) is a non-parallel EVC model that incorporates two discriminators to differentiate between natural and converted speech, along with a classifier to identify the underlying emotion from both types of speech. Recently, diffusion models have emerged as a promising approach in emotional speech generation (Ma et al., 2024), leveraging their ability to model complex data distributions and generate high-quality emotional transformations. EmoConv-Diff (Prabhu et al., 2024) presents a diffusion-based model utilizing non-parallel and in-the-wild data. It effectively disentangles lexical content, speaker identity, and emotional information through a diffusion-based decoder, enabling precise emotion transformation via reverse stochastic differential equations.

Current emotional TTS systems struggle to authentically capture human emotions due to reliance on over-simplified emotional labels and single-modality inputs. To solve this, the authors propose UMETTS (Li et al., 2025a), featuring (1) EP-Align that uses contrastive learning to align emotional features across text, audio, and visual modalities, and (2) EMI-TTS that integrates these aligned embeddings with state-of-the-art TTS models, resulting in significantly improved emotion accuracy and speech naturalness. EmoCat (Schnell et al., 2021), a language-agnostic EVC model designed to convert neutral speech to emotional speech, incorporates gradient inverter (Ganin & Lempitsky, 2015) blocks to suppress emotional leakage and uses a VAE-based encoder-decoder structure inspired by the CopyCat (Karlapati et al., 2020). EmoCat effectively reduces the emotional training data required in the target language by leveraging emotional data from another language. As a result, it achieves high-quality emotional conversion in German, using only 45 minutes of German emotional data while being supported by extensive emotional datasets in US English. EmoSphere++ (Cho et al., 2025a) introduces an emotion-adaptive spherical vector (EASV) that represents emotional style through angular position and intensity through radial distance in a spherical coordinate system, enabling fine-grained emotion control without predefined labels. The framework employs a joint attribute style encoder with orthogonality loss to effectively disentangle speaker identity and emotional features, achieving high-quality zero-shot emotional speech synthesis for unseen speakers while eliminating the

Table 5: Summary of challenges and solutions in emotional speech synthesis

| Challenge | Sub-challenge | Solutions |
|---|---|---|
| **Data-Related** | Parallel Data | Non-parallel data with Adversarial Networks (Zhou et al., 2020a; Liu et al., 2020; Rizos et al., 2020); Transformer (Fu et al., 2021); MSP-Podcast dataset (Lotfian & Busso, 2019) |
| | Scarcity & Cross-linguality | Emotional datasets in other languages (Schnell et al., 2021); ESD dataset (Zhou et al., 2022b); EmoSphere++ EASV (Cho et al., 2025a); UMETTS contrastive learning (Li et al., 2025a) |
| | Mono Speaker | ESD dataset (Zhou et al., 2022b) |
| | Acted vs. Natural | MSP-Podcast dataset (Lotfian & Busso, 2019) |
| **Model-Related** | Loss Function | Disentangled loss (Chou et al., 2024); Perceptual losses (Zhou et al., 2022c); Mutual information minimization (Yang et al., 2025) |
| | Speaker Dependency | VAW-GAN encoder-decoder (Zhou et al., 2020b); Seq2seq with attention (Choi & Hahn, 2021); Self-supervised distillation (Cho et al., 2025b) |
| | Emotion Leakage | Gradient reversal layer (Schnell et al., 2021) |
| | Long Sequence Processing | Long-sequence processing (Fu et al., 2021); Emotion alignment (Fu et al., 2021) |
| **Emotion-Related** | Emotion Generalization | VAW-GAN (Zhou et al., 2021b); Two-stage training (Chen et al., 2023b); Ranking-based SVM (Zhou et al., 2022a); Dual-granularity diffusion (Su et al., 2025) |
| | Intensity & Control | Continuous arousal scales (Prabhu et al., 2024); Two-stage training (Chen et al., 2023b); Expressive guidance (Chou et al., 2024); Seq2seq attention (Choi & Hahn, 2021); Activation steering (Xie et al., 2025) |
| **LLM-Related** | Duration Flexibility | Expressive guidance for enhanced emotional diffusion (Oh et al., 2024) |
| | Noisy Output | Emotion/intensity encoders (Zhang et al., 2025d); Prosody adjustment (Zhang et al., 2025d) |
| | Human Expectations | ECoT prompting (Cai & Li, 2024) |
| | Computational Overhead | Efficient model design (Mishra et al., 2023) |

need for additional discriminators.

### 6.2.2 Model-Related Challenges

Challenges associated with model structures include difficulties in adapting frameworks for emotional speech generation, defining effective loss functions for various components (Liu et al., 2020; Chou et al., 2024), preventing emotion leakage (Schnell et al., 2021), and overcoming speaker dependency (Zhou et al., 2020b; Choi & Hahn, 2021) and achieving disentangled emotion and speaker representations. Moreover, models must be capable of processing long sequences of data and accurately aligning each segment with its corresponding emotional expression (Fu et al., 2021). Although these challenges are significant, they are

generally considered less critical than those related to data.

Using minimum generation loss as the objective function in speech generation can constrain the learning process, often leading to over-smoothing of the generated speech parameters. In contrast, employing a more sophisticated loss function that incorporates perceptual aspects, diversity, and emotional qualities can enhance the emotional quality of speech generation, resulting in outputs that resonate more authentically with human listeners. (Liu et al., 2020) utilizes various losses, including adversarial, cycle consistency, and emotion classification, to effectively learn parameters in the training process. The adversarial loss measures how distinguishable the generated speech is from the true speech. Cycle consistency loss deposits that the input can keep its original form after passing through the two generators. To capture emotional aspects, this method employs an emotion classification loss. (Chou et al., 2024) uses a disentangled loss in the diffusion EVC model, which reduces the correlation between different speech representations, particularly emotion information and speaker identity. Additionally, an expressive guidance mechanism enhances emotional expressiveness throughout the reverse diffusion process. (Zhou et al., 2022c) leverages perceptual losses in the training process to enhance the intelligibility of the generated emotional speech. It uses a contrastive loss to ensure that the text and audio embeddings are similar, effectively disentangling linguistic and emotional elements. Additionally, by using pre-trained Speech Emotion Recognition (SER), this method predicts the emotion category of the generated speech and calculates the emotion classification loss at the utterance level.

Emotion leakage occurs when emotional intensity is weak during conversion, leading to a significant decline in signal quality and intelligibility. EmoCat (Schnell et al., 2021) addresses this issue by incorporating a gradient reversal block before the emotion classifier during training (Ganin & Lempitsky, 2015). This approach reverses the gradients during backpropagation to eliminate any input activation that could aid the emotion classifier, thereby reducing emotional leakage.

(ProEmo) (Zhang et al., 2025d) generates synthesized speech with refined emotional expressiveness and controlled intensity, which conventional TTS systems often lack. It extends the FastSpeech 2 framework (Ren et al., 2020) by incorporating an emotion encoder and an intensity encoder. The emotion encoder employs a fine-tuned HuBERT transformer (Hsu et al., 2021) with a classification head to derive robust emotion embeddings from the waveform, while the intensity encoder, also based on HuBERT and equipped with a regression head, estimates continuous emotion intensity through a speaker- and emotion-specific ranking function applied to acoustic features. During inference, GPT-4 (OpenAI, 2023) produces global and local scaling factors that modulate the FS2-predicted duration, energy, and pitch via multiplicative and additive adjustments, using a quadratic mapping function to yield speech with human-like expressiveness while preserving linguistic content.

There is a need to develop efficient emotional TTS systems that capture nuanced distinctions between emotions while generating expressive speech. Emo-DPO (Gao et al., 2024) combines a text encoder, an emotion-aware decoder, and a flow-matching model with a vocoder, enabling the synthesis of nuanced, controllable emotional speech with improved fidelity and expressiveness. The proposed Emo-DPO framework leverages Direct Preference Optimization (DPO) to refine emotional outputs by optimizing for preferred emotions. Integrating an LLM-based TTS architecture (LLM-TTS), Emo-DPO combines instruction tuning and DPO fine-tuning with Jensen-Shannon regularization to enhance emotional control. Leveraging emotional representation from various perspectives and levels is critical in developing TTS systems that disentangle various emotions. MsEmoTTS (Lei et al., 2022), a multi-scale framework, models emotions at global, utterance, and local levels. It employs a pre-trained BERT-based classifier for global emotion prediction, models intonation variations at the utterance level, and controls syllable-level strengths using a ranking-based method. Built on a Tacotron2 backbone with GMM-based attention, it unifies emotion transfer, prediction, and manual control, achieving superior performance in expressive speech synthesis.

Capturing the disentangled emotion and speaker representations in developing emotional speech generation is critical. Zhang, G. at (Zhang et al., 2023a) proposes iEmoTTS, a TTS system for cross-speaker emotion transfer that disentangles prosody from timbre. This approach allows robust emotion transfer even when the speaker and emotion features are entangled. iEmoTTS also supports zero-shot emotion transfer to unseen speakers through a timbre encoder with an information bottleneck mechanism, which retains only speaker-specific timbre features while excluding prosodic information. The system is trained end-to-end in a semi-supervised framework, reducing reliance on extensive labeled data. Additionally, it incorporates pre-trained models for style encoding and SER to reduce training data dependency. It introduces perceptual

loss functions to enhance emotion intelligibility and discrimination in converted speech.

According to independent emotion expression from the speaker style, (Zhou et al., 2020b) proposed a VAW-GAN-based encoder-decoder structure to learn the spectrum and prosody mapping in a speaker-independent manner. DurFlex-EVC(Oh et al., 2024) model uses de-stylize and stylize transformers, which separate the source style from input features and apply the target emotion style.

Cross-speaker emotion transfer in text-to-speech synthesis requires extracting speaker-independent emotion embeddings, yet existing timbre compression methods such as gradient reversal layers and vector quantization fail to fully separate speaker and emotion characteristics, causing speaker leakage and degraded synthesis quality. DiEmo-TTS (Cho et al., 2025b) addresses this through a self-supervised distillation approach that minimizes emotional information loss while preserving speaker identity. The method introduces cluster-driven sampling and information perturbation to retain emotional content while removing speaker-related factors, and proposes an emotion clustering and matching strategy using emotional attribute prediction and speaker embeddings to generalize to unlabeled data. A dual conditioning transformer integrates style features more effectively, and experimental results on the Emotional Speech Dataset confirm that the approach achieves state-of-the-art performance in learning speaker-irrelevant emotion embeddings, excelling in expressiveness, naturalness, and speaker identity preservation.

Current emotional text-to-speech and style transfer methods rely on reference encoders that compress the reference speech into a single global style or emotion vector, which fails to capture nuanced phoneme-level acoustic details and leaves timbre and emotion features entangled in the same representation. To overcome this limitation, (Yang et al., 2025) proposes a novel emotional text-to-speech method built on the FastSpeech 2 backbone that predicts fine-grained phoneme-level emotion embeddings while disentangling them from global timbre information. The architecture employs two parallel feature extractors within a dedicated style encoder, a global timbre extractor and a phoneme-aware emotion extractor that aligns reference acoustics with target phonemes via multi-head cross-attention. Mutual Information Neural Estimation explicitly minimizes the mutual information between the two representations, ensuring that the timbre embedding retains only speaker-specific information while the emotion embeddings capture prosodic nuance. Experimental results demonstrate that this combination of phoneme-level emotion modeling with principled feature disentanglement outperforms strong baselines in both naturalness and style similarity, producing well-separated emotion clusters and enabling more expressive and controllable emotional speech synthesis.

Emotional voice conversion faces persistent challenges due to the complexity of emotion features, which are deeply entangled with speaker identity and linguistic content characteristics, making it difficult to achieve high-quality any-to-any emotion conversion. DiffEmotionVC (Su et al., 2025) proposes a diffusion-based framework for any-to-any emotional voice conversion that integrates a dual-granularity emotion encoder capturing both utterance-level emotional context and frame-level acoustic details. The framework employs an orthogonality-constrained condition encoder that disentangles emotion features through gated cross-attention while preserving feature independence with an orthogonal loss. Additionally, multi-objective diffusion training enhances both reconstruction fidelity and emotion discriminability via contrastive learning. Experimental results demonstrate the effectiveness of the framework in maintaining speech quality while optimizing emotional expression.

### 6.2.3 Challenges Inherent to Emotions

The third category pertains to the intrinsic nature of emotions. Emotions are not merely discrete states; rather, they are highly dynamic (Prabhu et al., 2024; Zhou et al., 2021b; Chen et al., 2023b; Zhou et al., 2022a), varying in duration, intensity, valence, and strength (Prabhu et al., 2024; Chen et al., 2023b; Chou et al., 2024; Choi & Hahn, 2021; Oh et al., 2024). This variability introduces complexities that make it challenging to model and reproduce emotional expressions accurately. Among the three categories, this challenge ranks second in importance, underscoring its substantial impact on achieving human-like emotional speech generation.

EmoConv-Diff (Prabhu et al., 2024) uses conditioning on continuous arousal dimensions, allowing for effective control over emotional intensity. (Chou et al., 2024) utilized the generative power of diffusion models to tackle significant issues in previous deep learning approaches that use GANs and Autoencoders (AE), specifically concerning quality degradation and limited control over emotions. This method employs a

stochastic differential equation-based diffusion process to progressively transform speech features, enabling precise emotion transformation while preserving speaker identity and linguistic content. A disentangled loss distinguishes speaker and emotion representations, while an expressive guidance mechanism enhances emotional expressiveness during reverse diffusion. (Zhou et al., 2021b) employs a variational auto-encoding Wasserstein generative adversarial network (VAW-GAN) to transfer seen and unseen emotional style during training and run-time inference. Attention-based Interactive Disentangling Network (AINN)(Chen et al., 2023b) uses a two-stage training process to transfer emotional attributes, such as emotional strength and category, from a reference speech to a source speech while preserving the source's content. (Li et al., 2022a) addresses the challenges of speaker leakage, emotion strength control, and effective disentanglement of speaker and emotion features in cross-speaker emotion transfer for TTS. It proposes a modified Tacotron2-based framework incorporating an Emotion Disentangling Module (EDM), which uses emotion and speaker encoders with orthogonality constraints to ensure speaker-irrelevant and emotion-discriminative embeddings. Additionally, a scalar value is introduced to control emotion strength in synthetic speech, enabling flexible adjustments without dependency on manually labeled data. The challenges of mono-scale emotion modeling, limited flexibility in emotion transfer and prediction, and the lack of fine-grained emotion control in emotional speech synthesis are explored in (Lei et al., 2022). To address the challenge of intensity variation, Emovox (Zhou et al., 2022c), a Seq2Seq EVC framework, was developed. Emovox controls emotion intensity using relative attributes to capture fine-grained variations. Zhang, G. at (Zhang et al., 2023a) proposes iEmoTTS, which uses a probability-based method for emotion intensity control, enabling a nuanced generation of emotional speech with varying strengths. Although prior frameworks mark a significant advancement in the quality and versatility of EVC systems (Sisman et al., 2020), enabling applications like personalized speech synthesis and cross-lingual conversion, they process speech on a frame-by-frame basis, limiting their ability to modify speech duration and also intensity. The first issue was addressed in DurFlex-EVC (Oh et al., 2024). The DurFlex-EVC (Oh et al., 2024) model incorporates a style AE to disentangle emotional style from linguistic content. This is achieved using de-stylize and stylize transformers, which separate the source style from input features and apply the target emotion style. The unit aligner further compresses the features to unit-level representations and predicts durations, creating an efficient and context-aware framework for emotional style transformation.

Most existing text-to-speech systems offer only coarse and rigid emotion control, typically relying on discrete emotion labels or carefully crafted emotional text prompts, which makes fine-grained emotion manipulation either inaccessible or unstable, and these models require extensive high-quality datasets for training. EmoSteer-TTS (Xie et al., 2025) proposes a training-free approach to achieve fine-grained speech emotion control, including emotion conversion, interpolation, and erasure, through activation steering. The method builds on the empirical observation that modifying a subset of internal activations within a flow matching-based text-to-speech model can effectively alter the emotional tone of synthesized speech. It develops an efficient algorithm comprising activation extraction, emotional token searching, and inference-time steering that can be seamlessly integrated into a wide range of pretrained models. Extensive experiments demonstrate that this approach enables fine-grained, interpretable, and continuous control over speech emotion, outperforming the state of the art as the first method to achieve training-free and continuous fine-grained emotion control in text-to-speech synthesis.

Transferring a mix of primary emotions is an essential task in speech synthesis. Most existing methods focus on imitating a single emotion; however, to facilitate natural and engaging interactions between humans and agents, it is essential to incorporate mixed emotions into speech synthesis. Unfortunately, the development of such models is hindered by the lack of extensive multi-speaker corpora that contain mixed emotion labels (Kreibig & Gross, 2017). To tackle this issue, (Zhou et al., 2022a) uses a ranking-based SVM to model emotional styles as attributes reflecting the relevance of different emotions. This approach enables the system to quantify relationships between emotion pairs and synthesize new emotional mixtures by manually defining these attributes during conversion. Its architecture uses a seq2seq emotional voice conversion framework that integrates these attributes for mixed emotion synthesis. The architecture integrates a text encoder, an emotion encoder for embeddings, and a decoder with bidirectional LSTMs and attention mechanisms to generate natural and expressive emotional speech. Moreover, (Zhou et al., 2022d) uses a pre-trained model on massive speech corpora without emotional annotations, which is then fine-tuned using emotional speech data. It similarly uses a ranking function to determine the level of primary emotion according to variations between pairs of emotional speech samples. (Bott et al., 2024) deals with controllable emotional prosody

in TTS systems using natural language prompts. A FastSpeech 2-based architecture integrates prompt embeddings, derived from DistilRoBERTa, with speaker embeddings through a squeeze-and-excitation mechanism for accurate prosodic control. The system produces high-quality, emotionally expressive speech while preserving the speaker's identity by training with curriculum learning. Period VITS (Shirahata et al., 2023) focuses on the challenge of unstable pitch contours and artifacts in end-to-end emotional TTS systems caused by prosodic diversity. The proposed Period VITS integrates a periodicity generator for explicit pitch modeling, producing sample-level sinusoidal sources to enhance pitch stability and waveform quality. A frame pitch predictor within the prior encoder estimates frame-level prosodic features while normalizing flows augment prior distributions for richer acoustic variation. The HiFi-GAN-based decoder aligns pitch signals with latent acoustic features through down-sampling layers. The model is optimized end-to-end using variational inference with combined loss functions to ensure stability and expressiveness. The challenge of incorporating fine-grained intonation control, particularly questioning intonation, into emotional speech synthesis was considered in (Tang et al., 2023a). It claims existing TTS models can transfer emotions but struggle to model nuanced prosody like "angry question" versus "angry statement." The proposed QI-TTS builds on FastSpeech 2 and introduces a multi-style extractor to capture emotion at the sentence level and intonation at the final syllable level. By using relative attributes, it models intonation intensity in an unsupervised manner, enabling fine-grained control. A gradient reversal layer ensures content and style disentanglement to prevent interference.

Humans can experience roughly 34,000 distinct types of emotions. This includes eight basic emotions, along with secondary emotions that arise from combinations of these basics, as explained in the Theory of the Emotion Wheel (Plutchik, 2001). While secondary emotions are critical in social human interaction, synthesizing these emotions is often overlooked. EmoMix (Tang et al., 2023b) addresses the challenge of synthesizing emotional speech with controllable intensity and the ability to express mixed emotions, a significant limitation of current text-to-speech (TTS) systems. EmoMix overcomes this issue using a diffusion probabilistic model conditioned on emotion embeddings from a pre-trained SER model. It achieves flexible emotion control by blending predicted noise for different emotions during the sampling process and mixing neutral noise with the target emotion's noise. The architecture of EmoMix incorporates several components: GradTTS, a U-Net model with linear attention, the SER model, the HiFi-GAN vocoder, and a style reconstruction loss that ensures emotional consistency and naturalness in the synthesized speech.

### 6.2.4 LLM Challenges

The application of LLMs has shown great potential in regulating emotional expression in synthesized speech, particularly through prompt-based techniques (Guo et al., 2023; Sigurgeirsson & King, 2024). This approach enhances the expressiveness and naturalness of generated speech while preserving clarity and quality (Zhang et al., 2025d). While LLMs excel at producing diverse and contextually rich text, their output can sometimes be noisy and inconsistent when directly applied to emotional modifications in speech synthesis. In (ProEmo) (Zhang et al., 2025d), researchers noted that relying solely on LLM outputs for emotion control could compromise expressiveness due to output noise. They addressed this by integrating specific emotion and intensity encoders to guide prosody adjustment in systems like FastSpeech2. Moreover, approaches that modify backbone architectures (e.g., FastSpeech2) by adding emotion and intensity encoders have been explored to bridge the gap of effective integration with TTS systems. Such systems leverage the linguistic expressiveness of LLMs while fine-tuning acoustic outputs to convey emotion, although this remains an active area of research with room for improvement. Emotional expression is inherently subjective, and LLMs may generate outputs that do not align with human emotional expectations or ethical guidelines. (Li et al., 2024b) have proposed methods like the Emotional Chain-of-Thought (ECoT) prompting technique. This plug-and-play method guides LLMs through multiple reasoning steps to generate emotionally appropriate content, thereby improving human preference alignment in generated outputs.

Computational overhead and latency present significant challenges, particularly for real-time applications such as human-robot interaction. (Mishra et al., 2023) explored the real-time use of LLMs for tasks like emotion prediction in dialogue systems. The study indicated that efficient model design and careful engineering are crucial for the practical deployment of these technologies.

### 6.3 Comparative Synthesis of Emotional speech synthesis strategies

Emotional speech synthesis encompasses diverse methodological approaches, each offering distinct advantages depending on application requirements, data availability, and desired emotional control. This section provides actionable guidance for selecting appropriate methods based on key operational constraints and performance objectives.

**ESS vs. EVC:** Emotional Speech Synthesis (ESS/TTS) is optimal when generating speech from text with embedded emotional expression, making it ideal for applications such as virtual assistants, audiobook narration, and accessibility tools where linguistic content is available but prosodic emotional cues must be synthesized. ESS methods excel in controllability and linguistic coherence but require substantial paired text-emotion data. Emotional Voice Conversion (EVC), conversely, is preferable when transforming the emotional tone of existing speech recordings without altering linguistic content or speaker identity—suitable for post-production emotion editing, speech anonymization with emotion preservation, or creating emotional variations from a single neutral recording. EVC methods reduce data requirements by leveraging non-parallel approaches (CycleGAN, StarGAN, diffusion models) but face challenges in maintaining naturalness during extreme emotion transformations.

**Parallel vs. Non-Parallel EVC:** Parallel EVC methods, which require paired utterances (same content, same speaker, different emotions), achieve superior conversion quality and are recommended when high-fidelity emotion transformation is critical (e.g., professional voice acting, therapeutic applications). However, their dependence on expensive parallel datasets limits scalability. Non-parallel EVC approaches (CycleGAN-based, StarGANv2-VC, diffusion models like EmoConv-Diff) dominate in practical scenarios where parallel data is unavailable or prohibitively costly to collect. These methods trade minor quality degradation for dramatically improved data efficiency and are well-suited for multi-speaker, multi-emotion scenarios with limited per-emotion data.

**Categorical vs. Dimensional Emotion Models:** Categorical models (discrete emotion labels: happy, sad, angry, neutral) are preferable for applications requiring intuitive user control, alignment with existing emotion recognition systems, and straightforward evaluation (e.g., customer service bots, emotion-labeled speech datasets). They excel in interpretability but struggle with emotion intensity variation and blended states. Dimensional models (VAD-based) provide superior granularity for applications demanding fine control over emotional intensity, smooth transitions, or modeling of ambiguous/mixed emotions (e.g., expressive storytelling, mental health support systems). Methods like EmoConv-Diff (continuous arousal conditioning), ProEmo (intensity encoders), and EmoSphere++ (spherical emotion representation) demonstrate the efficacy of dimensional approaches. Hybrid strategies combining categorical labels with intensity scales (e.g., MsEmoTTS) offer balanced solutions for applications requiring both interpretability and expressiveness.

**GAN-based vs. Diffusion-based vs. LLM-guided Approaches:** GAN-based methods (CycleGAN, StarGAN) currently dominate non-parallel EVC due to mature architectures, fast inference, and proven effectiveness in many-to-many emotion mapping. They are recommended for real-time applications and scenarios requiring rapid prototyping. Diffusion models (EmoConv-Diff, DiffEmotionVC) offer superior generation quality, better handling of emotion intensity control, and reduced mode collapse risk, making them ideal for offline high-quality synthesis where inference latency is less critical (e.g., film production, content creation). LLM-guided approaches (ProEmo with GPT-4 scaling factors, ECoT prompting) excel in context-aware emotion prediction and natural language-driven emotion control but incur high computational costs and latency, limiting their use to offline or cloud-based applications where expressiveness and contextual appropriateness outweigh efficiency concerns.

**Speaker-Dependent vs. Speaker-Independent Models:** Speaker-dependent models achieve higher quality emotional speech for target speakers but require substantial speaker-specific data and separate models per speaker. They are recommended for single-speaker applications (e.g., personalized virtual assistants, celebrity voice synthesis). Speaker-independent or few-shot approaches (VAW-GAN, DiEmo-TTS, EmoSphere++ zero-shot synthesis) enable emotion control across diverse speakers with minimal per-speaker data, making them essential for scalable multi-speaker systems (e.g., multi-lingual assistants, user-customizable voices). Cross-speaker emotion transfer methods (iEmoTTS, DiEmo-TTS with self-supervised distillation) represent the current state-of-the-art for balancing quality and flexibility.

**Practical Deployment Considerations:** For *real-time interactive systems* (virtual agents, gaming), prioritize GAN-based non-parallel EVC or lightweight categorical ESS models with efficient neural vocoders. For *high-quality content creation* (audiobooks, film), diffusion-based models with dimensional emotion control provide optimal results. For *cross-lingual or low-resource scenarios*, leverage emotion transfer learning (EmoCat) or multilingual pre-trained models. For *dynamic emotional narratives*, employ models with temporal emotion modeling (MsEmoTTS utterance-level control, mixed-emotion synthesis). For *user-facing applications requiring transparency*, categorical models with interpretable emotion labels and intensity sliders balance usability with expressiveness.

In summary, method selection should prioritize: (1) *data constraints*—non-parallel methods for limited data, parallel for quality-critical applications; (2) *emotion model*—categorical for interpretability, dimensional for granularity; (3) *generative framework*—GANs for speed, diffusion for quality, LLMs for context-awareness; (4) *speaker requirements*—dependent for single-speaker quality, independent for scalability; and (5) *latency tolerance*—real-time methods for interaction, offline for content creation.

# 7 Emotional Face Synthesis

Emotional facial expressions are fundamental nonverbal signals that enable accurate assessment of internal states in psychiatric applications and bolster the integration of machine learning in mental-health diagnostics (Coda-Forno et al., 2023). In therapeutic contexts, these cues guide clinicians in recognizing and addressing patient emotions (Iftikhar et al., 2024), while in everyday social exchanges, their contextual interpretation is critical for effective communication and rapport building (Han et al., 2024). Beyond human-to-human interaction, facial expressions drive dynamic user engagement across interactive systems, enhancing empathetic response generation in dialogue agents (Wang et al., 2025b) and underpinning social intelligence in conversational AIs (Chen et al., 2024). The breadth of these roles underscores the centrality of facial affect in advancing emotion research and human-machine interaction (Chen & Moscholios, 2024). Emotional face synthesis builds on this foundation by enabling virtual avatars, social robots, and conversational agents to exhibit realistic emotional expressions, highlighting its transformative potential in interactive applications (Tan et al., 2024). By integrating multimodal cues—visual, auditory, and textual—the emotional authenticity and contextual sensitivity of AI systems are greatly enhanced, which is critical for education, therapy, and collaborative AI (Zhan et al., 2022). Moreover, precise lip synchronization combined with coordinated non-verbal signals (facial expressions, gaze, head pose) yields seamless, engaging animations for film, gaming, and HCI (Liang & Lu, 2024; Ma et al., 2024; Wang et al., 2024). These advancements not only meet ethical and equitable AI standards but also continue to drive innovation across virtual/augmented reality and teleconferencing environments. Here, we briefly outline three core approaches for generating realistic talking faces from audio inputs, static images, and their integration. We then analyze the primary challenges in this field and evaluate the various methods researchers have developed to overcome them. We highlight diverse application domains—virtual avatars, social robotics, sentiment analysis, and HCI—where advanced emotional synthesis drives deeper engagement and improved user experience (Pandey & Vishwakarma, 2024; Saunders & Namboodiri, 2023; Xu et al., 2024a; Pataranutaporn et al., 2021), and discuss ethical considerations such as bias mitigation and responsible deployment of facial manipulation technologies.

## 7.1 Approaches

Emotional face synthesis methods are significantly shaped by the underlying emotion representation model employed. **Categorical emotion models**, which classify emotions into discrete categories (e.g., happiness, sadness, anger, fear, surprise, disgust), have traditionally dominated facial expression research due to their alignment with well-established psychological theories (e.g., Ekman's basic emotions) and the availability of labeled datasets (e.g., CK+, FER2013). These models facilitate straightforward training objectives, typically using cross-entropy loss for classification-based synthesis or class-conditional generation in GANs. However, categorical approaches struggle to represent the continuous spectrum of human emotions, subtle variations in expression intensity, and blended emotional states (e.g., a mixture of happiness and surprise). **Dimensional emotion models**, which represent emotions in continuous spaces such as valence-arousal or valence-arousal-dominance (VAD), offer greater flexibility for modeling nuanced expressions and smooth

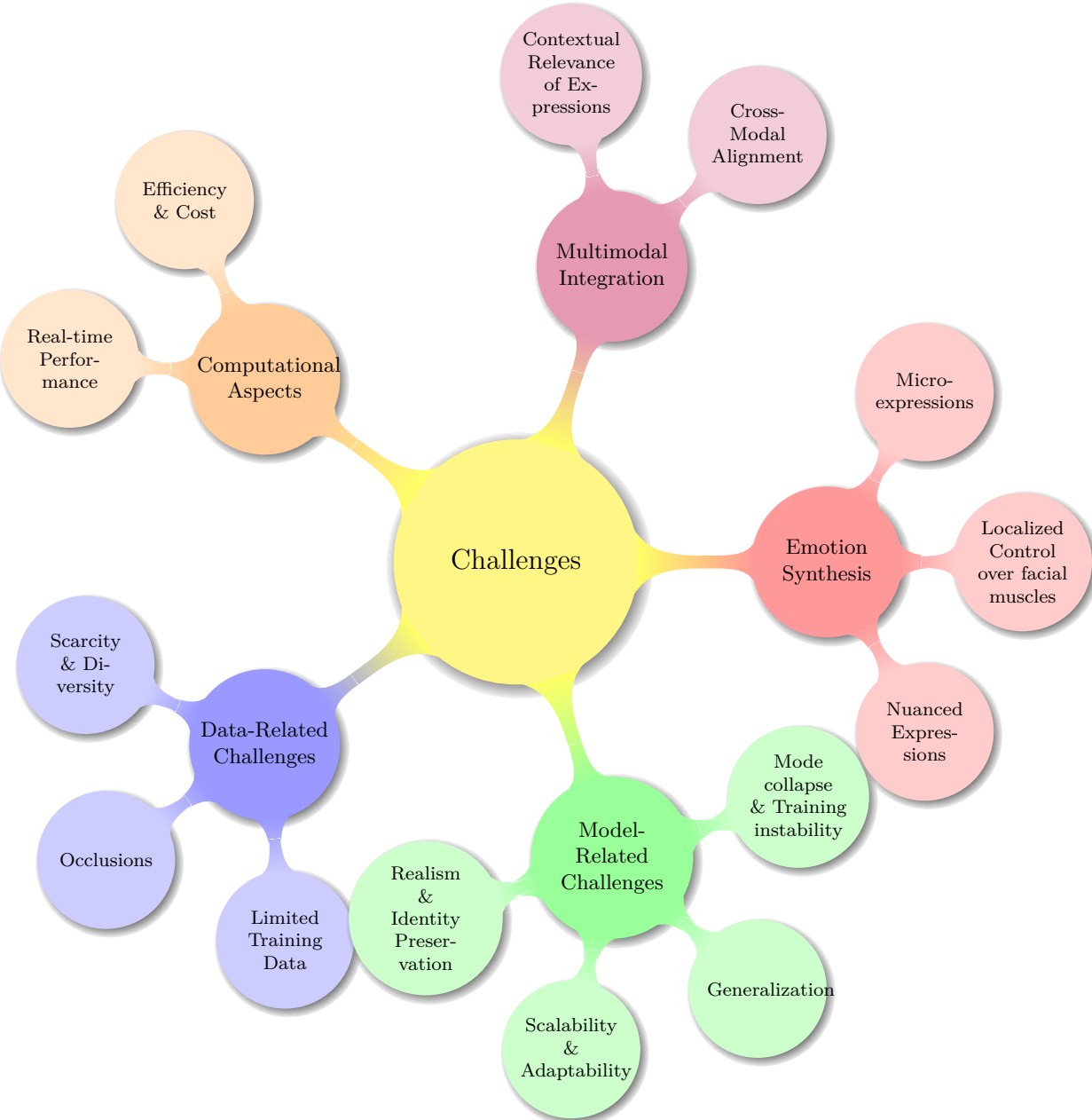

Figure 12: Challenges in emotional face synthesis

emotional transitions. These models enable fine-grained control over facial animation parameters, allowing synthesis systems to generate expressions with varying degrees of intensity and to interpolate between emotional states. Dimensional approaches often employ regression-based training objectives and can leverage Action Unit (AU) representations from the Facial Action Coding System (FACS), which decompose facial expressions into anatomically based muscle movements. However, dimensional models introduce challenges in evaluation (how to assess "correctness" in continuous space) and user interpretability (VAD coordinates are less intuitive than categorical labels). Recent hybrid approaches attempt to combine the strengths of both paradigms, using categorical labels for high-level emotion guidance while incorporating dimensional or AU-based controls for intensity and fine-grained manipulation. The choice of emotion model directly influences dataset annotation requirements, network architecture (classification vs. regression heads), loss

function design (cross-entropy vs. MSE/MAE), evaluation metrics (accuracy vs. correlation coefficients), and the interpretability of generated expressions.

***Photo-realistic facial animation:*** Photo-realistic facial animation techniques are advanced computational methods designed to generate digital facial animations that closely resemble real human expressions and movements in terms of both visual detail and emotional subtlety. The emotion model employed fundamentally shapes these techniques: categorical models typically generate distinct facial configurations for each emotion class, while dimensional models enable continuous modulation of facial features along emotional dimensions, producing smoother transitions and more naturalistic expressions. These techniques, such as the Warp-Guided GANs introduced by Geng et al. (Geng et al., 2018), advance visual fidelity and emotional nuance, significantly enhancing the realism of animated characters.

***Audio-driven synthesis:*** Audio-driven synthesis systems, exemplified by EDTalk (Tan et al., 2024), integrate auditory and visual inputs to create interactions imbued with emotional depth and realism. The integration of emotion models in these systems varies: categorical approaches may map discrete emotion labels (inferred from audio or text) to corresponding facial expression templates, while dimensional approaches can extract continuous arousal and valence estimates from prosodic features (pitch, energy, speaking rate) and map them to facial animation parameters in real-time. Unlike photo-realistic facial animation techniques, which prioritize high visual fidelity to achieve lifelike faces, these methods do not require a highly realistic facial appearance. Instead, they focus on synchronizing audio and visual cues to produce expressive and engaging facial animations that enhance the quality of virtual interactions.

***Multi-modal synthesis and editing:*** Multi-modal synthesis and editing techniques, encompassing GAN-based, autoregressive, diffusion, and NeRF approaches, enhance emotional authenticity and contextual awareness in facial animations (Zhan et al., 2022). The emotion model choice significantly impacts multimodal fusion strategies: categorical models may use emotion labels as discrete conditioning signals across modalities, while dimensional models can align continuous emotional representations (e.g., VAD vectors) extracted from text, audio, and visual inputs, enabling more coherent cross-modal synthesis and smoother emotion control.

## 7.2 Challenges

Emotional facial generation faces several critical challenges, including challenges related to data, models, emotions, multi-modal integration, and computational inefficiencies. Addressing these issues is essential for creating inclusive, robust systems capable of accurately representing the full spectrum of human emotions. These insights provide a foundation for advancing the field and overcoming current limitations. Figure 12 classifies these challenges into five major domains: data, learning models, emotion synthesis, multimodal, and computational issues. Table 6 summarizes these categories, outlining their respective sub-issues and highlighting studies that address these challenges. Next, we will discuss the challenges and the methods proposed to overcome them.

### 7.2.1 Data-Related Challenges

Dataset bias significantly impacts the generalization capability of affective face generation models, particularly in representing diverse demographics, emotional expressions, and cultural nuances. Limited training data diversity further restricts the generalizability of models to diverse emotional states (Siddiqui, 2022). Biased datasets often fail to capture the complexity of human emotions, leading to inequities in model performance and reduced inclusivity (Washington et al., 2021). This issue is compounded by the difficulty of achieving localized control over facial muscle movements and generating nuanced expressions (Varanka et al., 2024). The collection and annotation of emotional facial expression data pose unique challenges that are critical for training robust face synthesis systems. Data collection methodologies vary widely, ranging from laboratory-based posed expressions to spontaneous expressions captured in naturalistic settings. Posed expression datasets, while providing clear examples of prototypical emotions, often lack the spontaneity and subtle variations characteristic of genuine emotional displays. In contrast, spontaneous or "in-the-wild" datasets offer greater ecological validity but introduce significant annotation difficulties. Facial expressions in real-world contexts are often fleeting, ambiguous, and accompanied by occlusions (e.g., hands, hair, ac-

Table 6: Summary of challenges and solutions in emotional face synthesis

| Challenge | Sub-challenge | Solutions |
|---|---|---|
| **Data-Related** | Scarcity & Diversity | Promoting diversity in datasets (Xu et al., 2024b); Enhanced annotation for expressions (Chen et al., 2018) |
| | Limited Training Data | Synthetic data generation (Xu et al., 2024c); StyleGAN-generated data (Akamatsu et al., 2025); EmoVOCA dataset combination (Nocentini et al., 2025) |
| | Occlusions | Robust model design (Geng et al., 2018); Specialized training data (Xu et al., 2024c) |
| **Model-Related** | Realism & Identity | Advanced GANs (StyleGAN) (Jiang et al., 2023); Diffusion Models (Yin et al., 2022); Identity-emotion disentanglement (Tan et al., 2025); Instruction-driven 3D generation (Vo et al., 2025) |
| | Generalization | Robust architectures (Xu et al., 2024c); Domain adaptation (Geng et al., 2018); Long-range autoregressive diffusion (Zhang et al., 2025b) |
| | Scalability | Unified frameworks (UniPortrait) (He et al., 2024); Motion modeling (MotionGAN) (Otberdout et al., 2022); Temporal layers (Xu et al., 2024d); Sparse landmark methods (Tu et al., 2024) |
| | Training Stability | Improved GAN architectures (Jiang et al., 2023); Regularization techniques (Yin et al., 2022); Alternative loss functions (Jiang et al., 2023) |
| **Emotion Synthesis** | Nuanced Expressions | Fine-grained control (Action Units) (Yin et al., 2022); Continuous modeling (GC-GANs) (Qiao et al., 2018); Subtle cue analysis (Retsinas et al., 2024); Emotion-audio spatial attention (Ma et al., 2026); Continuous valence-arousal conditioning (Cha et al., 2025); FLAME-guided diffusion (Zhang et al., 2024c); Semantic expression parameters (Shen et al., 2025) |
| | Micro-expressions | High-res temporal modeling (Xu et al., 2024d) |
| | Localized Control | Advanced facial modeling (Retsinas et al., 2024); Anatomical models (Varanka et al., 2024) |
| **Multi-Modal** | Feature Alignment | Collaborative diffusion (Huang et al., 2023); Self-supervised learning (Pham et al., 2017); Audio-visual integration (C3D-DBN) (Nguyen et al., 2017); Text-to-expression synthesis (Cheng et al., 2024); Multi-modal emotion embedding (Yee et al., 2025) |
| | Contextual Relevance | Textual data integration (Cheng et al., 2024); Sentiment analysis (VECTN) (Pandey & Vishwakarma, 2024); Cross-reconstruction disentanglement (Liu et al., 2025) |
| **Computational** | Efficiency | Hybrid models (GANs+Diffusion) (Xu et al., 2024d); Compact representations (He et al., 2024); Efficient alignment (Liu et al., 2018); Model quantization/pruning (Liu et al., 2018); Knowledge distillation (He et al., 2024); Hardware acceleration (He et al., 2024) |

cessories) or variations in lighting, pose, and camera angle. Annotating such data requires expert knowledge to distinguish between genuine emotions and social displays (e.g., polite smiles), to identify blended or transitional emotions, and to account for individual differences in expressiveness. The choice of annotation framework also presents challenges: categorical approaches (e.g., labeling expressions as happy, sad, angry) may oversimplify the continuous nature of emotional expressions, while dimensional models (e.g., valence-arousal) or Action Unit-based coding systems (FACS) demand specialized training and are time-intensive. Furthermore, achieving high inter-annotator reliability is particularly challenging for subtle expressions, micro-expressions, and expressions from underrepresented demographic groups, where cultural display rules and individual variability can lead to divergent interpretations. These annotation inconsistencies propagate through the training pipeline, resulting in models that may perform well on controlled datasets but fail to generalize to diverse, real-world scenarios. Moreover, existing benchmarks inadequately capture the fluidity and diversity of emotional expressions, limiting the generalization capabilities of models in real-world scenarios (Xu et al., 2024a). Current methods relying on detailed face modeling encounter challenges in unconstrained environments with variations in pose, lighting, and expression (Wiles et al., 2018). The need for comprehensive annotation of facial attributes further restricts the manipulation of identity and attributes, impeding the development of robust models (Bao et al., 2018). Multi-modal datasets also struggle with aligning features across modalities, resulting in limited high-resolution, contextually relevant outputs (Zhan et al., 2022). The lack of diversity within and across identities in datasets exacerbates these issues, underscoring the importance of inclusive data collection practices (Xu et al., 2024b). ComFace (Akamatsu et al., 2025) is introduced to address the challenge of capturing intra-personal facial changes for health and emotion monitoring, hindered by the scarcity of temporally varying real-world face images. Its novel representation learning approach leverages StyleGAN-generated synthetic data to simultaneously learn both inter-personal facial differences and intra-personal facial changes within individuals. Remarkably, their method trained exclusively on synthetic data achieves comparable or superior performance to state-of-the-art approaches trained on real images across multiple facial change estimation tasks. Many approaches reconstructed 3D faces from 2D videos using parametric models (3DMMs), but these lacked precision for accurate lip-syncing. The authors solve this with EmoVOCA (Nocentini et al., 2025), a data-driven framework that combines VO-CAset (neutral 3D talking heads with speech) and Florence4D (expressive 3D faces without speech) using a Double Encoder/Shared Decoder architecture, where separate encoders learn speech and expression features while a shared decoder combines them to create synthetic emotional 3D talking heads that preserve both accurate lip movements and convincing expressions. Privacy concerns intersect with dataset bias, as existing methods often compromise either privacy or the quality of expression recognition (Chen et al., 2018). Ethical implications arise from biased datasets, which hinder fairness and generalization across diverse populations (Stahl et al., 2023). Addressing these concerns requires promoting diversity in datasets while maintaining identity consistency, as well as enhancing annotation processes to improve the representation of nuanced and localized expressions (Xu et al., 2024b; Varanka et al., 2024). These efforts are critical for fostering fairness, equity, and trust in affective face generation technologies.

### 7.2.2 Model-Related Challenges

Three primary sub-challenges reside in model-related challenges: preserving identity, ensuring the realism of the model's outputs, and achieving generalization to real-world scenarios. The choice of emotion model significantly influences how these challenges are addressed. **Categorical emotion models** simplify the training and evaluation process by framing facial expression generation as a discrete classification or class-conditional generation problem, which facilitates the use of established metrics (e.g., emotion classification accuracy). However, they may struggle to preserve identity across diverse emotional expressions, as discrete emotion-specific generators or discriminators can inadvertently encode identity information into emotion representations. **Dimensional emotion models** enable more natural preservation of identity by decoupling continuous emotional attributes (valence, arousal) from identity features through disentangled representation learning. However, they introduce complexity in ensuring realism, as continuous emotion spaces may generate "in-between" expressions that are anatomically plausible but emotionally ambiguous or unnatural. Hybrid approaches that combine categorical emotion labels with dimensional intensity controls or Action Unit-based representations offer a promising direction for balancing identity preservation, realism, and emotional expressiveness.

Advanced generative methodologies for generating emotionally expressive facial animations, such as GANs and diffusion models, could address these issues, and these frameworks have significantly improved the quality of affective facial generation, enabling applications in human-computer interaction, virtual avatars, and sentiment analysis. GANs have been instrumental in advancing affective face generation, producing high-quality animations that capture nuanced emotional states. Methods like EDTalk (Tan et al., 2024) enhance realism by disentangling facial features such as mouth shape, head pose, and emotional expression. Similarly, techniques integrating driving videos with single-image inputs, as demonstrated in (Averbuch-Elor et al., 2017), expand the versatility of GANs in synthesizing dynamic animations. Despite the greatness of GANs, challenges like generalization to real-world faces persist (Xu et al., 2024c). Diffusion models provide a robust alternative to GANs, leveraging iterative noise refinement to generate diverse emotional attributes while maintaining identity consistency. Conditional diffusion frameworks such as ID3 excel at preserving intra-class identity and generating emotionally coherent expressions (Wang et al., 2023). Innovations like Stable Animator address temporal stability, ensuring smooth transitions in dynamic animations (Tu et al., 2024). These models also bridge textual input and emotional synthesis, as demonstrated by frameworks like ContinuousText-to-Expression Generator (CTEG) and Globally-informed Gaussian Avatar (GiGA), which produce nuanced 3D avatars (Xu et al., 2024a). These models excel at capturing subtle emotional cues and offer advantages in emotional synthesis (Zhan et al., 2022). Innovations like READ Avatars, Emo3D (Xu et al., 2024a) generation, and UniPortrait (He et al., 2024) leverage adversarial loss, autoregressive models, and unified frameworks to enable applications in hyper-realistic avatars, empathetic human-computer interaction, and nuanced sentiment analysis (Saunders & Namboodiri, 2023; Pataranutaporn et al., 2021). Hybrid approaches combining GANs and diffusion models aim to leverage the strengths of both techniques. For instance, UniPortrait (He et al., 2024) employs a dual-module architecture to enhance identity preservation and adaptability, balancing the speed of GANs with the nuanced output of diffusion models. Additionally, frameworks utilizing Action Units for fine-grained expression control demonstrate the potential for bridging the gap between realism and expressiveness in affective face synthesis (Yin et al., 2022). By integrating the strengths of GANs and diffusion models, researchers are advancing the synthesis of nuanced, contextually appropriate facial expressions. These models enhance realism and emotional dynamics, enabling applications in virtual avatars, human-computer interaction, and personalized content creation (Pumarola et al., 2018; Bouzid & Ballihi, 2022; Siddiqui, 2022; Huang & Khan, 2017).

Existing diffusion-based talking head generation methods struggle to produce emotionally expressive portraits while preserving speaker identity, due to insufficient utilization of audio's inherent emotional cues, identity leakage in emotion representations, and isolated learning of emotion correlations. DICE-Talk (Tan et al., 2025) addresses these limitations through a framework that disentangles identity from emotion and then cooperates emotions with similar characteristics. It develops a disentangled emotion embedder that jointly models audio-visual emotional cues through cross-modal attention, representing emotions as identity-agnostic Gaussian distributions. A correlation-enhanced emotion conditioning module with learnable Emotion Banks explicitly captures inter-emotion relationships through vector quantization and attention-based feature aggregation, while an emotion discrimination objective enforces affective consistency during the diffusion process through latent-space classification. Experiments on the MEAD and HDTF datasets demonstrate superiority in emotion accuracy while maintaining competitive lip synchronization performance. Audio-driven portrait animation methods typically emphasize lip synchronization and short-range visual fidelity in constrained speaking scenarios, but fail to capture nuanced, dynamically evolving emotions that flow coherently with the rhythm and content of speech over long temporal contexts. X-Actor (Zhang et al., 2025b) presents an audio-driven portrait animation framework that generates lifelike, emotionally expressive talking head videos from a single reference image and an input audio clip, enabling actor-quality long-form portrait performance. Central to the approach is a two-stage decoupled generation pipeline: an audio-conditioned autoregressive diffusion model predicts expressive yet identity-agnostic facial motion latent tokens within a long temporal context window, followed by a diffusion-based video synthesis module that translates these motions into high-fidelity video animations. By operating in a compact facial motion latent space decoupled from visual and identity cues, the autoregressive diffusion model captures long-range correlations between audio and facial dynamics through a diffusion-forcing training paradigm, enabling infinite-length emotionally rich motion prediction without error accumulation. Generating 3D facial expressions from natural language instructions remains challenging because most existing methods rely on discrete emotion labels or predefined expression categories, which cannot capture the richness and specificity of textual descriptions for both static expres-

sions and dynamic expression transitions. (Vo et al., 2025) proposes an instruction-driven approach for 3D facial expression generation and transition that takes text instructions as input and produces corresponding 3D facial animations. The method leverages a language model to interpret free-form textual descriptions and maps them to parametric 3D face model expression parameters, enabling both the generation of target expressions and smooth transitions between emotional states. This text-based control paradigm offers greater flexibility and user accessibility compared to conventional label-driven or reference-driven approaches, broadening the applicability of 3D facial animation in interactive systems and content creation.

### 7.2.3 Multi-modal Integration Challenges

Multi-modal integration is pivotal for generating emotionally authentic facial expressions. By combining visual, auditory, and textual cues, multi-modal frameworks align generated expressions with intended emotional states, ensuring contextual relevance. Frameworks like EmotiveTalk (Wang et al., 2024) and Emotion-LLaMA (Cheng et al., 2024) use audio decoupling and self-supervised learning to align speech, lip movements, and expressions, producing realistic talking-head videos (Liang & Lu, 2024; Zhan et al., 2022). Collaborative Diffusion (Huang et al., 2023) exemplifies this by integrating pre-trained uni-modal diffusion models, enhancing emotional synthesis through modality synergy. EDTalk (Tan et al., 2024) exemplifies the importance of synthesizing multiple modalities for lifelike outputs. Audio-visual integration remains central to multi-modal systems; methods like C3D-DBN (Nguyen et al., 2017) align auditory cues (e.g., tone, pitch) with visual signals for coherent emotional outputs. Emotion-LLaMA further incorporates textual data, enabling contextually appropriate expression synthesis (Cheng et al., 2024). Techniques for compact multi-modal representations, like (Liu et al., 2018) improve real-time processing and scalability, making these systems suitable for virtual avatars and conversational AI. Multi-modal frameworks like VECTN integrate textual data for sentiment analysis, aligning emotional synthesis with contextual information and capturing complex states like sarcasm (Pandey & Vishwakarma, 2024). By synthesizing facial expressions aligned with multi-modal cues, these systems enhance user engagement and emotional resonance in AI-driven interactions (Zhan et al., 2022). Most existing emotion-aware talking face generation methods rely on a single modality, either audio or image, for emotion embedding, which limits their ability to capture nuanced affective cues, and conditioning on a single reference image restricts the representation of dynamic changes in actions or attributes across time. SynchroRaMa (Yee et al., 2025) introduces a framework that integrates a multi-modal emotion embedding by combining emotional signals from text via sentiment analysis and audio via speech-based emotion recognition and audio-derived valence-arousal features, enabling the generation of talking face videos with richer and more authentic emotional expressiveness. To ensure natural head motion and accurate lip synchronization, the framework includes an audio-to-motion module that generates motion frames aligned with the input audio. Additionally, scene descriptions generated by a large language model serve as additional textual input, capturing dynamic actions and high-level semantic attributes that enhance temporal consistency and visual realism. Audio-driven emotional 3D facial animation typically relies on static and predefined emotion labels, which limits the diversity and naturalness of generated expressions and prevents fine-grained dynamic emotional control. MEDTalk (Liu et al., 2025) proposes a framework for fine-grained and dynamic emotional talking head generation that first disentangles content and emotion embedding spaces from motion sequences using a carefully designed cross-reconstruction process, enabling independent control over lip movements and facial expressions. Beyond conventional audio-driven lip synchronization, the method integrates audio and speech text to predict frame-wise intensity variations and dynamically adjust static emotion features for realistic emotional expressions. Furthermore, multimodal inputs including text descriptions and reference expression images guide the generation of user-specified facial expressions, and the generated results are compatible with MetaHuman for integration into industrial production pipelines.

### 7.2.4 Computational Challenges

These systems often face computational inefficiencies and scalability challenges, particularly in synthesizing realistic facial expressions under diverse conditions. Advanced generative models, such as GANs and diffusion models, require substantial computational resources for training and inference, limiting their applicability in resource-constrained environments (Zhan et al., 2022). Slow inference speeds and extensive training data requirements further hinder real-time applications and large-scale deployments. Existing

methods, such as Warp-Guided GANs, struggle with generalizing to unseen data or scenarios with significant occlusions, highlighting the limitations of current frameworks (Geng et al., 2018). Additionally, techniques that assume a neutral face as a starting point or rely on simplified representations restrict adaptability to dynamic environments (Averbuch-Elor et al., 2017). Identity blending and the need for extensive fine-tuning also pose challenges, as frameworks like UniPortrait demonstrate the resource-intensive nature of achieving identity preservation and adaptability (He et al., 2024). The inability to effectively model dynamic and contextually adaptive facial animations further limits the realism of generated outputs. Many methods fail to capture intricate temporal dynamics, such as micro-expressions and head movements, which are critical for producing emotionally resonant animations (Otberdout et al., 2020; Xu et al., 2024c). Multi-modal frameworks introduce additional complexity during training and inference, requiring significant resources to align features and synthesize coherent outputs (Zhan et al., 2022). Efforts to address these challenges include optimizing algorithms and architectures to reduce resource consumption while maintaining high-quality synthesis. Addressing computational constraints and improving multi-modal integration will enhance the robustness and accessibility of affective face generation systems, facilitating their application in diverse domains (Liu et al., 2018; Zhan et al., 2022).

### 7.2.5 Emotion Synthesis Challenges

The challenges in synthesizing nuanced, temporally coherent emotional expressions are profoundly influenced by the underlying emotion model. **Categorical emotion models** face inherent difficulties in generating smooth emotional transitions and blended expressions, as discrete emotion boundaries do not naturally support interpolation between states. This leads to abrupt or unnatural transitions when animating dynamic sequences that involve emotional change (e.g., shifting from happiness to sadness). Techniques such as emotion-specific motion modeling or per-class temporal modules are often employed, but they struggle with generalization to unseen emotion combinations. **Dimensional emotion models**, conversely, naturally facilitate smooth temporal transitions by treating emotions as continuous trajectories in valence-arousal or VAD space. This enables the generation of fluid emotion dynamics and supports the modeling of micro-expressions (brief, subtle emotional signals) as small perturbations in the continuous emotion space. However, dimensional models introduce challenges in ensuring that generated expressions remain interpretable and correspond to recognizable emotional states, particularly for boundary regions in the emotion space. Hybrid approaches that combine categorical emotion labels with continuous intensity or AU-based controls offer enhanced capability for modeling both discrete emotional states and smooth transitions, addressing limitations of purely categorical or dimensional approaches.

Temporal consistency is a cornerstone of dynamic facial animation, preventing abrupt transitions that undermine emotional impact and is critical for video synthesis, ensuring smooth transitions and coherent animations (Zhan et al., 2022; Tan et al., 2024). Techniques like those in (Tan et al., 2024) disentangle temporal dynamics from other facial attributes, while methods such as driving-video-based animation ensure smooth transitions (Averbuch-Elor et al., 2017). Techniques like MotionGAN model expression transitions on a hypersphere, minimizing motion artifacts and enhancing temporal fidelity (Otberdout et al., 2022). Sparse landmark-based methods leverage anatomical priors for coherent facial deformations with reduced computational complexity (Xu et al., 2024d). Disentangling temporal dynamics from attributes like expression and pose further enhances coherence. Frameworks integrating temporal layers mitigate interference with spatial priors, ensuring stable animations (Averbuch-Elor et al., 2017; Tu et al., 2024; Xu et al., 2024d). Multi-modal cues, such as audio and textual signals, align facial dynamics with emotional intent, as seen in EmotiveTalk, which synchronizes lip movements and expressions with emotional audio cues (Pham et al., 2017; Liang & Lu, 2024; Wang et al., 2024). Innovations like Takin-ADA address challenges like expression leakage, while Action Unit-based techniques provide anatomically accurate representations of expressions. These advancements improve dynamic animations, broadening their applicability in teleconferencing, virtual reality, and digital media (Vougioukas et al., 2020) (Pumarola et al., 2018) (Lin et al., 2024). Techniques such as SMIRK (Retsinas et al., 2024) and Auxiliary Classifier GANs (Siddiqui, 2022) aim to address complex emotional nuances by enhancing the quality and diversity of generated expressions, yet challenges persist in capturing them. Occlusions often lead to assumptions of exaggerated expressions, compromising the fidelity of nuanced emotional synthesis (Retsinas et al., 2024). Temporal modeling is critical for generating contin-

uous and smooth facial expression videos, yet existing approaches often fail to simulate intricate transitions and micro-expressions effectively (Otberdout et al., 2020; Xu et al., 2024c). Discrete methods for facial expression generation also cannot capture the continuity of emotional transitions, though advancements like Geometry Contrastive GANs (GC-GANs) demonstrate the potential for high-fidelity continuous expressions (Qiao et al., 2018). Interactive generative adversarial networks (iGANs) and temporal behavioral biometrics illustrate the importance of incorporating temporal consistency and additional facial behaviors, such as head pose, to enhance emotional authenticity (Nojavanasghari et al., 2018; Agarwal et al., 2020). To advance nuanced emotional synthesis, researchers must focus on robust temporal modeling, effective handling of occlusions, and seamless multimodal integration. Innovations such as GC-GANs and advanced frameworks for continuous, contextually adaptive expression generation are pivotal for achieving emotionally intelligent systems. These advancements are essential for applications requiring high levels of emotional realism and human-like interaction capabilities, driving progress in affective face generation. While some studies have addressed the generation of facial videos driven by emotional audio, efficiently generating high-quality talking head videos that integrate both emotional expressions and style features remains a significant challenge, as most current audio-driven facial animation research primarily focuses on generating videos with neutral emotions. ESGaussianFace (Ma et al., 2026) proposes a framework for emotional and stylized audio-driven facial animation that leverages 3D Gaussian Splatting to reconstruct 3D scenes and render videos, ensuring efficient generation of 3D consistent results. The method introduces an emotion-audio-guided spatial attention mechanism that effectively integrates emotion features with audio content features, enabling more accurate reconstruction of facial details across different emotional states. Two 3D Gaussian deformation predictors achieve emotional and stylized deformations of the Gaussian points through emotion and style features, and a multi-stage training strategy enables step-by-step learning of the character's lip movements, emotional variations, and style features. 3D Gaussian splatting-based talking head synthesis has gained attention for its ability to render high-fidelity images with real-time inference speed, but since it is typically trained on only a short video that lacks diversity in facial emotions, the resultant talking heads struggle to represent a wide range of emotions. EmoTalkingGaussian (Cha et al., 2025) addresses this by proposing a lip-aligned emotional face generator that trains a 3D Gaussian splatting model capable of manipulating facial emotions conditioned on continuous emotion values, specifically valence and arousal, while retaining synchronization of lip movements with input audio. To achieve accurate lip synchronization for in-the-wild audio, the method introduces a self-supervised learning approach that leverages a text-to-speech network and a visual-audio synchronization network. Experiments on publicly available videos demonstrate improvements over existing methods in image quality, emotion expression accuracy, and lip synchronization. While many existing approaches to audio-driven portrait animation focus on lip synchronization and video quality, few tackle the challenge of generating emotion-driven talking head videos with fine-grained control over both emotion categories and intensities. EMOdiffhead (Zhang et al., 2024c) proposes a method for emotional talking head video generation that enables fine-grained control of emotion categories and intensities while supporting one-shot generation. Given the linearity of the FLAME 3D model in expression modeling, the method extracts expression vectors using the DECA approach and combines them with audio to guide a diffusion model in generating videos with precise lip synchronization and rich emotional expressiveness. This approach enables learning rich facial information from emotion-irrelevant data while facilitating the generation of emotional videos, effectively overcoming the limitations of emotional data such as the lack of diversity in facial and background information. Generating emotion-specific talking head videos from audio input is a complex challenge because emotion is a highly abstract concept with ambiguous boundaries, necessitating disentangled expression parameters to produce emotionally expressive results. EmoHead (Shen et al., 2025) presents a method to synthesize talking head videos via semantic expression parameters. An audio-expression module that can be specified by an emotion tag predicts expression parameters for arbitrary audio input, enhancing the correlation from audio input across various emotions. The method leverages a pre-trained hyperplane to refine facial movements by probing along the vertical direction, and the refined expression parameters regularize neural radiance fields to facilitate emotion-consistent generation of talking head videos. Experimental results demonstrate that semantic expression parameters lead to improved reconstruction quality and controllability.

### 7.3 Comparative Synthesis of Emotional Face synthesis strategies

The landscape of emotional face synthesis encompasses diverse methodological paradigms, each optimized for specific application requirements, computational constraints, and quality-realism trade-offs. This section synthesizes key decision criteria to guide practitioners in selecting appropriate approaches based on operational priorities.

**Photo-realistic vs. Audio-driven vs. Multi-modal Synthesis:** Photo-realistic facial animation methods (Warp-Guided GANs, StyleGAN-based approaches) prioritize visual fidelity and are optimal for applications demanding high-quality static or short animated sequences with detailed texture and lighting (e.g., film production, high-end gaming, virtual influencers). However, they typically require extensive per-identity training data and struggle with real-time generation. Audio-driven synthesis methods (EDTalk, EmotiveTalk, VASA-1) excel in generating synchronized talking-head videos from speech input, making them ideal for conversational AI, virtual assistants, and teleconferencing where lip-sync accuracy and emotional expressiveness are paramount. These methods trade some visual fidelity for temporal coherence and computational efficiency. Multi-modal synthesis approaches (UniPortrait, Collaborative Diffusion, Synchro-RaMa) integrate audio, text, and visual cues to achieve superior contextual appropriateness and emotional authenticity, recommended for applications requiring nuanced emotion control and cross-modal consistency (e.g., empathetic dialogue systems, therapeutic applications, educational content).

**GAN-based vs. Diffusion-based vs. NeRF-based vs. 3D Gaussian Splatting:** GAN-based methods (StyleGAN, EDTalk, MotionGAN) currently dominate real-time applications due to fast inference speeds and mature training procedures, though they suffer from mode collapse and training instability. They are recommended for production systems requiring consistent quality and low latency. Diffusion-based models (ID3, Stable Animator, EMOdiffhead, DICE-Talk) provide superior generation quality, better emotion intensity control, and improved training stability, making them ideal for offline high-quality synthesis and applications where generation diversity is critical. However, their iterative sampling process incurs significant latency, limiting real-time deployment. NeRF-based approaches (EmoHead) offer exceptional 3D consistency and view synthesis capabilities, optimal for 3D avatar applications and scenarios requiring novel view generation, but demand substantial computational resources and per-scene training. 3D Gaussian Splatting methods (EmoTalkingGaussian, ESGaussianFace) represent the current frontier, combining NeRF-quality rendering with near-real-time inference, recommended for next-generation interactive systems where both quality and speed are non-negotiable.

**Categorical vs. Dimensional vs. Action Unit-based Emotion Control:** Categorical emotion models align with user intuition and existing annotated datasets (CK+, FER2013, MEAD), making them suitable for user-facing applications requiring interpretable emotion labels. However, they struggle with emotion intensity variation and blended states. Dimensional models (VAD, valence-arousal) enable fine-grained intensity control and smooth emotional transitions, optimal for dynamic emotional narratives, expressive storytelling, and applications requiring continuous emotion modulation (EmoSphere++, EmoTalkingGaussian, DiffEmotionVC for face). Action Unit-based approaches (FACS-based control) provide the most precise anatomical control over facial muscles, enabling synthesis of subtle expressions, micro-expressions, and culturally diverse emotional displays. They are recommended for research applications, facial animation authoring tools, and scenarios demanding maximal expressiveness control, though at the cost of increased user complexity. Hybrid approaches combining categorical labels with dimensional or AU-based refinement (ID3 with intensity control, EMOdiffhead) offer practical compromises.

**Single-shot vs. Few-shot vs. Extensive Training Paradigms:** One-shot methods (VASA-1, EMOdiffhead) that generate animations from a single reference image excel in scenarios where per-identity data is limited or real-time personalization is required (e.g., user-uploaded photos, privacy-sensitive applications). Few-shot approaches (UniPortrait, EmoSphere++ zero-shot) balance quality and data efficiency, suitable for multi-user systems with limited per-user data collection. Extensive training methods (subject-specific NeRFs, personalized StyleGAN fine-tuning) achieve maximum quality for single identities and are recommended for celebrity avatars, brand mascots, or applications where investment in per-identity data collection is justified.

**2D vs. 3D Synthesis:** 2D synthesis methods (most GAN and diffusion approaches) offer computational efficiency and compatibility with existing 2D image/video pipelines, suitable for applications not requiring 3D consistency (social media filters, 2D animation, video editing). 3D synthesis approaches (NeRF, 3D Gaussian Splatting, FLAME-based models) provide view consistency and enable novel view synthesis, essential for VR/AR applications, 3D gaming, metaverse avatars, and scenarios requiring head pose variation or multi-view rendering. The choice depends critically on whether downstream applications require 3D geometry.

**Emotion Disentanglement and Identity Preservation:** For applications requiring robust identity preservation across diverse emotional expressions (virtual assistants, personalized avatars), prioritize methods with explicit disentanglement mechanisms (DICE-Talk, DiEmo-TTS adapted for face, iEmoTTS principles applied to face synthesis, EmoSphere++ with orthogonality loss). Methods without explicit disentanglement risk identity leakage into emotion representations, causing inconsistent appearance across emotions.

**Temporal Consistency and Dynamic Synthesis:** For video synthesis requiring smooth emotional transitions and temporal coherence (animated storytelling, conversational agents), prioritize methods with dedicated temporal modeling (MotionGAN hypersphere transitions, Stable Animator temporal layers, X-Actor long-range autoregressive diffusion, FIRES flexible thinking). Static or frame-independent methods produce temporally inconsistent results unsuitable for video applications.

**Practical Deployment Recommendations:** For *real-time interactive systems* (video calls, live streaming, gaming), deploy 3D Gaussian Splatting or optimized GAN-based methods with hardware acceleration. For *high-quality content creation* (film, advertising), use diffusion models or NeRF with extensive per-scene optimization. For *privacy-sensitive applications*, prioritize on-device one-shot methods over cloud-based extensive training approaches. For *cross-cultural applications*, employ AU-based or dimensional models with cultural adaptation rather than categorical Western emotion labels. For *accessibility tools* (emotion communication aids), prioritize interpretable categorical models with clear emotion labels. For *research and authoring*, use AU-based methods providing maximal expressiveness control.

In summary, selection criteria should prioritize: (1) *quality vs. speed*—GANs for speed, diffusion for quality, 3D Gaussian Splatting for both; (2) *emotion model*—categorical for interpretability, dimensional for granularity, AU for precision; (3) *data availability*—one-shot for limited data, extensive training for quality; (4) *dimensionality*—2D for efficiency, 3D for consistency; (5) *temporal requirements*—dedicated temporal models for video, frame-independent for static; and (6) *identity preservation*—explicit disentanglement for robust multi-emotion synthesis.

# 8 Discussion and Future Prospects

The rapid progress in affective computing highlights both remarkable achievements and unresolved challenges. To move closer to emotionally intelligent agents, future research must address key gaps in emotion understanding, affective cognition, and expression. This section outlines promising research directions, emphasizing the need for robust datasets, interpretable models, multimodal integration, and ethical frameworks to ensure trustworthy and human-centered development.

## 8.1 Future Prospects in Emotion Understanding

Despite significant advances, emotion recognition still faces persistent challenges that hinder practical deployment: dependence on limited or biased datasets (Thakur & Gupta, 2026), difficulty with complex, overlapping, or culturally nuanced emotions, and suboptimal multimodal fusion in noisy real-world environments (Nandini et al., 2025). Rather than reiterate these well-known limitations, we highlight six concrete research directions, each motivated by a specific gap in the methods surveyed above.

*Causal and counterfactual modeling.* Most surveyed models primarily learn statistical correlations and are therefore vulnerable to spurious cues and hidden biases. Future research should investigate causal and counterfactual learning frameworks that disentangle genuine emotional signals from contextual confounders,

enabling more robust, interpretable, and reliable emotion recognition (Yang et al., 2024a; Singhal et al., 2025).

***Continual learning under non-stationary affect.*** Emotional states evolve over time, across contexts, and between cultures, whereas most existing emotion recognition models are trained under static settings and remain vulnerable to catastrophic forgetting. Future research should develop continual learning frameworks tailored to affective computing, integrating replay, knowledge transfer, and adaptive regularization strategies to enable robust long-term adaptation without sacrificing previously acquired knowledge (Jain et al., 2024; Laurensi et al., 2024; Aslam et al., 2025).

***Privacy-preserving and on-device affective computing.*** Since emotion data is inherently biometric and highly sensitive, future research should develop privacy-preserving affective computing frameworks that combine federated learning, differential privacy, and on-device inference. Such approaches can reduce raw-data transmission while improving privacy protection and supporting learning from heterogeneous affective data across distributed users (Gahlan & Sethia, 2025; Tawfik et al., 2026).

***Self-supervised and parameter-efficient adaptation.*** Future research should reduce the reliance on large-scale labeled emotion datasets by leveraging self-supervised multimodal pretraining to learn transferable affective representations from unlabeled data (Zhang et al., 2025c; Ahuja & Sethia, 2025). In parallel, parameter-efficient fine-tuning techniques, such as low-rank adaptation (LoRA), should be further explored to enable scalable and resource-efficient adaptation of foundation models across diverse affective domains (Lashkarashvili et al., 2024; Chen et al., 2025; Fang et al., 2025).

***Retrieval-augmented grounding of LLMs.*** Future research should investigate retrieval-augmented generation (RAG) frameworks that ground emotional reasoning in external evidence to mitigate hallucinations and improve contextual understanding in LLM-based affective systems (Wognum et al., 2024; Fan et al., 2024; Lei et al., 2024). Emerging benchmarks further highlight the limitations of general-purpose LLMs in affective reasoning, underscoring the need for grounded and evidence-aware emotion understanding (Sabour et al., 2024; Yang et al., 2024b).

***Interpretability and evaluation beyond accuracy.*** Future research should improve the transparency of emotion recognition models by developing faithful and human-interpretable explanation methods beyond conventional "black-box" deep learning approaches (Di Luzio et al., 2025; Wang et al., 2026).

## 8.2 Future Prospects in Affective Cognition

Emotional intelligence has been studied from psychological, neuroscientific, and technological perspectives, including its integration into emotion-aware, human-centered AI and related ethics (Espinosa Gámez, 2026). Advancing human-centered AI requires integrating emotion prediction and elicitation within complex interactive scenarios (Zhang et al., 2024b). We highlight five concrete directions that translate this goal into open problems.

***Appraisal-grounded process models over moving averages.*** Treating emotion as a time-averaged state may obscure its temporal dynamics; appraisal- and reinforcement-learning models that simulate emotion as a continuous appraised process can predict and explain affect during interaction (Zhang et al., 2024b), and nonlinear dynamical models outperform sliding-window regression while remaining interpretable within appraisal theory (Barradas et al., 2025).

***Anticipatory emotion prediction in HCI.*** Moving beyond reactive recognition, dialogue systems can forecast future affective reactions from the system's current behavior before the user responds (Li et al., 2023c); extending such anticipatory prediction to multimodal, long-horizon settings is an open problem.

***LLM agents with cognitive architectures.*** LLMs currently contribute mainly to emotion understanding and generation, and emotional elicitation remains limited (Zall & Kangavari, 2022); genuine emotional

intelligence requires combining LLMs with cognitive frameworks that let memory, goals, and appraisal shape decisions and affect, as in LLM-driven social robots (Kang et al., 2024), affective BDI agents (Pico et al., 2024), spanning perception, cognition, and expression jointly (Zhao et al., 2024).

***Mechanistic, appraisal-grounded interpretability.*** Whether LLMs process emotion through psychologically plausible mechanisms, mapped onto cognitive appraisal theory, is an emerging, safety-relevant question (Tak et al., 2025) that should be tied to functional explainability rather than post-hoc rationalization. sectionDiscussion and Future Prospects The rapid progress in affective computing has established foundational capabilities in emotion understanding, affective cognition, and expression synthesis. However, achieving truly intelligent agents with human-like emotional intelligence requires addressing critical open problems that extend beyond current methodological limitations. This section articulates prioritized research directions that represent fundamental gaps in knowledge and capability, followed by a consolidated discussion of ethical considerations essential for responsible development and deployment of emotionally intelligent systems.

### 8.3 Future Prospects in Emotional Expression Synthesis

### 8.3.1 Cross-Cultural Emotional Expression Synthesis

Current emotional synthesis systems (text, speech, face) predominantly generate expressions aligned with Western cultural norms and fail to adapt to diverse cultural contexts where emotional expression conventions differ substantially. Recent evidence challenges the universality hypothesis, demonstrating that appropriate emotional expressions vary systematically across cultures in intensity, modality preference, timing, and display rules. A synthesized "happy" facial expression, empathetic speech prosody, or comforting text that resonates in one culture may appear unnatural, exaggerated, or inappropriate in another.

***Concrete Research Contributions Needed for Synthesis:*** *(1) Culture-Conditioned Expression Generation Architectures:* Develop synthesis models with explicit cultural conditioning controls. For emotional face synthesis, implement disentangled generators separating identity, base emotion, and cultural display modulation—enabling the same emotion (e.g., sadness) to be rendered with culture-appropriate intensity and expression patterns. For emotional speech synthesis, design prosody generators that adapt emotional vocal cues (pitch contours, intensity, speaking rate) based on cultural norms. For emotional text synthesis, develop style transfer models that preserve emotional intent while adapting linguistic expression patterns to cultural communication styles (direct vs. indirect, elaborate vs. succinct). *(2) Cross-Cultural Transfer Learning for Low-Resource Synthesis:* Most cultures lack large-scale emotional synthesis datasets. Design meta-learning frameworks enabling few-shot adaptation of synthesis models to new cultural contexts using limited data. Employ techniques like MAML or prototypical networks pre-trained on diverse cultural corpora, then fine-tuned with minimal culture-specific samples. Critical innovation: culturally-stratified meta-tasks enabling rapid generalization to unseen cultural expression patterns. *(3) Culturally-Diverse Synthesis Benchmarks with Perceptual Validation:* Establish standardized evaluation datasets covering 10+ cultural regions with parallel emotional content (same emotional scenarios expressed according to different cultural norms). Include human perceptual studies with both in-culture and cross-culture raters to validate that synthesized expressions are perceived as natural and emotionally appropriate within their target cultural context, not just technically accurate according to Western-trained models. *(4) Context-Aware Synthesis with Cultural Pragmatics:* Develop synthesis systems that jointly model situational context (formal/informal, public/private, hierarchical relationships) and cultural background to generate pragmatically appropriate expressions. For instance, synthesizing consolation speech that appropriately balances emotional expressiveness with cultural norms for emotional restraint in formal contexts.

### 8.3.2 Unified Multimodal Emotional Expression Synthesis

Current emotional synthesis systems treat text, speech, and facial animation generation as independent tasks with separate models, training pipelines, and control mechanisms. However, human emotional expression is inherently multimodal and synchronized—facial expressions, vocal prosody, and linguistic content must align temporally and semantically to convey authentic emotions. Fragmented synthesis pipelines produce

outputs where modalities may contradict (e.g., sad words with happy prosody, angry speech with neutral face), undermining perceived authenticity and emotional impact.

***Concrete Research Contributions Needed for Multimodal Synthesis:*** *(1) Joint Text-Speech-Face Synthesis Architectures:* Design end-to-end models that generate synchronized emotional expressions across all three modalities from a unified emotional intent specification. Employ shared multimodal encoders representing abstract emotional states, then branch into modality-specific decoders (text generation, prosody/waveform synthesis, facial animation) with cross-modal attention ensuring temporal and semantic alignment. Critical innovation: bidirectional constraints where generated text influences prosody targets, prosody timing constrains facial animation keyframes, and facial expression intensity modulates linguistic emotional vocabulary. *(2) Controllable Multi-Level Emotion Specification:* Move beyond single emotion labels toward hierarchical control enabling specification of: (a) base emotional state (categorical or dimensional), (b) intensity/arousal level, (c) expression modality emphasis (facially-dominant vs. vocally-dominant expression), and (d) temporal dynamics (onset, apex, offset timing). Implement this through hierarchical conditioning in diffusion models or structured latent spaces in VAEs, allowing users to precisely control how emotions manifest across modalities. *(3) Cross-Modal Consistency Learning and Evaluation:* Develop training objectives that explicitly enforce cross-modal emotional consistency beyond simple conditioning. Employ contrastive learning where correctly aligned multimodal emotional expressions are pulled together in embedding space while misaligned combinations (happy text + sad speech) are pushed apart. Design novel evaluation metrics quantifying cross-modal emotional alignment, moving beyond independent per-modality quality assessment toward holistic multimodal coherence metrics validated through human perceptual studies. *(4) Causal Emotion-to-Expression Generation:* Rather than directly mapping emotion labels to expressions, model the causal chain: emotional event → appraisal → physiological response → multimodal expression. This enables synthesis systems to generate expressions that reflect not just emotion categories but their underlying causes and intensity trajectories, producing more naturalistic temporal dynamics (gradual onset, maintained apex, decay) rather than static or abrupt emotional transitions.

### 8.3.3 Controllable and Interpretable Emotional Synthesis

Current emotional synthesis models function as black boxes where users specify high-level emotional targets (e.g., "generate angry speech") but lack fine-grained control over how emotions manifest or understanding of why specific expressions were generated. This opacity limits creative control for content creators, prevents error diagnosis when synthesis fails, and raises concerns about unintended emotional manipulations in generated content. Recent research has highlighted challenges in interpretability and control across text, speech, and facial synthesis, with LLMs introducing additional risks of unpredictable emotional outputs (Wognum et al., 2024). Additional barriers, such as model interpretability, the lack of unified evaluation metrics, and limited cross-domain generalizability, require sustained research attention. Deep learning models, while highly accurate, frequently operate as "black boxes," making it difficult to justify or explain their predictions in sensitive applications. This fundamental limitation underscores the need for new interpretability frameworks in emotion recognition (Di Luzio et al., 2025). To address the lack of explainability in existing multimodal aspect-based sentiment analysis methods, (Wang et al., 2026) reformulates this as a generative task using multimodal large language models, enabling joint aspect-level sentiment prediction and natural language explanation generation. The proposed framework incorporates dependency-syntax-guided sentiment cues to enhance aspect-oriented reasoning and improve the faithfulness of generated explanations. (Alharbi, 2024) demonstrated that employing explainable feature selection methods in virtual reality environments not only increases user trust but also facilitates clinical adoption, though effectively communicating technical insights to end-users remains challenging. With the emergence of LLMs and foundation models, the landscape of affective computing is rapidly evolving, offering new opportunities such as zero-shot and few-shot emotion recognition. Nonetheless, these models introduce critical risks, including hallucinated or spurious emotional attributions, difficulties in context understanding, and high annotation costs (Wognum et al., 2024). Future research should prioritize the creation of large-scale, diverse, and high-quality datasets that capture real-world variability across individuals and cultures. Further efforts are needed to develop robust and explainable emotion recognition models, along with standardized evaluation metrics tailored to the complexities of multimodal affective computing. Finally, realizing the full potential of LLMs and foundation models in this domain will require systematic solutions for mitigating hallucinations, improving

context awareness, and reducing labeling costs—ensuring that advances in hybrid affective systems translate to trustworthy and generalizable real-world applications.

### 8.3.4 Efficient Real-Time Emotional Synthesis for Interactive Applications

Current state-of-the-art emotional synthesis models (large diffusion models for faces, neural vocoders for speech, LLMs for text) require substantial computational resources and incur significant latency, making real-time interactive applications impractical. Conversational agents, live virtual avatars, gaming characters, and assistive technologies demand synthesis that responds within human perceptual thresholds (typically <100ms for speech, <50ms for facial animation) while maintaining emotional expressiveness and quality.

***Concrete Research Contributions Needed for Efficient Synthesis:*** *(1) Fast Emotional Synthesis Architectures:* Develop synthesis models optimized for low-latency generation. For emotional speech: design streaming neural vocoders that generate audio incrementally rather than requiring full-sequence processing, enabling <50ms first-audio latency. For emotional faces: employ GAN-based or flow-based models offering single-pass generation faster than iterative diffusion sampling, or develop few-step diffusion samplers preserving quality. For emotional text: optimize transformer decoding with speculative sampling, cached key-value attention, or mixture-of-depths architectures reducing computational cost per token. *(2) Quality-Latency Trade-off Management:* Implement adaptive synthesis systems that dynamically adjust model complexity based on latency constraints and context importance. Use cascaded architectures where fast, lightweight models handle routine synthesis while complex models engage for emotionally critical moments. Develop anytime synthesis algorithms that produce progressively refined outputs, allowing early termination when latency budgets are exceeded while still delivering acceptable quality. *(3) Model Compression for Emotional Synthesis:* Apply knowledge distillation to transfer capabilities from large teacher synthesis models (e.g., GPT-based text generation, large diffusion models for faces) to compact student models suitable for real-time deployment. Develop synthesis-specific compression techniques: for speech, distill prosody modeling separately from acoustic generation; for faces, compress temporal consistency modules while preserving expression diversity; for text, distill emotional language models into smaller variants retaining emotional expressiveness. *(4) Hardware-Accelerated Synthesis:* Design synthesis architectures optimized for deployment on GPUs, mobile NPUs, or dedicated AI accelerators. Leverage hardware-specific optimizations: INT8/INT4 quantization for reduced memory bandwidth, fused kernels for common synthesis operations (attention, convolution, upsampling), and model architectures matching hardware parallelism patterns. Particularly focus on energy-efficient synthesis for battery-constrained devices (smartphones, AR glasses, wearable robots).

### 8.3.5 LLM-Guided Emotional Text and Multimodal Synthesis

Large language models have transformed text generation capabilities, but their application to emotionally expressive synthesis remains underdeveloped. Current LLM-based systems often generate emotionally flat or inappropriate text, lack mechanisms for coordinating text generation with emotional speech and facial synthesis, and struggle to maintain emotional consistency across extended interactions.

***Concrete Research Contributions Needed for LLM-Based Emotional Synthesis:*** *(1) Emotion-Conditioned LLM Fine-Tuning for Expressive Text Generation:* Fine-tune LLMs with explicit emotion conditioning signals enabling controlled generation of emotionally expressive text. Develop training objectives that jointly optimize for linguistic quality, emotional appropriateness, and stylistic consistency. Employ reinforcement learning from human feedback (RLHF) specifically targeting emotional expressiveness, where human raters assess whether generated text conveys intended emotions naturally and contextually. Create emotion-annotated instruction datasets covering diverse emotional scenarios (consolation, celebration, persuasion, conflict de-escalation) for supervised fine-tuning. *(2) LLM-Guided Multimodal Synthesis Orchestration:* Design architectures where LLMs serve as high-level controllers orchestrating emotional text, speech, and facial synthesis. The LLM generates not only linguistic content but also structured specifications for prosody (pitch contours, speaking rate, pauses), facial expression timing (onset, apex, offset), and cross-modal synchronization. Implement this through LLM output parsing into synthesis control parameters or through end-to-end training where LLM embeddings directly condition speech and face generators. *(3) Emotionally Consistent Long-Form Synthesis:* Develop mechanisms enabling LLMs to maintain emotional coherence across extended generation (long conversations, narratives, multi-turn interactions). Implement

explicit emotional state tracking where the LLM maintains a running representation of current emotional context, updating it as conversation progresses. Design attention mechanisms allowing the LLM to reference earlier emotional expressions when generating subsequent outputs, ensuring consistency (e.g., maintaining sympathetic tone throughout a consolation dialogue). *(4) Controllable Emotional Style Transfer:* Leverage LLMs for emotional style transfer where neutral text is transformed into emotionally expressive variants while preserving semantic content. Develop prompt-based or fine-tuning approaches enabling control over target emotion, intensity, and stylistic features (formal vs. colloquial, elaborate vs. concise). Extend this to multimodal scenarios where LLM-transformed text drives corresponding speech prosody and facial expression modifications.

### 8.3.6 Standardized Evaluation for Emotional Synthesis Quality and Authenticity

The emotional synthesis field lacks standardized evaluation frameworks, relying heavily on subjective human studies that are expensive, time-consuming, and difficult to reproduce. Different studies employ incompatible metrics (perceptual ratings, classification accuracy, physiological responses), diverse evaluation protocols (paired comparison, Likert scales, forced choice), and inconsistent human subject pools, making it nearly impossible to meaningfully compare synthesis methods or track field-wide progress.

***Concrete Research Contributions Needed for Synthesis Evaluation:*** *(1) Multi-Dimensional Synthesis Quality Metrics:* Develop comprehensive evaluation frameworks assessing: (a) technical quality (audio fidelity, visual realism, linguistic fluency), (b) emotional accuracy (does output convey intended emotion?), (c) emotional naturalness (are expressions humanlike vs. exaggerated/artificial?), (d) cross-modal consistency (do facial, vocal, textual expressions align?), and (e) contextual appropriateness (does expression fit situational context?). Design automated proxy metrics correlating with human perceptual judgments, validated through large-scale human studies. Establish standardized reporting requirements including all dimensions rather than cherry-picking favorable metrics. *(2) Perceptual Benchmarks with Diverse Rater Pools:* Create standardized human evaluation protocols for emotional synthesis using demographically and culturally diverse rater populations. Develop online platforms enabling large-scale perceptual studies where synthesized outputs are rated on emotional authenticity, naturalness, and appropriateness. Critically, include both forced-choice tasks (comparing two synthesis methods) and Likert-scale ratings (absolute quality assessment), as these capture different aspects of perceptual quality. Publish benchmark datasets with pre-collected human ratings enabling reproducible comparison without requiring each study to run new human evaluations. *(3) Automatic Synthesis Quality Assessment Models:* Train learned metrics predicting human perceptual judgments of emotional synthesis quality. For speech, develop models trained to predict mean opinion scores (MOS) for emotional expressiveness and naturalness. For faces, train perceptual quality networks assessing emotional authenticity beyond pixel-level reconstruction metrics. For text, develop language models fine-tuned to assess emotional appropriateness and linguistic naturalness. These learned metrics enable rapid iteration during model development while correlating with expensive human evaluations. *(4) Fairness and Robustness Evaluation for Synthesis:* Establish protocols assessing synthesis system fairness across demographics and robustness to input variations. Evaluate whether synthesis quality degrades for underrepresented identities (ethnic minorities, gender diversity, age groups, accents). Test robustness to noisy or ambiguous emotional specifications, out-of-distribution requests, and adversarial inputs attempting to produce inappropriate outputs. Develop standardized test suites probing specific failure modes: emotional ambiguity, cross-cultural expressions, micro-expressions, blended emotions.

### 8.3.7 Ethical and Societal Challenges

Affective computing is inherently dual-use: the same techniques that enable more natural, supportive, and personalized interactions can also be used for surveillance, manipulation, discrimination, and other forms of misuse (Stark & Hoey, 2021). As affective systems increasingly integrate emotion understanding, affective reasoning, and emotional expression generation, ethical considerations become a fundamental requirement rather than an optional design consideration. This subsection summarizes the major cross-cutting ethical and societal challenges that span the entire affective computing pipeline.

***Privacy, consent, and affective surveillance.*** Emotion understanding relies on highly sensitive affective signals, including facial expressions, vocal prosody, physiological measurements, behavioral cues, and textual

content (Cowie, 2015). These signals can reveal intimate information about users' mental states, vulnerabilities, and social relationships, often beyond what individuals intentionally disclose (Iren et al., 2023). The collection and processing of such information without meaningful informed consent or transparent disclosure create risks of affective surveillance, particularly in workplaces, educational environments, healthcare, and public spaces (McStay, 2020). Furthermore, anonymization alone is often insufficient because facial images, voice recordings, and behavioral patterns remain vulnerable to re-identification. Responsible deployment therefore requires transparent data governance, explicit consent mechanisms, clear limitations on secondary data use, and privacy-preserving learning approaches such as federated learning, differential privacy, and on-device inference (Wu, 2024).

***Bias, fairness, and cultural sensitivity.*** Emotion recognition and affective reasoning models are commonly trained on datasets that inadequately represent demographic, cultural, and neurodiverse populations. Annotation protocols frequently encode culturally specific assumptions regarding emotional expression, causing systems to systematically misinterpret users from different social or cultural backgrounds (Stark & Hoey, 2021). Recent studies further challenge the universality of emotional expressions, demonstrating substantial cross-cultural variation in both emotion production and perception (Katirai, 2024). Consequently, systems developed primarily from Western datasets may perform unfairly in healthcare, education, recruitment, or security applications. Addressing these issues requires representative datasets, culturally aware annotation strategies, subgroup bias auditing, and evaluation protocols that explicitly measure robustness across demographic and cultural populations (Ballesteros et al., 2024; Cowie, 2015).

***Manipulation, emotional exploitation, and dual-use risks.*** The ability not only to recognize but also to synthesize emotions introduces significant risks of emotional manipulation. Emotion-aware systems can enable highly personalized persuasive content, targeted advertising, or behavioral influence based on users' inferred emotional states. Likewise, advances in expressive text, speech, and facial synthesis increase the potential for emotionally convincing deepfakes, misinformation, fraud, and psychological manipulation (Katirai, 2024). The same technologies that support valuable applications such as mental-health assistance, accessibility, and adaptive education may therefore also be repurposed for surveillance, coercion, or discrimination, highlighting the inherently dual-use nature of affective computing (Wu, 2024).

***Transparency, user autonomy, and accountability.*** Responsible affective AI requires users to understand when emotional information is being collected, how it is analyzed, and how resulting inferences influence system decisions. Beyond informed consent, users should be able to opt out of emotion recognition, review or contest incorrect emotion predictions, and retain meaningful control over affective interactions. These requirements are particularly important for vulnerable populations and high-stakes domains, including healthcare, education, and employment (Alharbi, 2024). At the system level, accountability demands regular bias auditing, deployment impact assessments, continuous post-deployment monitoring, and clear mechanisms for redress when harms occur. Multi-stakeholder governance involving researchers, policymakers, ethicists, domain experts, and affected communities is essential to ensure that affective computing technologies remain trustworthy, equitable, and socially beneficial (Ballesteros et al., 2024).

Overall, the responsible advancement of affective computing depends on integrating ethical principles throughout the entire system lifecycle, from data collection and model development to deployment and governance. Privacy protection, fairness, transparency, user autonomy, and accountability should therefore be treated as core design objectives alongside technical performance, enabling affective AI systems that are both effective and socially trustworthy.

## 9    Conclusion

This study has provided a comprehensive exploration of the integration of emotional intelligence into intelligent agents, highlighting the critical roles of emotion understanding, affective cognition, and emotion expression in fostering naturalistic and empathetic human-computer interactions. By systematically analyzing the challenges, ranging from dataset limitations and model interpretability in emotion understanding to contextual and cognitive complexities in affective cognition and multimodal synchronization in emotion expression, we have underscored the multifaceted hurdles that impede progress in affective computing. We outline and explore recent solutions for revealing promising pathways to overcome these

obstacles. This work not only synthesizes current advancements but also proposes a roadmap for future research, paving the way for the development of emotionally intelligent agents capable of truly adaptive, empathetic, and human-like interactions. Our findings underscore the necessity of continued innovation in data collection, model design, and evaluation frameworks to overcome existing barriers and pave the way for emotionally intelligent agents that can foster trust, empathy, and more effective human-computer interaction.

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
