# OpenReview forum: "Intelligent Agents with Emotional Intelligence: Current Trends, Challenges, and Future Prospects"
_TMLR — Under review for TMLR_

### Review · Reviewer_rsZT · 2026-06-04

**Summary Of Contributions:**

This is a survey paper focusing on "emotional intelligence" and discussing the broader field and what has been done here. The paper does a good job of going through what has been done in the field, laying out the existing work, highlighting the limitation of existing work and proposing some ideas on how to overcome the limitations.

In terms of strengths as this is a summary paper it does do a good job of covering a wide literature and going through it and highlighting limitations but in terms of weakness this could have been presented in a much better manner both in terms of verbosity as well in terms of how the material is presented as long paragraphs. Additionally, the paper is light in terms of how to address the weakness beyond stating the obvious.

Since this is a summary paper some of this is expected but there is enough space for this to become a better paper in terms of summarizing the broader field and presenting the information better.

**Audience:**

Yes

**Audience Explanation:**

As I mentioned earlier, it is a summary paper so folks that are interested in the field will benefit from this. I am not an expert in this specific field so I am not sure how much of the audience for this conference will be interested in the paper.

Its a 50 page paper mostly presented as paragraphs so there is a lot of content in the paper that goes through what is in the field, which can be another thing that can put off folks from picking up the paper

**Claims And Evidence:**

Yes

**Claims Explanation:**

Yes, it's a summary paper so it's mostly quoting other people's work alongside with appropriate evidence presented. Paper does a good job of laying out the various dimension along which they want to talk about the work and then keep quoting what has been done in the field, what has worked and what has not.

**Requested Changes:**

There are 2 changes that are easy to make that can make this paper much more effective

1. better presentation. switching away from long paragraph way of presenting the summary to engage the reader more. and highlight which parts when comparing are more relevant and relatively new.

2. contributions in terms of new contributions and how to advance the field are pretty light and obvious. while the paper is summary paper I understand the focus is not on how to advance the field, but since the paper talks about it, it might be useful to make it more beefy rather than keeping it really light.

---

### Review · Reviewer_zfGJ · 2026-06-22

**Summary Of Contributions:**

This survey maps affective computing onto three capabilities of an emotionally intelligent agent: emotion understanding, affective cognition, and emotional expression synthesis across text, speech, and face. It follows a PRISMA protocol that screens 2,500 records down to 298, and organizes each capability around a recurring challenge taxonomy of data, model, problem-nature, multimodal, and LLM issues, pairing each sub-challenge with representative solutions in summary tables. The central claim is integrative scope, namely covering all three capabilities and all three expressive modalities in one framework.

This survey would be useful to new researchers to the field of affective computing. The Challenge-to-solution table would be particularly useful for indexing relevant papers tackling a specific problem. The main weakness is that the subsections are enumerating the papers instead of synthesizing them for the audience. There is little analysis or comparison of which methods win under which conditions. There is also no consolidated datasets and benchmarks table.

**Audience:**

Yes

**Audience Explanation:**

This survey would be useful to researchers new to the field of affective computing to gain some general knowledge of the field. The Challenge-to-solution table would be particularly useful for indexing relevant papers tackling a specific problem. Many recent papers are included (papers published in 2026, though some are listed on arxiv only).

**Broader Impact Concerns:**

No broader impact concerns.

**Claims And Evidence:**

Yes

**Claims Explanation:**

The citations are valid except two need fixing:
- there is a placeholder citation printed as "Authors. Flexible thinking... 2025"
- there is also an unrendered "cite zhou2022" in Section 6.2.1

**Requested Changes:**

- The quantitative accounting does not reconcile. The 32/30/25/13 percent split in Section 2.4 implies about 95 understanding and 89 expression papers, but the bar charts sum to 86 (Fig. 3), 107 (Fig. 4), and 71 (Fig. 5). The figures also reverse the paper's central ranking, showing expression as the largest category (107 vs. 86) where the text claims understanding is. Please reconcile the subtotals, percentages, and pie chart, and clarify any double-counting across capabilities. Ideally the per-category paper list should be provided.
- The two citation issues need to be fixed:
  -- there is a placeholder citation printed as "Authors. Flexible thinking... In Findings of EMNLP 2025, 2025"
  -- there is also an unrendered citation "Most EVC systems are implemented in this manner [cite zhou2022]" in Section 6.2.1
- The paper would be materially stronger with at least one comparative synthesis per major section stating which methods dominate under which conditions, and with tighter future-work sections that pose prioritized open problems rather than restating challenges already covered. The ethics discussion is present but thin for a dual-use topic and would be better consolidated into one subsection.

- Minor proofreading is also needed, for example "an LLM might lose an emotion" in Section 3.2.5 and "Emotional based leaning" in Figure 1.

---

### Review · Reviewer_82VC · 2026-07-06

**Summary Of Contributions:**

The paper provides a systematic review of affective computing approaches for intelligent systems.
It analyzes the recent publications to identify challenges and future directions.

The paper summarizes the challenges and current trends with the list of publications they have.

There are a couple of aspects that need addressing. Please see below.

**Audience:**

Yes

**Audience Explanation:**

If or when the above clarifications are addressed, the paper could be of use to the Machine Learning, especially the Affective Computing, community.

Although the findings of the paper is limited in terms of the extent of the papers reviewed, the future directions and challenges referred in the paper could inspire researchers.

**Broader Impact Concerns:**

In my opinion, the paper does not need an ethical implications statement. Ethical considerations are already addressed in terms of the reviewed papers. Although it would be better to create a separate section to combine all ethical considerations for higher readability.

**Claims And Evidence:**

No

**Claims Explanation:**

1) How do the authors define an intelligent agent?

2) The query provided in the manuscript is not appropriate for acquiring all the necessary publications.
The query only contains a subset of the keywords that address the emotional intelligence.
It does not include variations of the provided keywords
It does not contain keywords such as emotion elicitation, emotion experiences, gesture, emotion generation
The placement of OR and AND in the query limits the works they can acquire and eliminates a lot of studies that do not contain the exact match.
The summary claims cannot be confirmed without the actual query used.

3) They mention that they excluded studies that are published in arxiv or in proceedings if they are not highly cited. What is the threshold for this? More recent papers might be addressing several of these challenges and might not have the number of citations the authors used to cutoff their review.
Also, there is an ambiguous statement is section 2.1. The authors claim that their review focuses on works between 2017 and 2025. However in the next sentence they mention they inlcuded works between 1990 and 2025 to emphasize modern developments. Which statement is true?
Furthermore, the authors performed backward and forward cited papers of key publications. How did they determine a paper is a key publication?

4) Figure 6 provides an overview of an emotion recognition framework that does not properly address all different types of current approaches. The overview explained in the corresponding sections also has the same issue.
There are studies that does not extract manual features or features that can be easily named with known concepts.
They can extract features with feature extraction models that can be black box.
The multimodal path shows early fusion and late fusion approaches as a continuous process. Although this can be employed, not all studies uses both approaches.
In its current form the figure is misleading.

5) The categories extracted for each concept in challenges are not consistent. For example, in recognition, computational resources is part of multimodal and in synthesis, it is a separate category or part of LLMs only.
Also computational cost is not only relevant for multimodal or LLM approaches. It is a common limitation.

6) The paper also does not address data collection and annotation challenges explicitly, which is one of the major challenges in Affective Computing. In this specific context, this is an issue since proper training and validation in intelligent systems requires addressing this challenge.

7) The paper does not address how the papers utilize different emotion models. The interpretation of several challenges and solutions listed should differ based on the model used since a categorical emotion model and a continuous emotion model is addressed differently.

**Requested Changes:**

The clarification of the above mentioned items is needed to be included into the manuscript.

The citations are made with in-text citations. Should follow (Name, Year) or [Number] styles for normal citations and Name (Year) of [Number] for narrative citations. Most probably, this is a latex conflict issue, which I had encountered myself in some templates.